# Early-childhood linear growth faltering in low- and middle-income countries

Jade Benjamin-Chung[1,2,3 ✉], Andrew Mertens[2], John M. Colford Jr[2], Alan E. Hubbard[2], Mark J. van der Laan[2], Jeremy Coyle[2], Oleg Sofrygin[2], Wilson Cai[2], Anna Nguyen[1,2], Nolan N. Pokpongkiat[2], Stephanie Djadjadi[2], Anmol Seth[2], Wendy Jilek[2], Esther Jung[2], Esther O. Chung[2], Sonali Rosete[2], Nima Hejazi[2], Ivana Malenica[2], Haodong Li[2], Ryan Hafen[4], Vishak Subramoney[5], Jonas Häggström[6], Thea Norman[7], Kenneth H. Brown[8], Parul Christian[9], Benjamin F. Arnold[10,11 ✉] & The Ki Child Growth Consortium*

Globally, 149 million children under 5 years of age are estimated to be stunted (length more than 2 standard deviations below international growth standards)[1,2]. Stunting, a form of linear growth faltering, increases the risk of illness, impaired cognitive development and mortality. Global stunting estimates rely on cross-sectional surveys, which cannot provide direct information about the timing of onset or persistence of growth faltering—a key consideration for defining critical windows to deliver preventive interventions. Here we completed a pooled analysis of longitudinal studies in low- and middle-income countries (*n* = 32 cohorts, 52,640 children, ages 0–24 months), allowing us to identify the typical age of onset of linear growth faltering and to investigate recurrent faltering in early life. The highest incidence of stunting onset occurred from birth to the age of 3 months, with substantially higher stunting at birth in South Asia. From 0 to 15 months, stunting reversal was rare; children who reversed their stunting status frequently relapsed, and relapse rates were substantially higher among children born stunted. Early onset and low reversal rates suggest that improving children's linear growth will require life course interventions for women of childbearing age and a greater emphasis on interventions for children under 6 months of age.

In 2018, 149 million children under 5 years of age (22% globally) were stunted (length-for-age z-score (LAZ) > 2 standard deviations below the median of the growth standard for age and sex), with the largest burden in South Asia and Africa[1,2]. Early-life stunting is associated with increased risk of mortality[3], diarrhoea, pneumonia and measles in childhood[4,5] and impaired cognition and productivity in adulthood[6–8]. Global income would increase by an estimated US$176.8 billion per year if linear growth faltering could be eliminated[9]. The World Health Organization (WHO) 2025 global nutrition targets[10] and Sustainable Development Goal 2.2.1 (ref. 11) propose to reduce stunting prevalence among children under 5 years from 2012 levels by 40% by 2025.

In low-resource settings, the first thousand days of life—including the prenatal period—is considered the critical window in which to intervene to prevent stunting[12]. Intrauterine growth restriction and preterm birth are strongly associated with stunting at 24 months of age[13]. Most linear growth faltering occurs by the age of 2 years, and 70% of absolute length deficits by the age of 5 years occur before the age of 2 years[6]. Children who experience linear growth faltering before the age of 2 years can experience catch-up growth at older ages, particularly with

improvements to their nutrition, health and environment[14–18]. However, the extent of catch-up growth depends on the timing and severity of early-life linear growth faltering[19].

Granular information about the age of linear growth faltering onset and its persistence in early life will best inform when and how to intervene with preventive measures. Yet, most studies of the global epidemiology of stunting have used nationally representative, cross-sectional surveys—predominantly Demographic and Health Surveys (DHS)—to estimate age-specific stunting prevalence[15,20–22]. Analyses of cross-sectional studies cannot identify longitudinal patterns of linear growth faltering or reversal. Further, they may be subject to survivor bias and fail to include those children most vulnerable to undernutrition. Few studies have estimated age-specific incidence within the first 2 years of life[23–27].

We estimated linear growth faltering incidence and reversal and linear growth velocity in 32 longitudinal cohorts in low- and middle-income countries (LMICs) with multiple, frequent measurements. The analysis provides new insights into the timing of onset and duration of linear growth faltering, with important implications for interventions.

[1]Department of Epidemiology & Population Health, Stanford University, Stanford, CA, USA. [2]Division of Epidemiology & Biostatistics, University of California, Berkeley, Berkeley, CA, USA. [3]Chan Zuckerberg Biohub, San Francisco, CA, USA. [4]Hafen Consulting, LLC, West Richland, WA, USA. [5]DVPL Tech, Dubai, United Arab Emirates. [6]Cytel Inc., Waltham, MA, USA. [7]Quantitative Sciences, Bill & Melinda Gates Foundation, Seattle, WA, USA. [8]Department of Nutrition, University of California, Davis, Davis, CA, USA. [9]Center for Human Nutrition, Department of International Health, Johns Hopkins Bloomberg School of Public Health, Baltimore, MD, USA. [10]Francis I. Proctor Foundation, University of California, San Francisco, San Francisco, CA, USA. [11]Department of Ophthalmology, University of California, San Francisco, San Francisco, CA, USA. *A list of authors and their affiliations appears at the end of the paper. ✉e-mail: jadebc@stanford.edu; ben.arnold@ucsf.edu

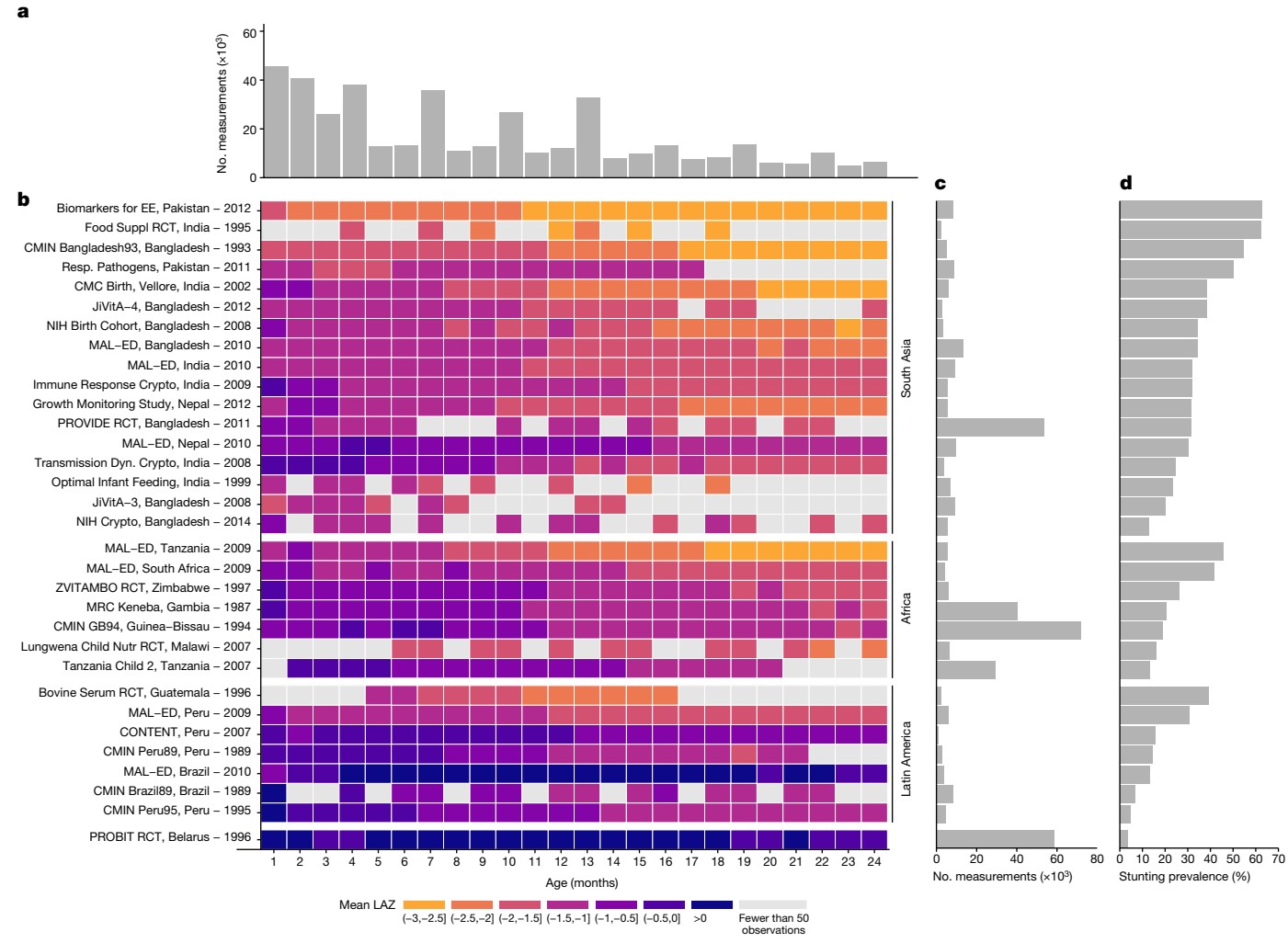

**Fig. 1 | Summaries of included Ki cohorts. a**, Number of observations (thousands) by age in months. **b**, Mean LAZ by age in months for each cohort. Cohorts are sorted by geographic region and mean LAZ. **c**, Number of observations contributed by each cohort. **d**, Overall stunting prevalence in each cohort, defined as proportion of measurements with LAZ < −2.

We found that linear growth faltering occurs very early in the prenatal and postnatal phase—before the age of 6 months when most postnatal linear growth interventions begin. Our findings confirm the importance of the first 1,000 days as a critical window to intervene to prevent linear growth faltering but motivate a renewed focus on prenatal and early postnatal interventions.

## Pooled longitudinal analyses

Here we report a pooled analysis of 32 longitudinal cohorts from 14 LMICs in South Asia, sub-Saharan Africa and Latin America followed between 1987 and 2017. Our objective was to estimate age-specific incidence and reversal of stunting and linear growth velocity from 0 to 24 months. Companion articles report results for child wasting (weight-for-length z-score < 2 standard deviations below the reference median)[28] and household, maternal and child-level risk factors associated with linear growth faltering[29]. These data were aggregated by the Bill & Melinda Gates Foundation Knowledge Integration (Ki) initiative and comprise approximately 100 longitudinal studies on child birth, growth and development. We included cohorts from the database that met five inclusion criteria: conducted in LMICs; had a median year of birth in 1990 or later; enrolled children between birth and the age of 24 months and measured their length and weight repeatedly over time; did not restrict enrolment to acutely ill children; and collected anthropometry measurements at least every 3 months (Extended Data Fig. 1). These criteria ensured that we could rigorously evaluate the timing and onset of stunting among children who were broadly representative of general populations in LMICs. Thirty-two cohorts met inclusion criteria, including 52,640 children and 412,458 total measurements from 1987 to 2017 (Fig. 1 and Supplementary Tables 1 and 2). Cohorts were located in South Asia (*n* = 17 cohorts in 4 countries), Africa (*n* = 7 in 6 countries), Latin America (*n* = 7 in 3 countries) and Eastern Europe (*n* = 1; Extended Data Fig. 2). Twenty-one cohorts measured children at least monthly, and 11 measured children every 3 months. Cohort sample sizes varied from 119 to 14,074 children. In most cohorts, more than 80% of enrolled children had LAZ measurements at each age of measurement (Extended Data Figs. 3 and 4).

We calculated LAZs using WHO 2006 growth standards[30]. We dropped 859 of 413,317 measurements (0.2%) because LAZ was unrealistic (>6 or <−6 z), and we defined stunting as LAZ < −2 and severe stunting as LAZ < −3 (ref. 30). Unless otherwise indicated, estimates that pool across cohorts used random-effects models fitted with restricted maximum-likelihood estimation[31,32]. Within each cohort, the monthly mean LAZ ranged from −3.06 to +1.31, and the monthly proportion stunted ranged from 0% to 91% (Fig. 1).

To assess Ki cohort representativeness, we compared LAZ from the Ki cohorts with contemporary population-based, cross-sectional DHS data in the same countries. Ki cohorts and DHS z-score distributions

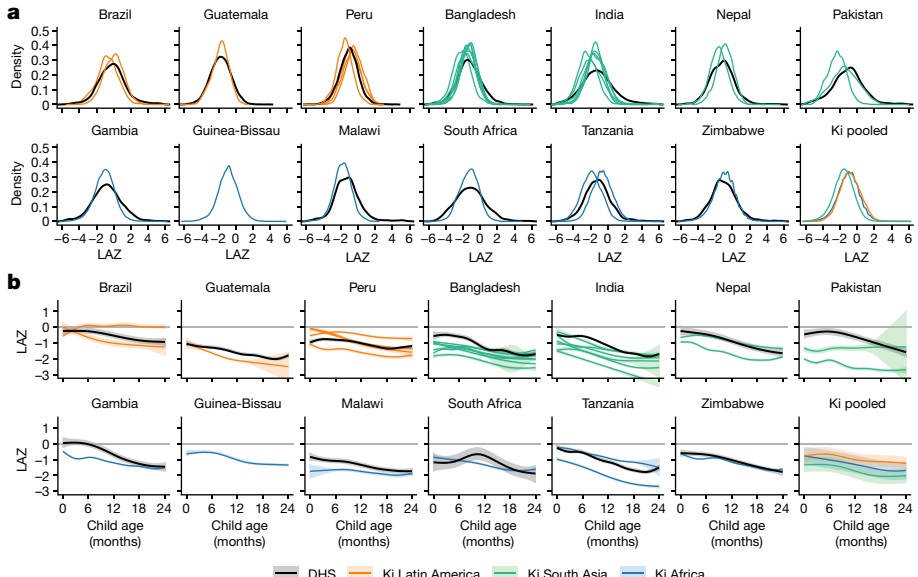

**Fig. 2 | LAZs by age and region. a**, Kernel density distributions of LAZ in DHS countries that overlap with Ki cohorts (black lines) and in each Ki cohort (coloured lines). **b**, Mean LAZs by age for DHS countries overlapping with Ki cohorts (black lines) and pooled across Ki longitudinal cohorts with at least quarterly measurement (coloured lines) estimated with cubic splines. Shaded bands are approximate 95% simultaneous confidence intervals. The DHS survey was not conducted in Guinea-Bissau during the study period. Each panel includes $n = 125,046$ children from DHS data and $n = 52,640$ children from Ki cohorts.

were similar (Fig. 2a). The distribution of LAZ was shifted to the left for Ki cohorts in South Asia compared to those in Latin America and Africa. Mean LAZ by age was generally lower in Ki cohorts than in DHS surveys, especially in South Asia, but was slightly higher at certain ages in two Peruvian cohorts (Fig. 2b).

## Growth faltering as a whole-population condition

In approximately half of cohorts, the 95th percentile of the LAZ distribution dropped below 0 by the age of 15 months (Extended Data Fig. 5). This pattern is consistent with the characterization of linear growth faltering as a 'whole-population' condition[21]. In most cohorts, as children aged, LAZ distributions shifted downwards (Extended Data Fig. 6), and standard deviations and skewness were similar across ages (Extended Data Fig. 7).

## Onset of stunting in early life

To measure the timing of stunting onset, we classified a child as a new incident case in three-month age periods if their LAZ dropped below −2 for the first time in that age period. The percentage of children that were stunted at birth ranged from 0.3% to 42% in each cohort and was 13% overall (Fig. 3a). The percentage that experienced incident stunting onset between birth and 3 months ranged from 7% to 57% in each cohort and was 18% overall. Children stunted between birth and 3 months accounted for 23% of all children who experienced stunting by the age of 24 months (69% of children). Trends were similar for severe stunting (Supplementary Note 1).

Early onset of stunting was consistent across geographic regions and countries with different levels of health spending, poverty and under-5 mortality. Very early-life stunting onset was most common in South Asia, where 20% of children were stunted at birth, and another 21% became stunted by the age of 3 months (Fig. 3a). In Africa and Latin America, the percentage stunted at birth was also lower than the percentage that became stunted between birth and the age of 3 months. In all regions, the rate of onset was lower at subsequent ages. Overall, the proportion stunted at birth or by the age of 3 months was higher, and onset was lower at subsequent ages in countries with a lower proportion

of gross domestic product devoted to health spending, higher child mortality and a higher percentage of the population living on less than US$1.90 per day (Extended Data Figs. 8–10).

We summarized age trends in LAZ stratified by geographic region and timing of stunting onset (Fig. 3b and Extended Data Fig. 11). Among children stunted at birth, LAZ differed markedly between geographic regions: mean LAZ rose in the first month of life in all regions and then remained close to −0.5 in Latin America, close to −2 in Africa and close to −2.5 in South Asia. Regional differences were less pronounced among children stunted at later ages, although children in South Asian cohorts had consistently lower mean LAZ than children from African and Latin American cohorts. Children who became stunted between birth and the age of 6 months started at low birth LAZ (mean = −2.7) and had moderate rates of decline, whereas children who became stunted between ages 6 and 15 months started at higher birth LAZ (mean = −1.4) but had much faster rates of decline in LAZ, from above −1 $z$ at birth to below −2 $z$ by the age of 15 months. Children who were never stunted still experienced a drop of approximately 0.5 $z$ in mean LAZ from birth to the age of 15 months in all regions, showing that even children not classified as 'stunted' on average experienced substantial, postnatal linear growth faltering.

## Stunting reversal and relapse

We reasoned that: lower than average linear growth (LAZ < 0) would persist among children who experienced stunting reversal (that is, LAZ increased from below −2 to above −2); and children who experienced stunting reversal would experience stunting relapse at later ages. To test these hypotheses, we classified a child's change in stunting status from birth to the age of 15 months among monthly measured cohorts. The percentage stunted was highest at birth and declined steadily to 3.3% per month by the age of 4 months (Fig. 4a), a pattern that was most marked in South Asia (Extended Data Fig. 12). Proportions of new and relapse stunting exceeded those of reversal at all ages, new results that illustrate the underlying dynamics of a gradually accumulating stunting burden as children age: by the age of 15 months, 34.0% of children were stunted, 50.5% had ever been stunted, and 16.5% had experienced stunting reversal and were no longer stunted (Fig. 4a). Stunting relapse following reversal ranged from 2.0 to 3.5%

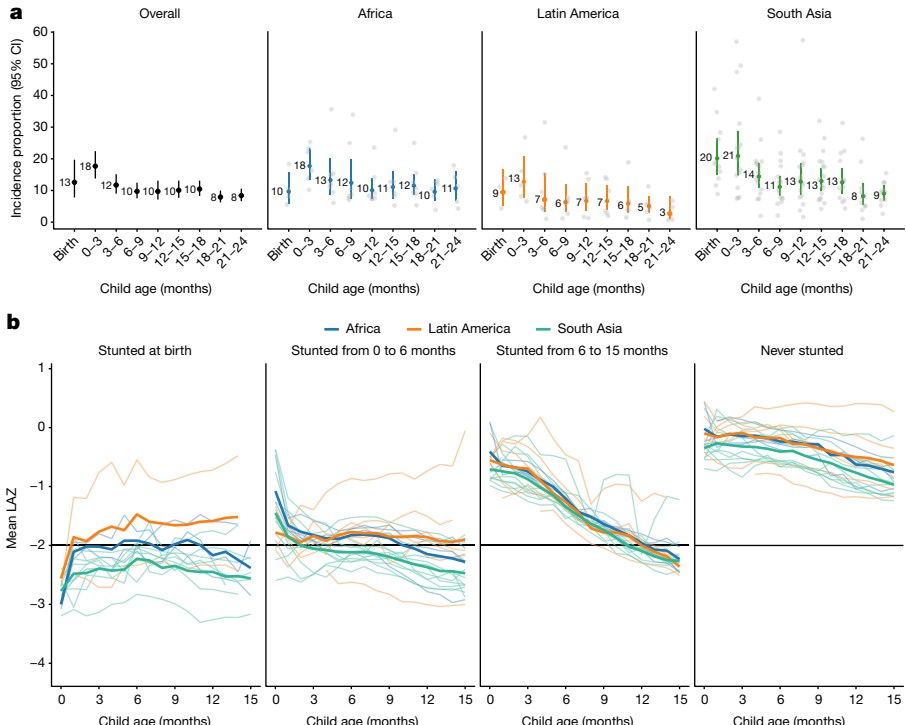

**Fig. 3 | Incidence of stunting and mean LAZ by age and region. a**, Proportion of children experiencing incident stunting onset overall (*n* = 19–32 studies; *n* = 11,929–42,902 children) and stratified by region (Africa: *n* = 4–8 studies, *n* = 5,529–15,837 children; Latin America: *n* = 3–7 studies, *n* = 413–1,528 children; South Asia *n* = 11–17 studies, *n* = 4,514–17,802 children). '0–3' includes the age of 2 days up to 3 months. Analyses include cohorts with at least quarterly measurements; vertical bars indicate 95% confidence intervals. Grey points indicate cohort-specific estimates. The median *I²* statistic measuring heterogeneity in each meta-analysis was 95 (interquartile range (IQR) = 77–98) overall, 85 (IQR = 83–97) in Africa, 67 (IQR = 45–87) in Latin America and 91 (IQR = 79–96) in South Asia. **b**, Mean LAZ stratified by age of incident stunting

from birth to the age of 15 months (*n* = 21 cohorts that measured children at least monthly between birth and the age of 15 months, *n* = 11,243 children). Horizontal black lines indicate stunting the cutoff of −2 LAZ. 'Never stunted' includes children who did not become stunted by the age of 15 months. Pooled results were derived from random-effects models with restricted maximum-likelihood estimation. Thinner lines indicate cohort-specific estimates. The median *I²* statistic measuring heterogeneity in each meta-analysis was 91 (IQR = 83–96) overall, 85 (IQR = 63–94) in Africa, 94 (IQR = 88–96) in Latin America and 85 (IQR = 78–92) in South Asia. Extended Data Fig. 11 contains pooled means from **b** with 95% confidence intervals.

per month from ages 6 to 15 months, and patterns were similar across regions (Extended Data Fig. 12). In South Asia, stunting reversal declined as children aged, but percentages were stable across ages in Africa and Latin America; overall reversal was slightly less common in Latin America (Extended Data Fig. 12).

To assess whether a child's birth length influenced their propensity to recover from stunting, we summarized stunting, relapse and reversal rates stratified by birth LAZ subgroup in monthly measured cohorts (Fig. 4b). Eighty-six per cent of children who ever became stunted had LAZ < 0 at birth. Percentages of stunting relapse increased with age and were generally higher among children who were born stunted. Stunting reversal was more common at young ages for children born with LAZ < −2, which probably reflects regression to the mean. After the age of 6 months, stunting reversal levels were similarly low among children with birth LAZ < −2 (<7% per month) and birth LAZ −2 to 0 (<5% per month). These results indicate that linear growth faltering at birth is a key determinant of children's linear growth trajectories in early life, recovery is rare among all children who become stunted by the age of 15 months, and children who are stunted at birth are more prone to transient stunting reversal followed by stunting relapse.

We next studied the distribution of improvement in LAZ by age of stunting reversal to assess whether reversal at different ages was associated with more sustained improvement in LAZ. For children who experienced stunting reversal, we summarized the LAZ distribution at subsequent ages and estimated the mean difference in LAZ measured at older ages compared to when stunting was reversed. At the time of stunting reversal,

the LAZ distribution mode was close to the −2 cutoff (Fig. 5a and Extended Data Fig. 13). As children aged, LAZ distributions gradually shifted downwards, illustrating that linear growth deficits continued to accumulate. Among children who experienced stunting reversal before the age of 6 months, mean difference in LAZ 9 months later was −0.69 (95% confidence interval −0.84, −0.55; cohort-specific range: −1.04, −0.22; Fig. 5b). Children who were older at the time of reversal experienced a larger decline in subsequent LAZ compared to that of younger children (Fig. 5b). Overall, improvements in LAZ following stunting reversal were neither sustained nor large enough to erase linear growth deficits and did not resemble a biological recovery process for most children.

## Growth velocity by age and sex

We defined linear growth velocity as a child's change in length between two time points divided by the number of months between the time points (cm per month). From 0 to 3 months, cohort-specific length velocity ranged from below the 1st percentile of the WHO standard to above the 50th for boys and above the 75th percentile for girls (Fig. 6a). At subsequent ages, length velocity in each cohort was mostly between the 15th and 50th percentiles of the WHO standard, except in one cohort in Belarus, which had a higher length velocity. Larger deficits at the youngest ages were consistent with highest incidence of stunting from birth to the age of 3 months (Fig. 3a). From the ages of 3 to 24 months, on average, children's change in length was between 0.75 and 1.25 cm per month. We also estimated within-child rates of LAZ change per

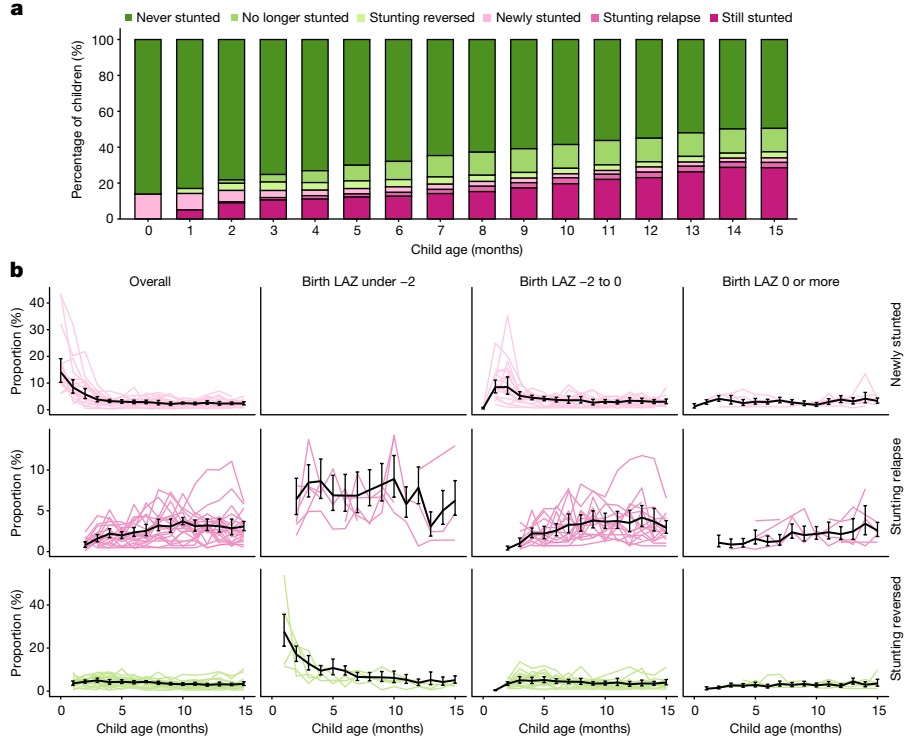

**Fig. 4 | Stunting reversal and relapse. a**, Percentage of children with stunting reversal and relapse by age. **b**, Proportion of new stunting, stunting relapse and stunting reversal by age and birth LAZ, defined as the first LAZ measurement before the age of 30 days. The black line presents estimates pooled using random effects with restricted maximum-likelihood estimation (*n* = 168 models); in 11 models, alternative pooling methods were used because the restricted maximum-likelihood estimator did not converge (fixed-effects *n* = 8 models; maximum-likelihood *n* = 3 models). Coloured lines indicate cohort-specific estimates. In the panel for birth LAZ under −2 and newly stunted children, no data are shown because all children were stunted at birth by definition. Vertical black error bars indicate 95% confidence intervals. The number of children ranged from 1,831 to 9,965 in the panels for birth LAZ under −2, 34,427 to 43,753 in the panels for birth LAZ = −2 to 0, and 10,450 to 14,862 in the panels for birth LAZ 0 or more. The median *I*² statistic measuring heterogeneity in each meta-analysis was 55 (IQR = 47–70). Extended Data Fig. 12 presents similar estimates stratified by geographic region. Both panels include data from 21 cohorts in 10 countries with at least monthly measurement (*n* = 11,435). Both panels contain data up to the age of 15 months because in most cohorts, measurements were less frequent above the age of 15 months.

month, which compares changes in a child's length relative to the WHO standard over time. The difference in LAZ within child per month was largest from 0 to 3 months; after the age of 3 months, the mean change in LAZ within child was <0.3 between different age intervals (Fig. 6b). Generally, velocity within age was higher in Latin America than in South Asia and Africa (Extended Data Fig. 14).

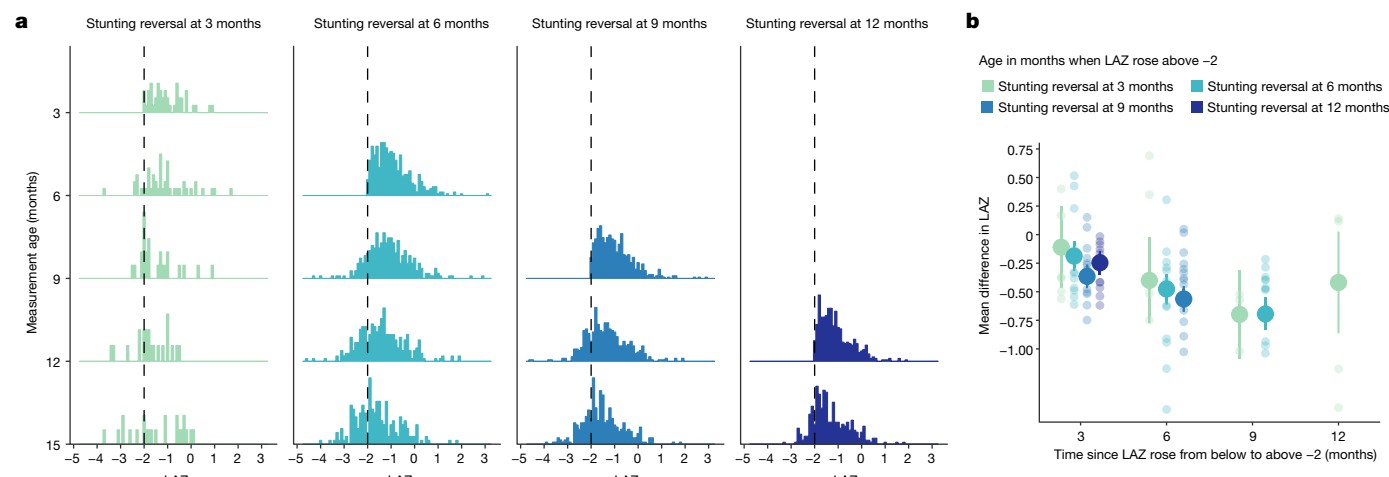

**Fig. 5 | Subsequent LAZ among children with stunting reversal at different ages. a**, Distribution of LAZ at subsequent measurements among children who experienced stunting reversal at the ages of 3, 6, 9 and 12 months. Vertical black dashed lines indicate stunting the cutoff of −2 LAZ. **b**, Mean difference in LAZ following stunting reversal at each subsequent age of measurement by age of reversal. Smaller, partially transparent points indicate cohort-specific estimates. Estimates include data from 21 cohorts in 10 countries with at least monthly measurement (*n* = 11,271). Vertical bars indicate 95% confidence intervals. All panels contain data up to the age of 15 months because in most cohorts, measurements were less frequent above the age of 15 months.

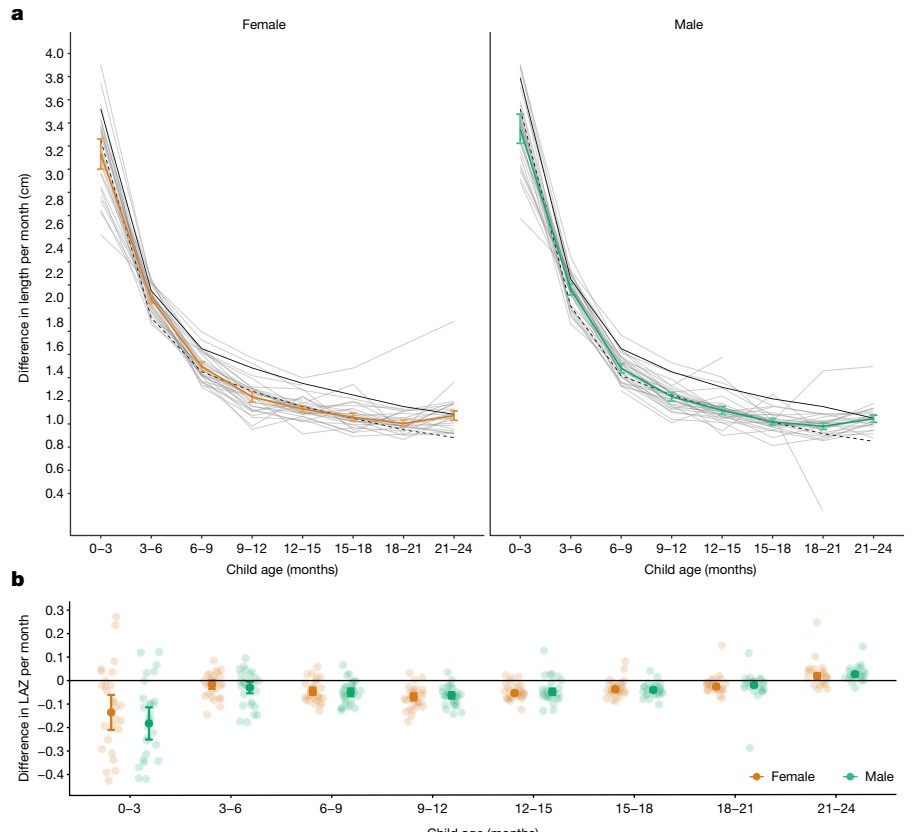

**Fig. 6 | Linear growth velocity by age and sex. a**, Within-child difference in length in centimetres per month stratified by age among male (green line) and female (orange line) children; 25th percentile of the WHO growth velocity standards (dashed black lines); and the 50th percentile (solid black line). Light grey lines indicate cohort-specific linear growth velocity curves. The median $I^2$ statistic measuring heterogeneity in each meta-analysis was 90 (IQR = 83–94).

**b**, Within-child difference in LAZ per month by age and sex. Smaller partially transparent points indicate cohort-specific estimates. The median $I^2$ statistic measuring heterogeneity in each meta-analysis was 89 (IQR = 78–92). Both panels include 32 Ki cohorts in 14 countries that measured children at least quarterly (*n* = 52,640 children) pooled using random-effects models fitted with restricted maximum-likelihood estimation. Vertical bars indicate 95% confidence intervals.

## Discussion

This large-scale analysis of 32 longitudinal cohorts from LMICs revealed new insights into the timing, persistence and recurrence of linear growth faltering from birth to the age of 2 years. Previous cross-sectional studies found that stunting prevalence increased gradually with age[15,20–22]. By contrast, we found that incident stunting onset was highest between birth and the age of 3 months, a pattern consistent across geographic regions, and was most pronounced in countries with a lower proportion of gross domestic product devoted to health spending, higher under-5 mortality rates and higher poverty levels (Fig. 3a and Extended Data Figs. 8–10). Stunting at birth was a key predictor of children's linear growth trajectories to the age of 15 months: stunting relapse in the first year of life was substantially higher among children who were stunted at birth compared to those who were not born stunted (Fig. 4b). The burden and persistence of very early-life linear growth faltering was most stark in South Asia, where 20% of children were stunted at birth (Fig. 3a) and children who were stunted at birth had a mean LAZ of approximately −2.5 at all subsequent ages, substantially lower than that for children in other regions (Fig. 3b). Most children who experienced stunting reversal continued to experience linear growth deficits, and more than 20% who achieved reversal were stunted again at later measurements (Fig. 5a). Even among children who never met criteria for stunting, mean LAZ steadily declined by over 0.5 *z* by the age of 15 months (Fig. 3b)—a result that shows that linear growth faltering among children in LMICs is a whole-population phenomenon, with both stunted and not stunted children experiencing suboptimal growth trajectories in early life[21].

Two key conclusions from a recent series on child maternal and child undernutrition[33] were that improving children's linear growth will require a life course approach with an emphasis on women's health and that targeting interventions by age and geography may yield greater benefits than one-size-fits-all approaches. Our results provide new quantitative evidence that strengthens these conclusions and enables more precise statements about the extent of the whole-population burden, age windows for preventive interventions, and the uniquely high incidence and low reversal rates among children in South Asia compared with those in other geographic regions.

Highest stunting onset in the first 3 months of life and greater stunting relapse among children who were born stunted underscore the importance of pre-pregnancy and prenatal interventions to reduce stunting. These interventions include maternal micronutrient and macronutrient supplementation[34,35], increasing women's autonomy and education[36], reducing adolescent pregnancies in LMICs by delaying the age of marriage and first pregnancy[37], and promoting family planning[38]. Interventions to prevent prenatal infections, such as intermittent preventive treatment for malaria, may also increase fetal linear growth in regions where such infections co-occur with linear growth faltering[39]. Our finding that stunting incidence at birth was lower in countries with a greater level of national health expenditures suggests that overall investments in healthcare systems may also improve linear growth.

In South Asia in particular, where stunting at birth was highest, intervening to improve the health of women of childbearing age may be critical to improving children's linear growth. Previous work has identified South Asian women's nutrition before and during pregnancy and poor sanitation conditions as key contributors to stunting at birth[40].

Regarding sanitation, in 2020 the prevalence of open defaecation was 18% in sub-Saharan Africa, 12% in South Asia and 2% in Latin America, and access to basic sanitation was lower in sub-Saharan Africa than in South Asia[41]. Recent trials found that improving household-level sanitation did not improve children's linear growth, but studies did not measure impacts on mothers[42]. A more likely explanation for higher stunting at birth in South Asia is women's nutritional status. Prevalence of low body mass index in women is highest in South Asia (24%), with much higher prevalence in some geographic hotspots[33]. In addition, 40–70% of women in South Asia are less than 150 cm tall[43], and the prevalence of infants born small for gestational age is 34% in South Asia compared to 17% in sub-Saharan Africa and 9% in Latin America[44]. Our analysis of risk factors for stunting in a companion paper in this series reports that maternal height, weight and body mass index were the strongest predictors of stunting at birth and child linear growth trajectories[29]. These findings point to the need to tailor interventions to the unique factors influencing women's nutrition and prenatal health in South Asia.

In this study, 25% of children became stunted between birth and the age of 6 months, yet few child nutrition interventions are recommended by the WHO in this age range. In the neonatal period, those interventions include delayed cord clamping, neonatal vitamin K administration and kangaroo mother care[45]. Beyond the neonatal period, the sole recommended intervention is exclusive breastfeeding[45], which substantially reduces the risk of mortality and morbidity but has not been found to reduce infant stunting[4,46–49]. Further research is needed to identify interventions that prevent linear growth faltering between birth and the age of 6 months, including nutritional support of the lactating parent and the vulnerable infant[50]. Interventions may need to focus on upstream risk factors, such as maternal pre-conception and prenatal health and nutrition, and microbiota.

We found that 31% of children became stunted during the complementary feeding phase (age of 6–24 months). Meta-analyses evaluating the effectiveness of interventions during this phase on linear growth have reported modest impacts of lipid-based nutrient supplements[51], modest or no impact of micronutrient supplementation[52], and no impact of water and sanitation improvements, deworming or maternal education[52]. The dearth of effective postnatal interventions to improve linear growth motivates renewed efforts to identify alternative, possibly multisectoral, interventions and to improve intervention targeting and implementation[53,54].

There were several limitations to the analyses. First, length estimates may be subject to measurement error; stunting reversal and relapse analyses that rely on thresholds are more sensitive to such errors. However, detailed assessments of measurement quality indicated that measurement quality was high across cohorts (Supplementary Note 2). Second, estimates of LAZ at birth using the WHO child growth standards overestimate stunting in preterm infants[55]. Accurate estimates of gestational age were not available in included cohorts; seven cohorts measured gestational age by recall of last menstrual period or newborn examination, and one cohort measured gestational age by ultrasound. In a sensitivity analysis adjusting for gestational age pooling across cohorts that measured it, stunting prevalence at birth was 1% lower (Extended Data Fig. 15). Third, included cohorts were not inclusive of all countries in the regions presented here, and linear growth faltering was more common in included African and South Asian cohorts than in corresponding contemporary representative surveys. The consistency between attained linear growth patterns in this and nationally representative DHS surveys (Fig. 2) suggests that overall, our results have reasonably good external validity. For growth velocity, the cohorts represented populations close to the 25th percentile of international standards (Fig. 5a). Fourth, the included cohorts measured child length every 1–3 months, and ages of measurement varied, so different numbers of children and cohorts contributed to each estimate. However, when we repeated analyses in cohorts with monthly measurements from birth to 24 months (n = 18 cohorts in 10 countries, 10,830 children),

results were similar (Supplementary Note 3). Finally, our inferences are limited to the first 2 years of life as very few included studies measured children at older ages. Other studies, however, have found that stunting status in early life is associated with health outcomes later in life, and the timing and extent of early-life linear growth faltering is associated with the magnitude of later catch-up growth[6–8,16,17,19].

## Conclusion

Current WHO 2025 global nutrition targets and Sustainable Development Goal 2.2.1 aim to reduce stunting prevalence among children under 5 years by 2025. Our findings suggest that defining stunting targets at earlier ages (for example, stunting by 3 or 6 months) would help focus attention on the period when interventions may be most impactful. In addition, our results motivate a life course approach that targets interventions to women of childbearing age and includes interventions for children during their first months of life.

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

**The Ki Child Growth Consortium**

**Souheila Abbeddou[12], Linda S. Adair[13], Tahmeed Ahmed[14], Asad Ali[15], Hasmot Ali[16], Per Ashorn[17,18], Rajiv Bahl[19], Mauricio L. Barreto[20], France Begín[21], Pascal Obong Bessong[22], Maharaj Kishan Bhan[23], Nita Bhandari[24], Santosh K. Bhargava[25], Zulfiqar A. Bhutta[26], Robert E. Black[27], Ladaporn Bodhidatta[28], Delia Carba[29], Ines Gonzalez Casanova[30], William Checkley[27], Jean E. Crabtree[31], Kathryn G. Dewey[32], Christopher P. Duggan[33], Caroline H. D. Fall[34], Abu Syed Golam Faruque[14], Wafaie W. Fawzi[35], José Quirino da Silva Filho[36], Robert H. Gilman[27], Richard L. Guerrant[37], Rashidul Haque[14], Sonja Y. Hess[32], Eric R. Houpt[37], Jean H. Humphrey[27], Najeeha Talat Iqbal[15], Elizabeth Yakes Jimenez[38,39], Jacob John[40], Sushil Matthew John[41], Gagandeep Kang[42], Margaret Kosek[37], Michael S. Kramer[43,44], Alain Labrique[9], Nanette R. Lee[29], Aldo Ângelo Moreira Lima[36], Mustafa Mahfuz[14], Tjale Cloupas Mahopo[45], Kenneth Maleta[46], Dharma S. Manandhar[47], Karim P. Manji[48], Reynaldo Martorell[30], Sarmila Mazumder[24], Estomih Mduma[49], Venkata Raghava Mohan[40], Sophie E. Moore[50,51], Ishita Mostafa[14], Robert Ntozini[52], Mzwakhe Emanuel Nyathi[53], Maribel Paredes Olortegui[54], William A. Petri[37], Prasanna Samuel Premkumar[40], Andrew M. Prentice[51], Najeeb Rahman[15], Harshpal Singh Sachdev[55], Kamran Sadiq[15], Rajiv Sarkar[40], Naomi M. Saville[56], Saijuddin Shaikh[16], Bhim P. Shrestha[57], Sanjaya Kumar Shrestha[58], Alberto Melo Soares[36], Bakary Sonko[51], Aryeh D. Stein[30], Erling Svensen[59], Sana Syed[15,37], Fayaz Umrani[15], Honorine D. Ward[60], Keith P. West Jr[9], Lee Shu Fune Wu[9], Seungmi Yang[43] & Pablo Penataro Yori[37]**

[12]Department of Public Health and Primary Care, Ghent University, Ghent, Belgium. [13]Department of Nutrition, University of North Carolina at Chapel Hill, Chapel Hill, NC, USA. [14]International Centre for Diarrhoeal Disease Research, Bangladesh, Dhaka, Bangladesh. [15]Department of Pediatrics and Child Health, The Aga Khan University, Karachi, Pakistan. [16]JiVitA Project, John Hopkins Bangladesh, Rangpur, Bangladesh. [17]Center for Child, Adolescent, and Maternal Health Research, Department of Medicine and Health Technology, Tampere University, Tampere, Finland. [18]Tampere University Hospital, Tampere, Finland. [19]World Health Organization, Geneva, Switzerland. [20]Center of Data and Knowledge Integration for Health, Fundação Oswaldo Cruz, Salvador, Brazil. [21]UNICEF, New York, NY, USA. [22]HIV/AIDS & Global Health Research Programme, University of Venda, Thohoyandou, South Africa. [23]Indian Institute of Technology, New Delhi, India. [24]Centre for Health Research and Development, Society for Applied Studies, New Delhi, India. [25]Sunder Lal Jain Hospital, New Delhi, India. [26]Centre of Excellence in Women & Child Health, Institute for Global Health & Development, The Aga Khan University, Karachi, Pakistan. [27]Johns Hopkins University Bloomberg School of Public Health, Baltimore, MD, USA. [28]Armed Forces Research Institute of Medical Sciences, Bangkok, Thailand. [29]USC Office of Population Studies Foundation Inc., University of San Carlos, Cebu, The Philippines. [30]Rollins School of Public Health, Emory University, Atlanta, GA, USA. [31]Leeds Institute for Medical Research, St. James's University Hospital, University of Leeds, Leeds, UK. [32]Department of Nutrition and Institute for Global Nutrition, University of California Davis, Davis, CA, USA. [33]Center for Nutrition, Boston Children's Hospital, Boston, MA, USA. [34]MRC Lifecourse Epidemiology Centre, University of Southampton, Southampton, UK. [35]Department of Global Health and Population, Harvard TH Chan School of Public Health, Boston, MA, USA. [36]Federal University of Ceará, Fortaleza, Brazil. [37]University of Virginia, Charlottesville, VA, USA. [38]Department of Pediatrics, University of New Mexico Health Sciences Center, Albuquerque, NM, USA. [39]Department of Internal Medicine, University of New Mexico Health Sciences Center, Albuquerque, NM, USA. [40]Christian Medical College, Vellore, India. [41]Low Cost Effective Care Unit, Christian Medical College, Vellore, India. [42]Translational Health Science and Technology Institute, Faridabad, India. [43]McGill University, Montreal, Quebec, Canada. [44]McGill University Health Centre, Montreal, Quebec, Canada. [45]Department of Nutrition, School of Health Sciences, University of Venda, Thohoyandou, South Africa. [46]School of Public Health and Family Medicine, College of Medicine, University of Malawi, Zomba, Malawi. [47]Mother and Infant Research Activities, Kathmandu, Nepal. [48]Department of Pediatrics and Child Health, Muhimbili University School of Health and Allied Sciences, Dar es Salaam, Tanzania. [49]Haydom Lutheran Hospital, Haydom, Tanzania. [50]Department of Women and Children's Health, King's College London, London, UK. [51]MRC Unit The Gambia at London School of Hygiene and Tropical Medicine, Banjul, The Gambia. [52]Zvitambo Institute for Maternal and Child Health Research, Harare, Zimbabwe. [53]Department of Animal Sciences, School of Agriculture, University of Venda, Thohoyandou, South Africa. [54]AB PRISMA, Lima, Peru. [55]Department of Pediatrics and Clinical Epidemiology, Sitaram Bhartia Institute of Science and Research, New Delhi, India. [56]Institute for Global Health, University College London, London, UK. [57]Health Research and Development Forum, Kathmandu, Nepal. [58]Walter Reed/AFRIMS Research Unit, Kathmandu, Nepal. [59]Haukeland University Hospital, Bergen, Norway. [60]Tufts Medical Center, Tufts University School of Medicine, Boston, MA, USA.

## Methods

The analysis was pre-specified at https://www.synapse.org/#!Synapse:syn11855121/wiki/513724.

### Study designs and inclusion criteria

We included all longitudinal observational studies and randomized trials available through the Ki project on April 2018 that met five inclusion criteria (Extended Data Fig. 1) as follows: studies that were conducted in LMICs (children in these countries have the largest burden linear growth faltering and are the key target population for preventive interventions); studies that had a median year of birth in 1990 or later (this restriction resulted in a set of studies spanning the period from 1987 to 2017 and excluded older studies that are less applicable to current policy dialogues); studies that enrolled children between birth and age 24 months and measured their length and weight repeatedly over time (we were principally interested in growth faltering during the first 1,000 days (including gestation), thought to be the key window for linear growth faltering); studies that did not restrict enrolment to acutely ill children (our focus on descriptive analyses led us to target, to the extent possible, the general population; we thus excluded some studies that exclusively enrolled acutely ill children, such as children who presented to hospital with acute diarrhoea or who were severely malnourished); studies that collected anthropometry measurements at least every 3 months (to ensure that we adequately captured incident episodes and recovery).

Thirty-two longitudinal cohorts in 14 countries followed between 1987 and 2017 met inclusion criteria. All children from each eligible cohort were included in the study. There was no evidence of secular trends in LAZ (Supplementary Note 4). We calculated cohort measurement frequency as the median days between measurements. If randomized trials found effects on growth within the intervention arms, the analyses were limited to the control arm. We included all measurements under 24 months of age, assuming months were 30.4167 days. We excluded extreme measurements of LAZ > 6 or LAZ < −6 following WHO growth standard recommendations[30]. In many studies, investigators measured length shortly after birth because deliveries were at home, but most measurements were within the first 7 days of life (Supplementary Note 5); for this reason, we grouped measurements in the first 7 days as birth measurements. Gestational age was measured in only five cohorts that measured birth length (three cohorts measured it by recall of last menstrual period; one measured it by newborn examination; one measured it by ultrasound); thus, we did not attempt to exclude preterm infants from the analyses.

### Quality assurance

The Ki data team assessed the quality of individual cohort datasets by checking the range of each variable for outliers and values that were not consistent with expectation. $z$-scores were calculated using the median of replicate measurements and the 2006 WHO child growth standards[30]. In a small number of cases, a child had two anthropometry records at the same age, in which case we used the mean of the records. Analysts reviewed bivariate scatter plots to check for expected correlations (for example, length by height; length, height or weight by age; length, height or weight by corresponding $z$-score). Once the individual cohort data were mapped to a single harmonized dataset, analysts conducted an internal peer review of published articles for completeness and accuracy. Analysts contacted contributing investigators to seek clarification about potentially erroneous values in the data and revised the data as needed.

### Outcome definitions

We used the following summary measures in the analysis.

**Incident stunting episodes.** Incident stunting episodes were defined as a change in LAZ from above −2 $z$ in the previous measurement to below −2 $z$ in the current measurement. Similarly, we defined severe stunting episodes using the cutoff of −3 $z$. Children were considered at risk of stunting at birth, so children born stunted were considered to have an incident episode of stunting at birth. Children were also assumed to be at risk of stunting at the first measurement in non-birth cohorts and trials. Children whose first measurement occurred after birth were assumed to have experienced stunting onset at the age halfway between birth and the first measurement. Most children were less than 5 days of age at their first measurement (Supplementary Note 5).

**Incidence proportion.** We calculated the incidence proportion of stunting during a defined age range (for example, 3–6 months) as the proportion of children at risk of becoming stunted who became stunted during the age range (the onset of new episodes).

**Changes in stunting status.** Changes in stunting status were classified using the following categories—never stunted: children with LAZ ≥ −2 at previous ages and the current age; no longer stunted: children who previously reversed their stunting status with LAZ ≥ −2 at the current age; stunting reversal: children with LAZ < −2 at the previous age and LAZ ≥ −2 at the current age; newly stunted: children whose LAZ was previously always ≥ −2 and with LAZ < −2 at the current age; stunting relapse: children who were previously stunted with LAZ ≥ −2 at the previous age and LAZ < −2 at the current age; still stunted: children whose LAZ was <−2 at the previous and current age.

**Growth velocity.** Growth velocity was calculated as the change in length in centimetres between two time points divided by the number of months between the time points. We compared measurements of change in length in centimetres per month to the WHO child growth standards for linear growth velocity[56]. We also estimated within-child rates of change in LAZ per month.

### Measurement frequency

Analyses of incidence and growth velocity (Figs. 3 and 5) included cohorts with at least quarterly measurements to include as many cohorts as possible. Analyses of stunting reversal (Fig. 4) were restricted to cohorts with at least monthly measurements to allow evaluation of changes in stunting status with higher resolution.

### Subgroups of interest

We stratified the above outcomes within the following subgroups: child age, grouped into one- or three-month intervals (depending on the analysis); the region of the world (Asia, sub-Saharan Africa, Latin America); sex of child; and the combinations of those categories. We obtained country-level data on the percentage of gross domestic product devoted to healthcare goods and spending from the United Nations Development Programme[57] and the percentage of the country living on less than US$1.90 per day and under-5 mortality rates from the World Bank[58]. In years without available data, we linearly interpolated values from the nearest years with available data and extrapolated values within 5 years of available data using linear regression models based on all available years of data. We also considered additional subgroups, including decade in which data were collected, gross domestic product[58], gender development index[57], gender inequality index[57], coefficient of human inequality[58] and the Gini coefficient[58]. However, for these variables, subgroup levels were strongly correlated with geographic region, making it impossible to separate the effects of each (Supplementary Table 3). Thus, we did not conduct subgroup analyses for these variables.

### Statistical analysis

All analyses were conducted in R version 3.4.2 (ref. 59).

**Estimation of mean LAZ by age in DHS and Ki cohorts.** We downloaded standard DHS individual recode files for each country from the

DHS Program website (https://dhsprogram.com/). We used the most recent standard DHS datasets for the individual women's, household, and height and weight datasets from each country. We obtained variables for country code, sample weight, cluster number, primary sampling unit and design stratification from the women's individual survey recode files. From the height and weight dataset, we used standard recode variables corresponding to the 2006 WHO growth standards for height-for-age.

After excluding missing observations, restricting to measurements of children of 0–24 months of age and restricting to z-scores within WHO-defined plausible values, surveys were collected from 1996 to 2018 in countries that overlapped with Ki cohorts with the exception of Guinea-Bissau because the DHS survey was not conducted there during the study period (Extended Data Table 1).

We classified countries into regions (South Asia, Latin America and Africa) using the WHO regional designations with the exception of the classification for Pakistan, which we included in South Asia to be consistent with previous linear growth studies using DHS[20]. One included cohort was from Belarus, and we chose to exclude it from region-stratified analyses as it was the only European study.

We estimated the age-stratified mean from ages of 0 to 24 months within each DHS survey, accounting for the complex survey design and sampling weights. We then pooled estimates of mean LAZ for each age in months across countries using a fixed-effects estimator (details below). We compared DHS estimates with mean LAZ by age in the Ki study cohorts, which we estimated using penalized cubic splines with bandwidth chosen using generalized cross-validation[60]. We used splines to estimate age-dependent mean LAZ in the Ki study cohorts to smooth any age-dependent variation in the mean caused by less frequently measured cohorts.

**Distribution models.** To investigate how the mean, standard deviation and skewness of LAZ distributions varied by age, we fitted linear models with skew-elliptical error terms using maximum-likelihood estimation. We fitted models separately by cohort.

**Fixed- and random-effects models.** Several analyses pooled results across study cohorts. We estimated each age-specific mean using a separate estimation and pooling step. We first estimated the mean in each cohort, and then pooled age-specific means across cohorts, while allowing for a cohort-level random effect. This approach enabled us to include the most information possible for each age-specific mean, while accommodating slightly different measurement schedules across the cohorts. Each cohort's data contributed only to LAZ or stunting incidence estimates at the ages for which it contributed data.

The primary method of pooling was using random-effects models. This modelling approach assumes that studies are randomly drawn from a hypothetical population of longitudinal studies that could have been conducted on children's linear growth in the past or future. We also fitted fixed-effects models as a sensitivity analysis (Supplementary Note 6); inferences about estimates from fixed-effects models are restricted to only the included studies[61].

Random-effects models assume that the true population outcomes $\theta$ are normally distributed ($\theta \sim N(\mu, \tau^2)$), in which $N$ indicates a normal distribution and $\theta$ has mean $\mu$ and variance $\tau^2$. To estimate outcomes in this study, the random-effects model is defined as follows for each study in the set of $i = 1, ..., k$ studies:

$$y_i = \mu + u_i + e_i \tag{1}$$

in which $y_i$ is the observed outcome in study $i$, $u_i$ is the random effect for study $i$, $\mu$ is the estimated outcome for study $i$, and $e_i$ is the sampling error within study $i$. The model assumes that $u_i \sim N(0, \tau^2)$ and $e_i \sim N(0, v_i)$, in which $v_i$ is the study-specific sampling variance. We fitted random-effects models using the restricted maximum-likelihood

estimator[31,32]. If a model failed to converge, we attempted to fit models with a maximum-likelihood estimator. If random-effects models failed to converge owing to the number of stunting cases being zero, we used a fixed-effects estimator. The quantity $\mu$ is the estimated mean outcome in the hypothetical population of studies (that is, the estimated outcome pooling across study cohorts).

We also fitted inverse-variance-weighted fixed-effects models defined as follows:

$$\bar{\theta}_w = \frac{\sum_{i=1}^{k} w_i \theta_i}{\sum_{i=1}^{k} w_i} \tag{2}$$

in which $\bar{\theta}_w$ is the weighted mean outcome in the set of $k$ included studies, and $w_i$ is a study-specific weight, defined as the inverse of the study-specific sampling variance $v_i$. $\theta_i$ is the estimate from study $i$.

For both types of outcome, we pooled binary outcomes on the logit scale and then back-transformed estimates after pooling to constrain confidence intervals between 0 and 1. Although the probit transformation more closely resembles common distributions for physiologic variables, in practice the logit transformation produces nearly identical estimates and is more convenient for estimation. For cohort-stratified analyses, which did not pool across studies, we estimated 95% confidence intervals using the normal approximation (Supplementary Note 7).

**Estimation of incidence.** We estimated incidence as defined above in 3-month age intervals within specific cohorts and pooled within region and across all studies (Fig. 3). Pooled analyses used random-effects models for the primary analysis and fixed-effects models for sensitivity analyses as described above.

**Estimation of changes in stunting status.** To assess fluctuations in stunting status over time, we conducted an analysis among cohorts with at least monthly measurements from birth to the age of 15 months to provide sufficient granularity to capture changes in stunting status. We estimated the proportion of children in each stunting category defined in the section 'Changes in stunting status' at each month from birth to the age of 15 months. To ensure that percentages summed to 100%, we present results that were not pooled using random effects. Analyses using random effects produced similar results (Supplementary Note 6.3).

To examine the distribution of LAZ among children with stunting reversal, we created subgroups of children who experienced stunting reversal at ages 3, 6, 9 and 12 months and then summarized the distribution of the children's LAZ at ages 6, 9, 12 and 15 months. Within each age interval, we estimated the mean difference in LAZ at older ages compared to the age of stunting reversal and estimated 95% confidence intervals for the mean difference. Pooled analyses used random-effects models for the primary analysis and fixed-effects models for sensitivity analyses as described above.

**Linear growth velocity.** We estimated linear growth velocity within 3-month age intervals stratified by sex, pooling across study cohorts (Fig. 6) as well as stratified by geographic region (Extended Data Fig. 14) and study cohort (Supplementary Note 7.4). Analyses included cohorts that measured children at least quarterly. We included measurements within a 2-week window around each age in months to account for variation in the age of each length measurement. Pooled analyses used random-effects models for the primary analysis and fixed-effects models for sensitivity analyses as described above (Supplementary Note 6.4).

### Sensitivity analyses
We conducted three sensitivity analyses. First, to assess whether inclusion of PROBIT, the single European cohort, influenced our overall pooled inference, we repeated analyses excluding the PROBIT cohort.

Results were very similar with and without the PROBIT cohort (Supplementary Note 8). Second, to explore the influence of differing numbers of cohorts contributing data at different ages, we conducted a sensitivity analysis in which we subset data to cohorts that measured anthropometry monthly from birth to the age of 24 months ($n = 21$ cohorts in 10 countries, 11,424 children; Supplementary Note 3). Third, we compared estimates pooled using random-effects models presented in the main text with estimates pooled using fixed-effects inverse-variance-weighted models. The random-effects approach was more conservative in the presence of study heterogeneity (Supplementary Note 6).

## Inclusion and ethics

This study analysed data that were collected in 14 LMICs that were assembled by the Bill & Melinda Gates Foundation Ki initiative. Datasets are owned by the original investigators that collected the data. Members of the Ki Child Growth Consortium were nominated by each study's leadership team to be representative of the country and study teams that originally collected the data. Consortium members reviewed their cohort's data within the Ki database to ensure external and internal consistency of cohort-level estimates. Consortium members provided substantial input on the statistical analysis plan, interpretation of results and manuscript writing. Per the request of consortium members, the manuscript includes cohort-level and regional results to maximize the utility of the study findings for local investigators and public health agencies. Analysis code has been published with the manuscript to promote transparency and extensions of our research by local and global investigators.

## Reporting summary

Further information on research design is available in the Nature Portfolio Reporting Summary linked to this article.

## Data availability

The data that support the findings of this analysis are a combination of data from multiple principal investigators and institutions. The data are available, upon reasonable request, to the requestor by contacting these individual principal investigators. The following link lists the individuals and their respective contact information that may be used to request access to the data: https://www.synapse.org/#!Synapse:syn51570682/wiki/. The analysis dataset is at https://www.synapse.org/#!Synapse:syn51570682/datasets/. This dataset is access-controlled and not available publicly for privacy reasons.

## Code availability

Replication scripts for this analysis are available at https://doi.org/10.5281/zenodo.11246818 and https://www.synapse.org/#!Synapse:syn51570682/.

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

**Acknowledgements** This research was financially supported by a global development grant (OPP1165144) from the Bill & Melinda Gates Foundation to the University of California, Berkeley. J.B.-C. acknowledges funding from the National Institute of Allergy and Infectious Diseases under award K01AI141616. J.B.-C. is a Chan Zuckerberg Biohub investigator. We also thank the following collaborators on the included cohorts and trials for their contributions to study planning, data collection and analysis: M. Sharif, S. Kerio, Urosa, Alveen, S. Hussain, V. Paudel, A. Costello, B. Torun, L. M. Locks, C. M. McDonald, R. Kupka, R. J. Bosch, R. Kisenge, S. Aboud, M. Wang, Azaduzzaman, A. A. Shamim, R. Haque, R. Klemm, S. Mehra, M. Mitra, K. Schulze, S. Taneja, B. Nayyar, V. Suri, P. Khokhar, J. E. Rohde, T. Kumar, J. Martines and all other members of the study staff and field teams. We also thank all study participants and their families for their important contributions.

**Author contributions** J.M.C., A.E.H., M.J.v.d.L. and B.F.A. obtained funding for the study. J.B.-C., B.F.A., A.M., J.M.C., A.E.H., M.J.v.d.L., P.C., K.H.B. and T.N. conceptualized the study. J.B.-C., B.F.A., A.M., J.M.C., A.E.H., M.J.v.d.L., P.C. and K.H.B. contributed to study design. J.B.-C., A.M., J.M.C, J.C., O.S., N.H., I. Malenica, A.E.H., M.J.v.d.L., K.H.B., P.C. and B.F.A. contributed to study methodology. Ki Child Growth Consortium members contributed data. J.B.-C., A.M., J.C., O.S., W. Cai, A.N., N.N.P., W.J., E.J., E.O.C., S.R., A.S., S.D., N.H., I. Malenica, H.L., R. Hafen., V.S., J.H. and T.N. curated the data. J.B.-C., A.M., J.C., O.S., W. Cai, N.H., I. Malenica, H.L., A.N., N.N.P., W.J., E.J., E.O.C., S.D., N.H., V.S. and B.F.A analysed the data. J.B.-C., A.M., A.N., N.N.P., S.D., A.S., J.C., R. Hafen, E.J., K.H.B., P.C. and B.F.A. conducted data visualization. J.B.-C., A.M. and B.F.A wrote the manuscript with input from all co-authors and Ki Child Growth Consortium members.

**Competing interests** T.N. is an employee of the Bill & Melinda Gates Foundation. P.C. is a former employee of the Bill & Melinda Gates Foundation, and K.H.B. is a former employee of and occasional consultant to the Bill & Melinda Gates Foundation. J.C., V.S., R. Hafen and J.H. work as research contractors financially supported by the Bill & Melinda Gates Foundation.

**Additional information**
**Correspondence and requests for materials** should be addressed to Jade Benjamin-Chung or Benjamin F. Arnold.

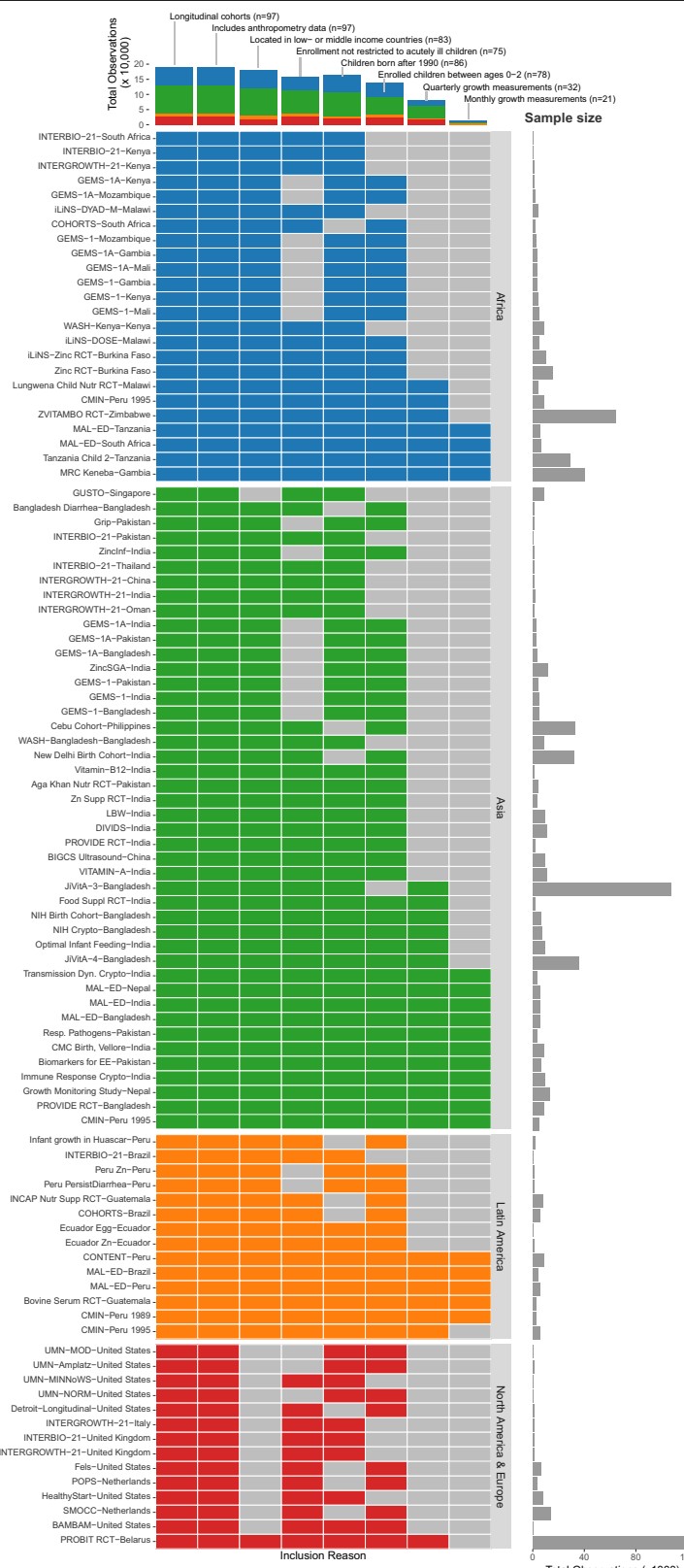

**Extended Data Fig. 1 | *ki* cohort selection.** Analyses focused on longitudinal cohorts to enable the estimation of prospective incidence rates and growth velocity. In April 2018, there were 97 longitudinal studies on GHAP. From this set, we applied five inclusion criteria to select cohorts for analysis. Our rationale for each criterion follows. (1) Studies were conducted in lower income or middle-income countries. (2) Studies had a median year of birth in 1990 or later. (3) Studies measured length and weight between birth and age 24 months. (4) Studies did not restrict enrollment to acutely ill children. (5) Studies collected anthropometry measurements at least every 3 months. Each colored cell indicates a criterion that was met. For studies that met all inclusion criteria, all cells in their row are colored. The bars at the top of the plot show the number of observations in each study that met each inclusion criterion by region.

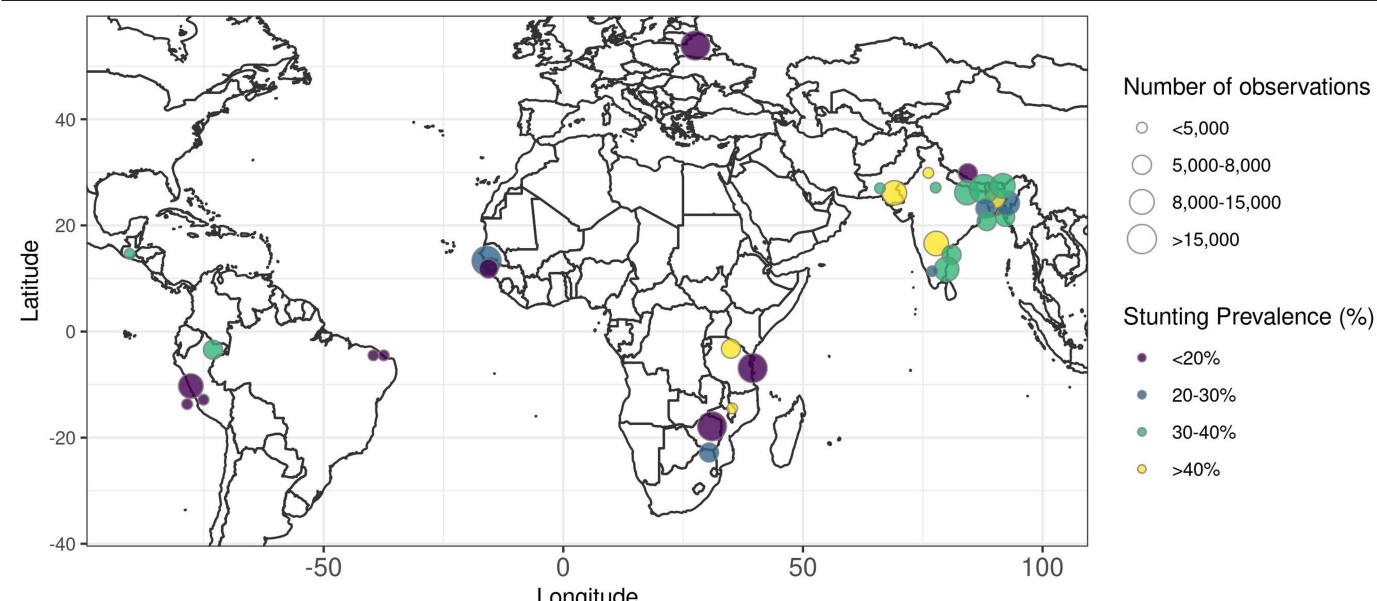

**Extended Data Fig. 2 | Stunting prevalence by geographic location of *ki* cohorts.** Locations are approximate, represented as nation-level centroids and jittered slightly for display. The size of each centroid indicates the number of observations contributing to each estimate. The color of each centroid indicates the level of stunting prevalence.

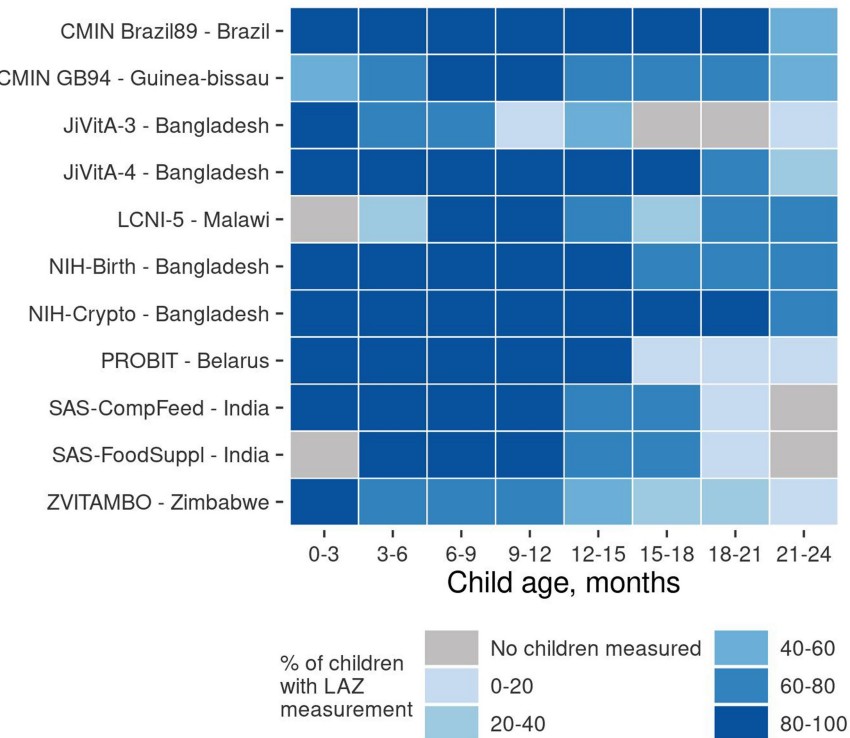

**Extended Data Fig. 3 | Percentage of enrolled children measured in each *ki* cohort with quarterly measurements.** Each colored cell indicates the percentage of children with a length-for-age Z-score measurement for a given cohort at a particular child age range. Gray cells indicate that no children had a length-for-age Z-score measurement for that age.

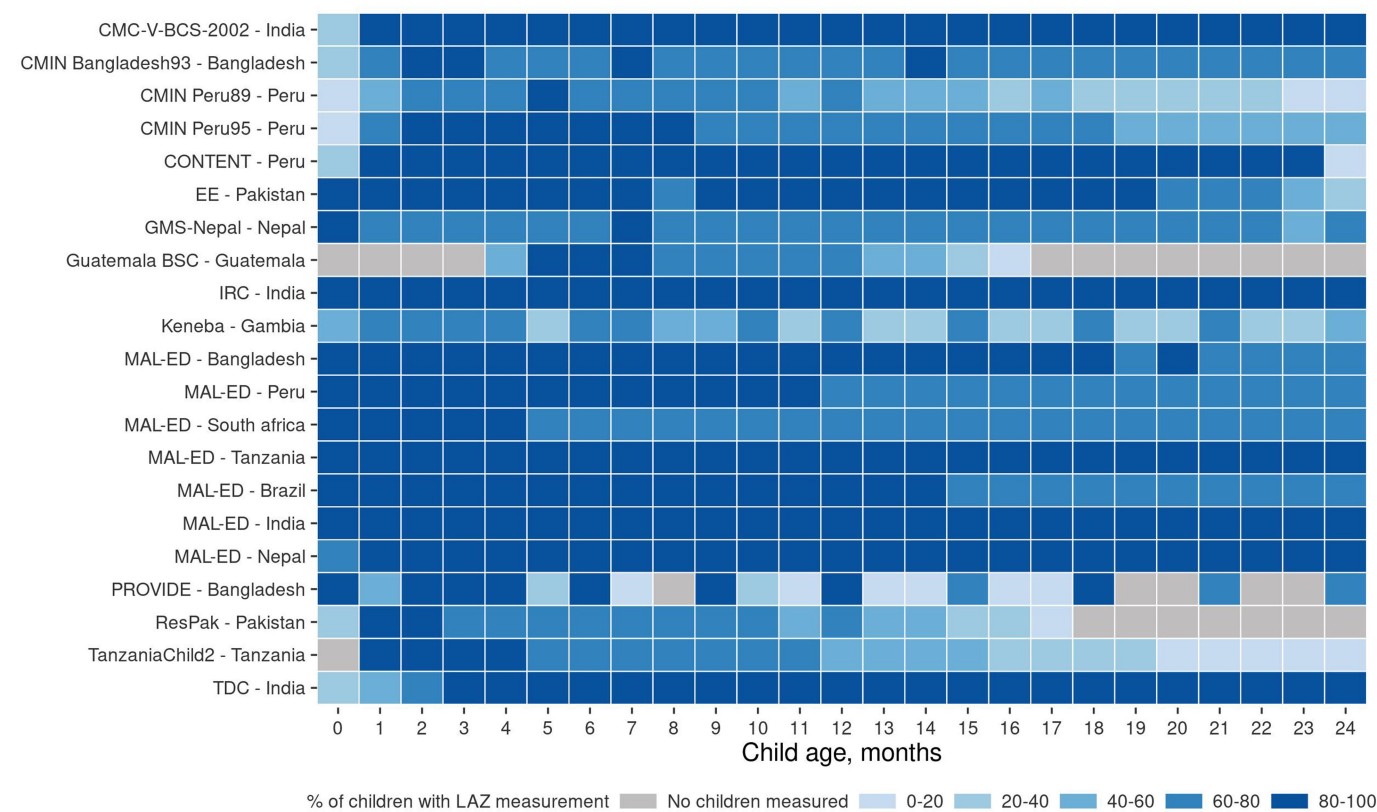

**Extended Data Fig. 4 | Percentage of enrolled children measured in each *ki* cohort with monthly measurements.** Each colored cell indicates the percentage of children with a length-for-age Z-score measurement for a given cohort at a particular child age. Gray cells indicate that no children had a length-for-age Z-score measurement for that age.

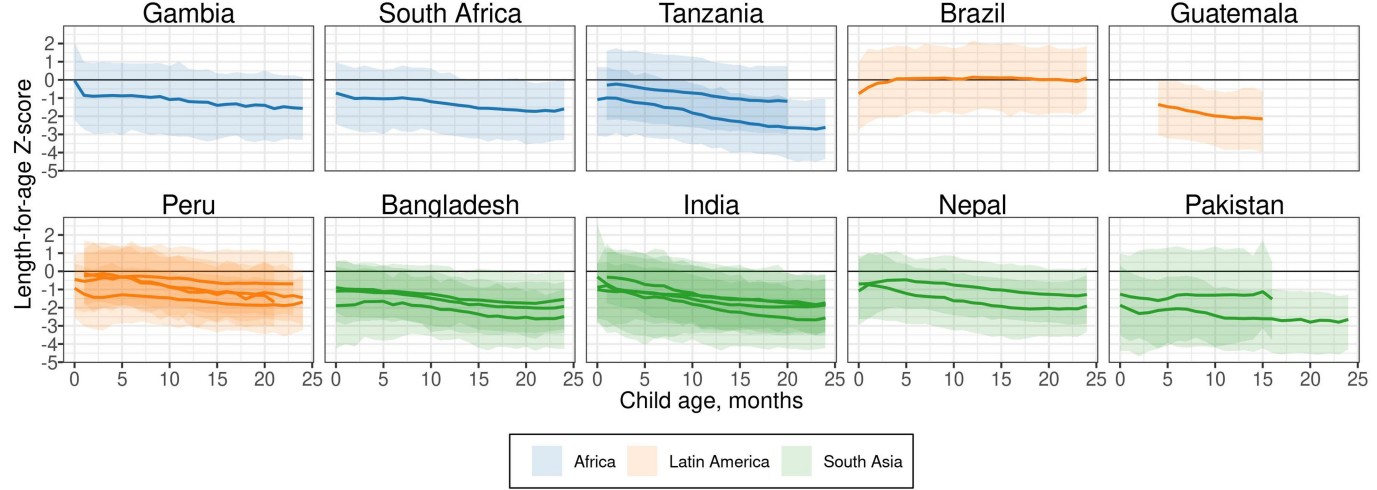

**Extended Data Fig. 5 | Distribution of length-for-age Z-score by age.** Mean, 5[th] and 95[th] percentile of length-for-age Z-score by age in *ki* longitudinal cohorts estimated with cubic splines in cohorts with at least monthly measurement. The shaded bands span the 5[th] to the 95[th] percentile of length-for-age Z-score in each cohort. The solid line indicates the mean in each cohort at each age (N = 21 cohorts that measured children at least monthly, N = 11,424 children).

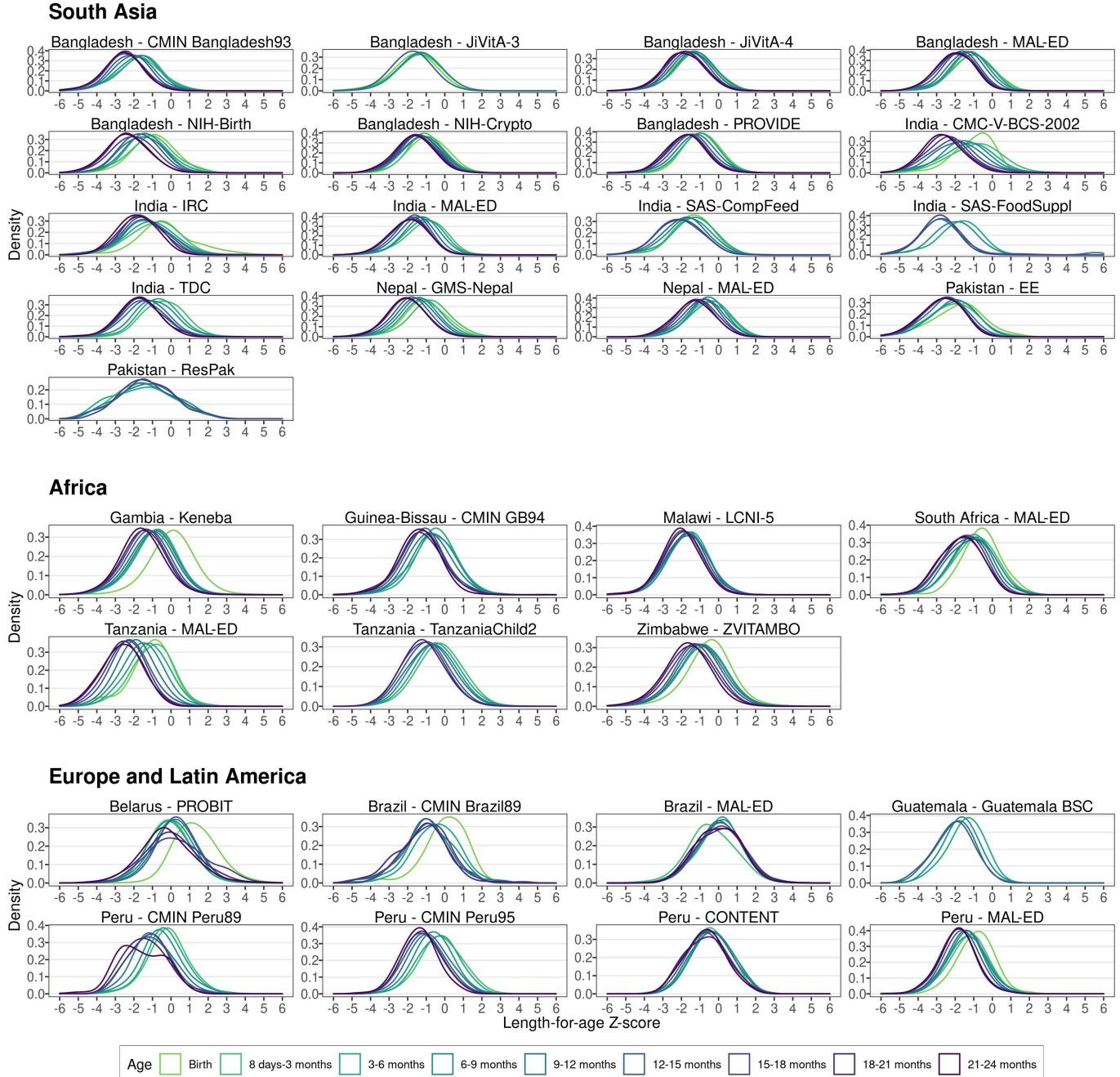

**Extended Data Fig. 6 | Kernel density of length-for-age Z-score by age and cohort.** In South Asia, includes data from 17 cohorts, 21,223 children, and 159,884 measurements. In Africa, includes 7 cohorts, 21,671 children, and 164,431 measurements. In Europe and Latin America, includes 8 cohorts, 9,746 children, and 88,143 measurements.

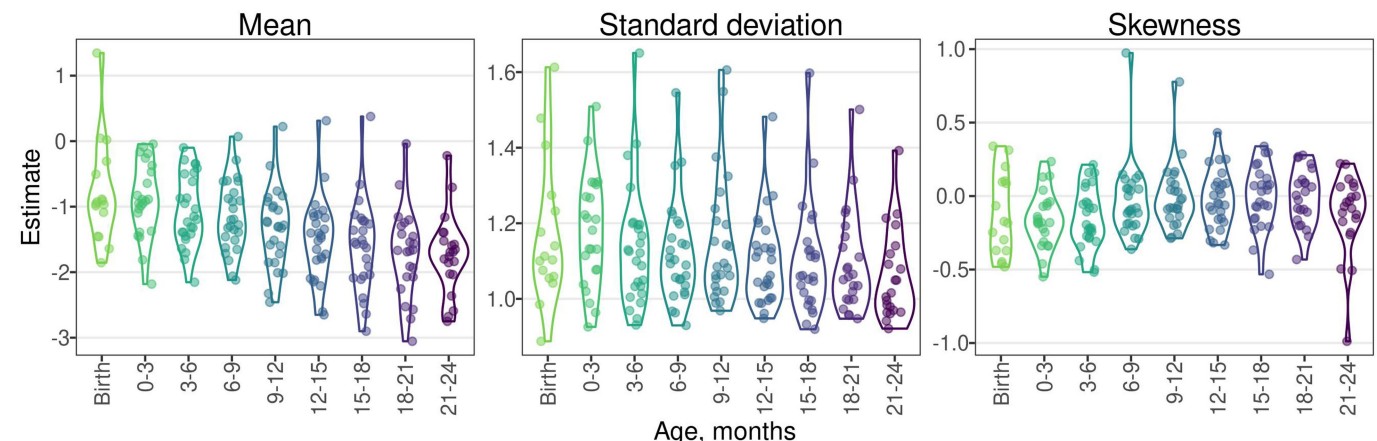

**Extended Data Fig. 7 | Parametric mean, standard deviation, and Pearson's index of skewness estimates by age and cohort.** Estimates were obtained from linear models with skew-elliptical error terms fit using maximum likelihood estimation. Includes 412,458 measurements from 52,640 children in 32 cohorts.

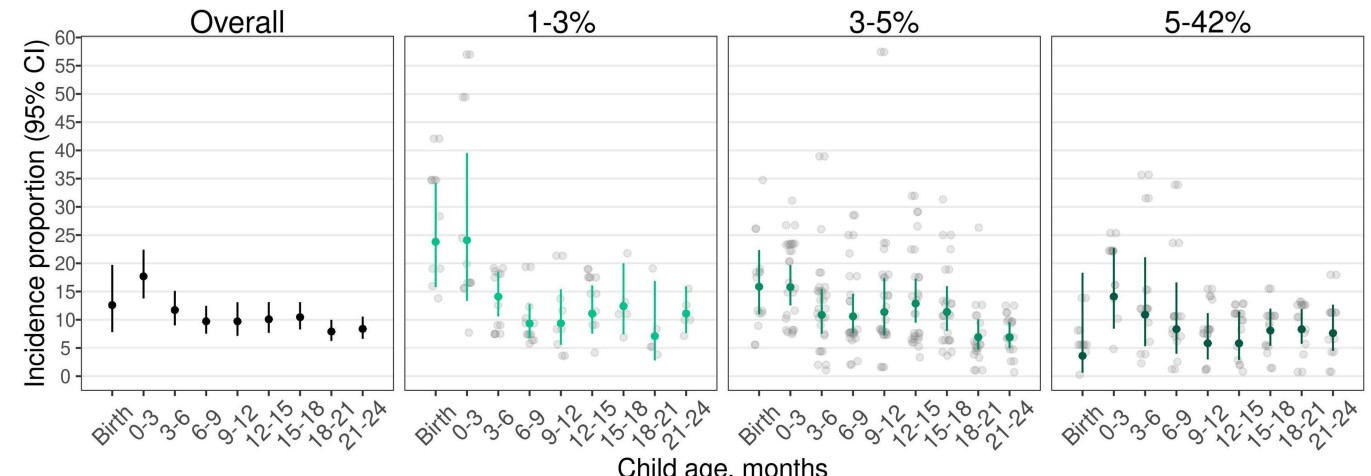

**Extended Data Fig. 8 | Incidence of stunting by age and national health expenditures as a percentage of gross domestic product.** Proportion of children experiencing incident stunting onset by national health expenditures as a percentage of gross domestic product (1–3%: N = 6–9 studies, N = 2,039–12,076 children; 3–5%: N = 11–19 studies, N = 4,467–16,030 children; 5–42%: N = 5–8 studies, N = 5,423–15,578 children). "0–3" includes age 2 days up to 3 months. Analyses include cohorts with at least quarterly measurements; vertical bars indicate 95% confidence intervals. Gray points indicate cohort-specific estimates. Pooled results were derived from random effects models with restricted maximum likelihood estimation.

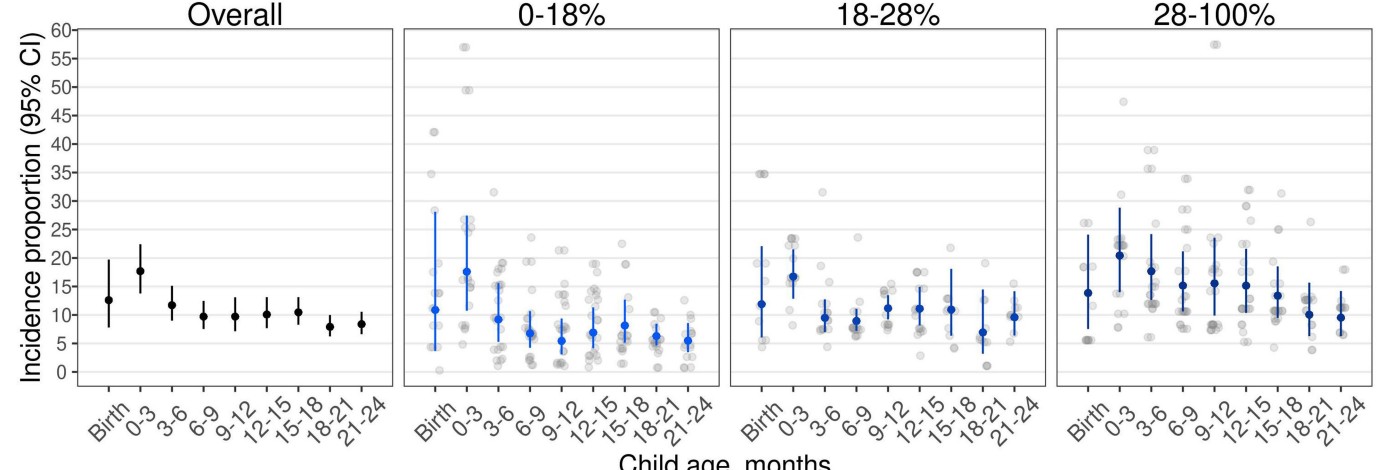

**Extended Data Fig. 9 | Incidence of stunting by age and national percentage of individuals living on less than $1.90 US per day.** Proportion of children experiencing incident stunting onset by national percentage of individuals living on less than $1.90 US per day (0–18%: N = 9–14 studies, N = 6,156–23,493 children; 18–28%: N = 7–10 studies, N = 1,602–14,639 children; 28–100%: N = 5–11 studies, N = 2,333–7,622 children). "0–3" includes age 2 days up to 3 months. Analyses include cohorts with at least quarterly measurements; vertical bars indicate 95% confidence intervals. Gray points indicate cohort-specific estimates. Pooled results were derived from random effects models with restricted maximum likelihood estimation.

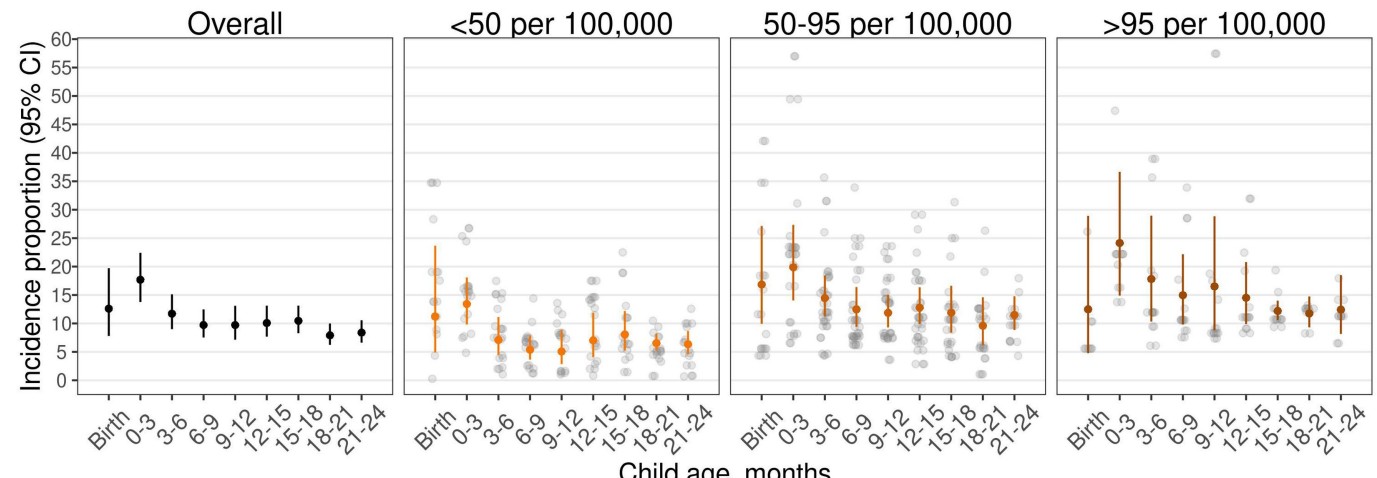

**Extended Data Fig. 10 | Incidence of stunting by age and national under-5 mortality rate.** Proportion of children experiencing incident stunting onset by national under-5 mortality rate (<50 per 100,000: N = 10–13 studies, N = 4,170–17,997 children; 50–95 per 100,000: N = 9–18 studies, N = 3,244–12,296 children; >95 per 100,000: N = 3–7 studies, N = 4,450–15,177 children). "0–3" includes age 2 days up to 3 months. Analyses include cohorts with at least quarterly measurements; vertical bars indicate 95% confidence intervals. Gray points indicate cohort-specific estimates. Pooled results were derived from random effects models with restricted maximum likelihood estimation.

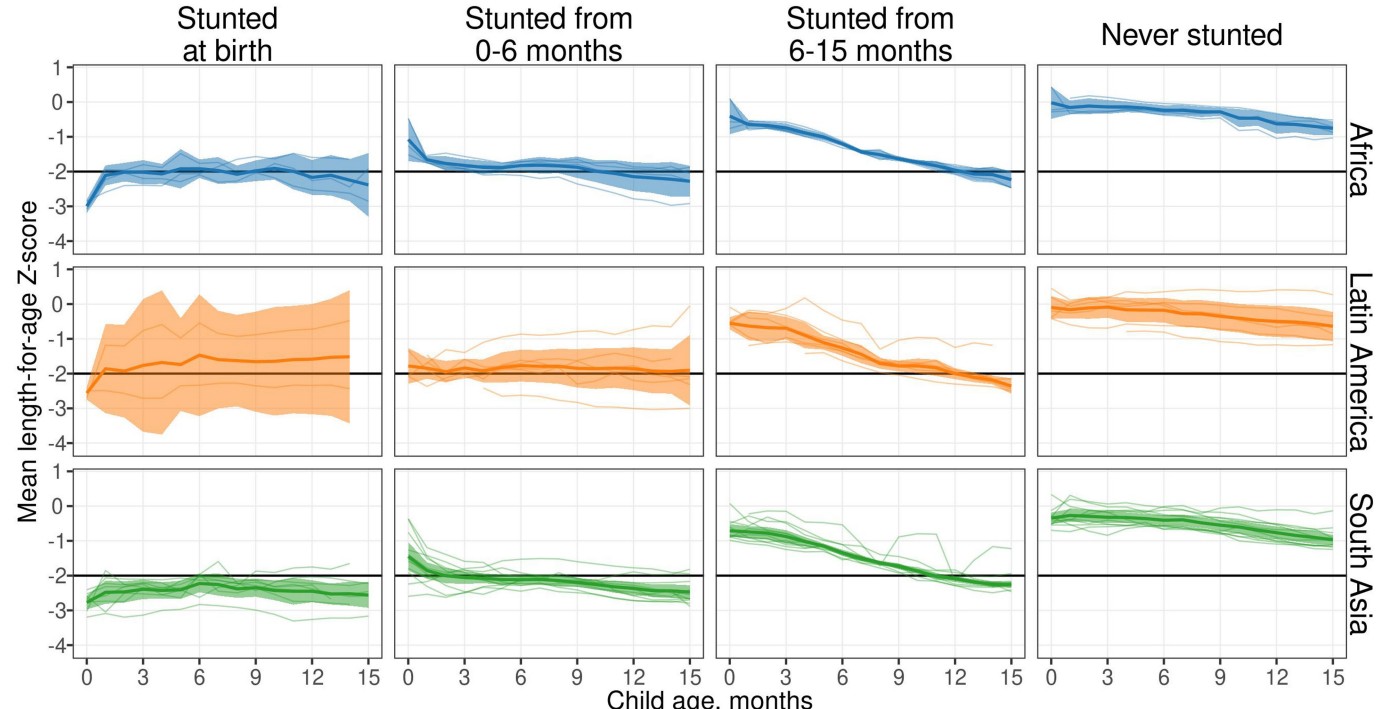

**Extended Data Fig. 11 | Mean LAZ by age and region with 95% confidence intervals.** Mean length-for-age Z-score (LAZ) stratified by age from birth to age 15 months (N = 21 cohorts that measured children at least monthly between birth and age 15 months, N = 11,243 children). "Never stunted" includes children who did not become stunted by age 15 months. Shaded ribbons indicate 95% confidence intervals. Pooled results were derived from random effects models with restricted maximum likelihood estimation. Thinner lines indicate cohort-specific estimates.

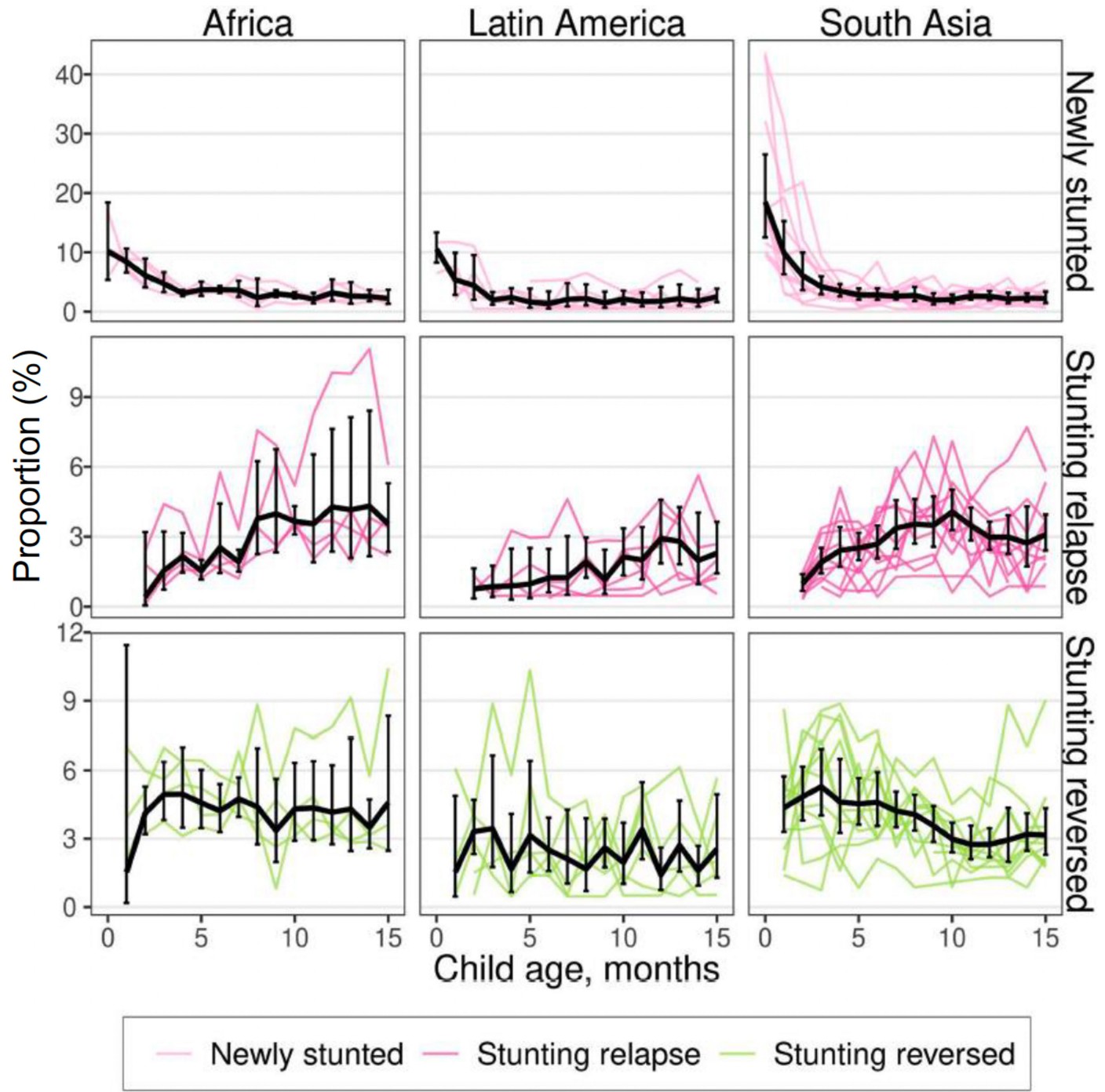

**Extended Data Fig. 12 | Stunting reversal and relapse by region.** Proportion of new stunting, stunting relapse, and stunting reversal by age. The black line presents estimates pooled using random effects with restricted maximum likelihood estimation. Colored lines indicate cohort-specific estimates. Vertical black error bars indicate 95% confidence intervals. Estimates include data from 21 cohorts in 10 countries with at least monthly measurement (N = 11,435) and are presented through age 15 months because in most cohorts, measurements were less frequent above 15 months.

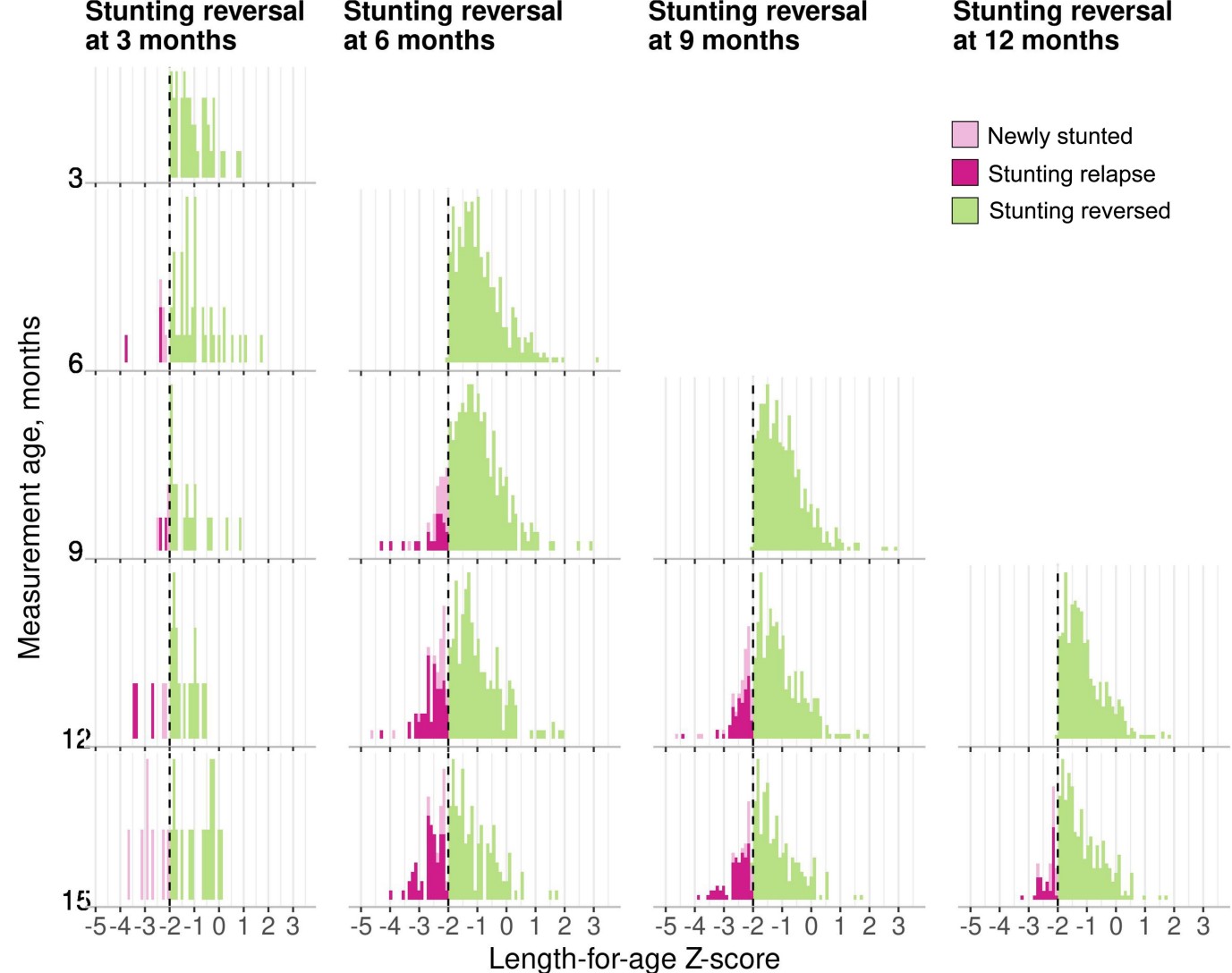

**Extended Data Fig. 13 | Distribution of LAZ at subsequent measurements after stunting reversal.** Includes data from 21 cohorts in 10 countries with at least monthly measurement (N = 11,271 children). All panels contain data up to age 15 months because in most cohorts, measurements were less frequent above 15 months. The underlying data is equivalent to that displayed in Fig. 5a; this figure uses a different color palette to emphasize observations in which children experienced new stunting, stunting relapse, or stunting reversal.

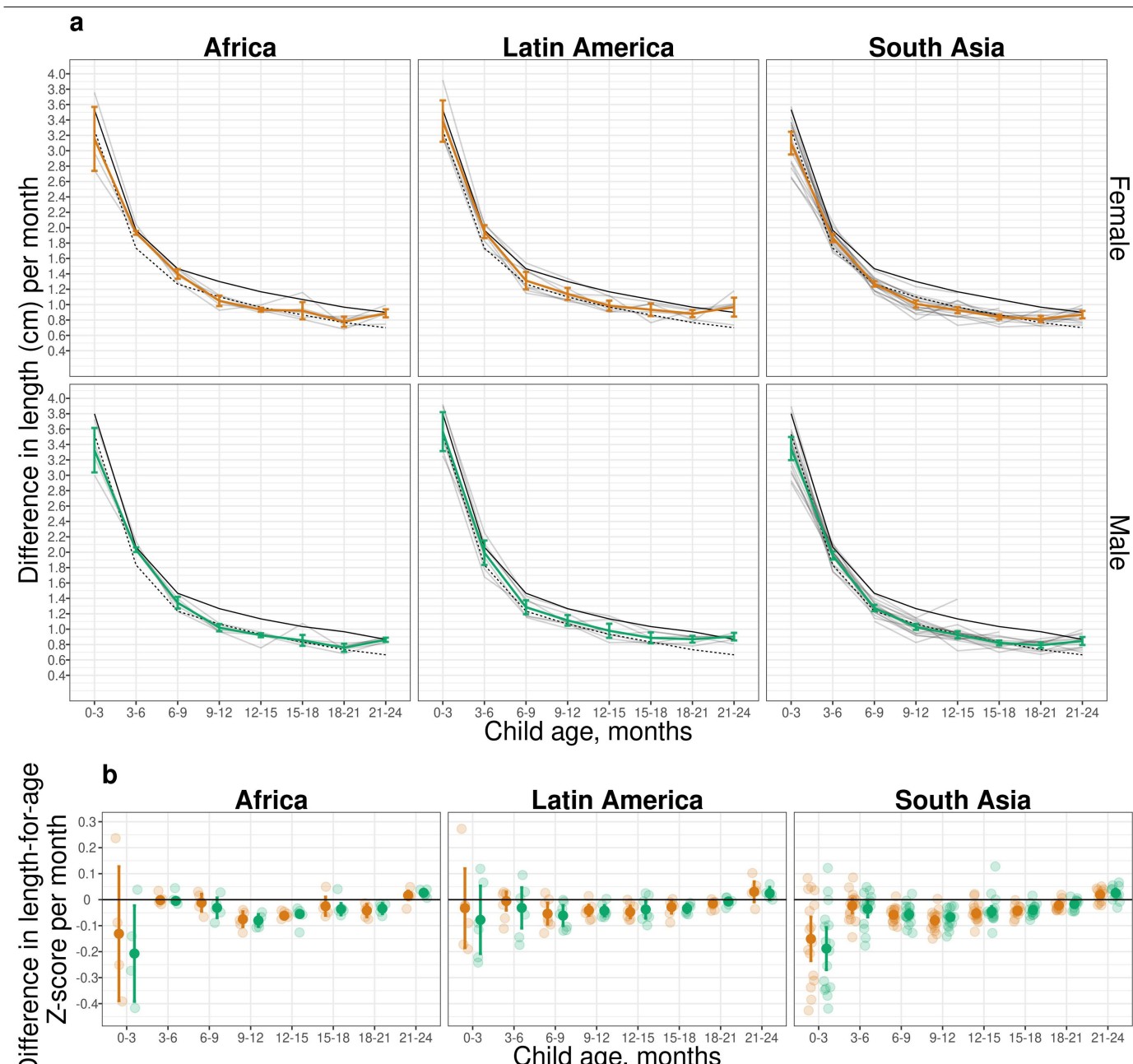

**Extended Data Fig. 14 | Linear growth velocity by age and sex stratified by region. a)** Within-child difference in length in centimeters per month stratified by age, sex, and region. Dashed black line indicates 25th percentile of the WHO Growth Velocity Standards; solid black line indicates the 50th percentile. Colored lines indicate and vertical bars indicate 95% confidence intervals for *ki* cohorts. Light gray lines indicate cohort-specific linear growth velocity curves. (**b**) Within-child difference in length-for-age Z-score per month by age, sex, and region. Smaller partially transparent points indicate cohort-specific estimates. Results shown in all panels were derived from 32 *ki* cohorts in 14 countries that measured children at least quarterly (n = 52,640 children).

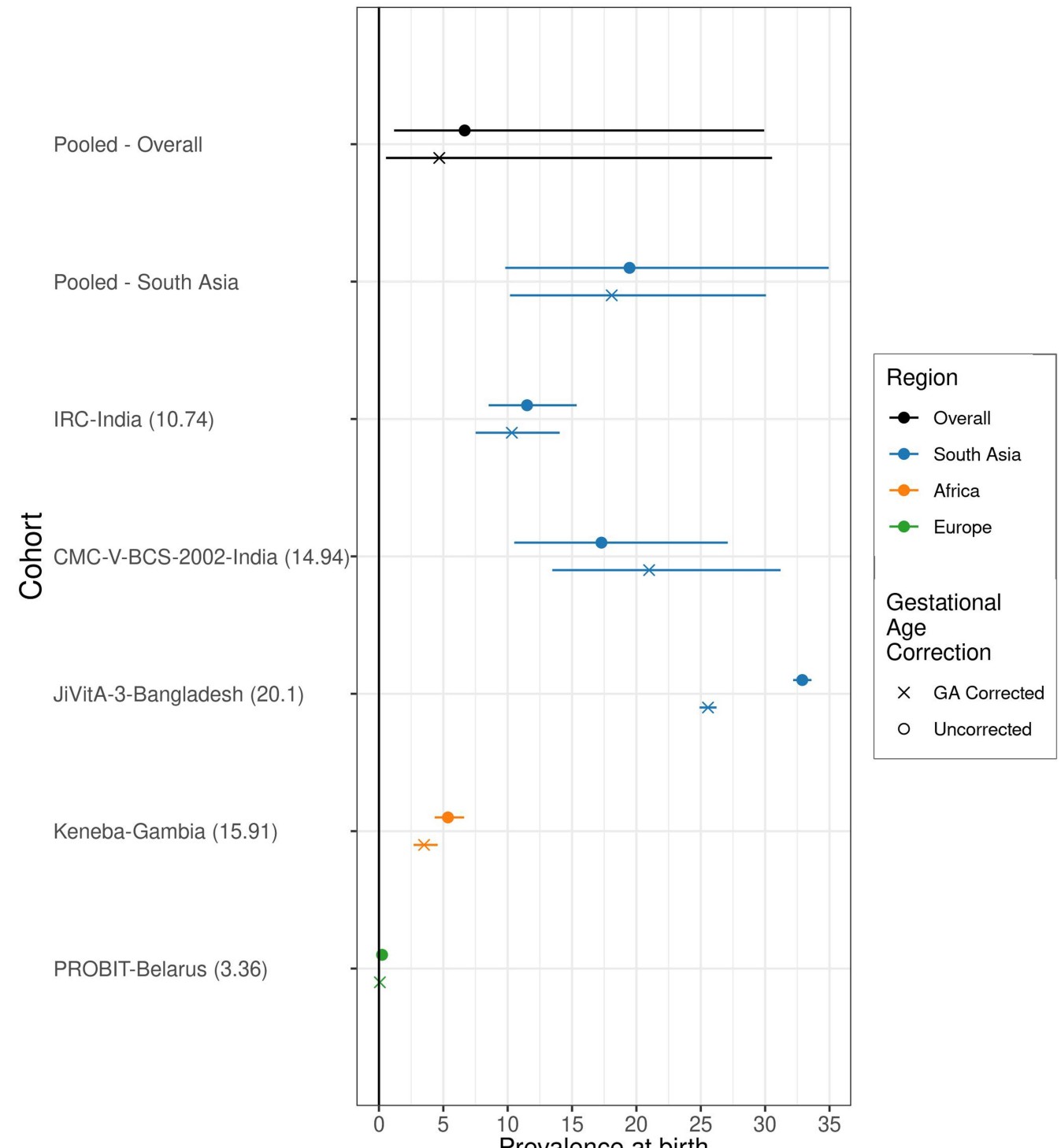

**Extended Data Fig. 15 | Comparison of stunting prevalence at birth with and without gestational age correction.** This figure includes the results from correcting at-birth Z-scores in the *ki* cohorts that measured gestational age (GA) for 37,218 measurements in 5 cohorts. The number in the parentheses following each cohort name indicates the prevalence of pre-term birth in each cohort. The corrections are using the Intergrowth standards and are implemented using the R *growthstandards* package (https://ki-tools.github.io/growthstandards/). Overall, the stunting prevalence at birth decreased slightly after correcting for gestational age, but the cohort-specific results are inconsistent. Observations with GA outside of the Intergrowth standards range (<168 or > 300 days) were dropped for both the corrected and uncorrected data. Prevalence increased after GA correction in some cohorts due to high rates of late-term births based on reported GA. Gestational age was estimated based on mother's recall of the last menstrual period in the Jivita-3, IRC, and CMC-V-BCS-2002 cohorts, was based on the Dubowitz method (newborn exam) in the Keneba cohort and was based on ultrasound measurements in the PROBIT trial.

**Extended Data Table 1 | Countries and survey years included
in the analysis of DHS data**

| Africa | |
|---|---|
| Gambia | 2013 |
| Malawi | 2015-2016 |
| Tanzania | 2015-2016 |
| South Africa | 2016 |
| Zimbabwe | 2015 |
| **South Asia** | |
| Bangladesh | 2014 |
| India | 2015-2016 |
| Nepal | 2016 |
| Pakistan | 2017-2018 |
| **Latin America** | |
| Brazil | 1996 |
| Guatemala | 2014-2015 |
| Peru | 2012 |

# nature research

# Reporting Summary

Nature Research wishes to improve the reproducibility of the work that we publish. This form provides structure for consistency and transparency in reporting. For further information on Nature Research policies, see Authors & Referees and the Editorial Policy Checklist.

## Statistics

For all statistical analyses, confirm that the following items are present in the figure legend, table legend, main text, or Methods section.

| n/a | Confirmed | |
|---|---|---|
| ☐ | ☒ | The exact sample size (*n*) for each experimental group/condition, given as a discrete number and unit of measurement |
| ☐ | ☒ | A statement on whether measurements were taken from distinct samples or whether the same sample was measured repeatedly |
| ☒ | ☐ | The statistical test(s) used AND whether they are one- or two-sided<br>*Only common tests should be described solely by name; describe more complex techniques in the Methods section.* |
| ☐ | ☒ | A description of all covariates tested |
| ☐ | ☒ | A description of any assumptions or corrections, such as tests of normality and adjustment for multiple comparisons |
| ☐ | ☒ | A full description of the statistical parameters including central tendency (e.g. means) or other basic estimates (e.g. regression coefficient) AND variation (e.g. standard deviation) or associated estimates of uncertainty (e.g. confidence intervals) |
| ☐ | ☒ | For null hypothesis testing, the test statistic (e.g. *F*, *t*, *r*) with confidence intervals, effect sizes, degrees of freedom and *P* value noted<br>*Give P values as exact values whenever suitable.* |
| ☒ | ☐ | For Bayesian analysis, information on the choice of priors and Markov chain Monte Carlo settings |
| ☐ | ☒ | For hierarchical and complex designs, identification of the appropriate level for tests and full reporting of outcomes |
| ☐ | ☒ | Estimates of effect sizes (e.g. Cohen's *d*, Pearson's *r*), indicating how they were calculated |

*Our web collection on statistics for biologists contains articles on many of the points above.*

## Software and code

Policy information about availability of computer code

| Data collection | This manuscript is a secondary data analysis of existing study data from 31 cohorts and trials, so we were not involved in original data collection. |
|---|---|
| Data analysis | All analyses were conducted using R statistical software, and scripts that reproduce all analyses are available on Github here: https://github.com/child-growth/ki-longitudinal-growth |

For manuscripts utilizing custom algorithms or software that are central to the research but not yet described in published literature, software must be made available to editors/reviewers. We strongly encourage code deposition in a community repository (e.g. GitHub). See the Nature Research guidelines for submitting code & software for further information.

## Data

Policy information about availability of data

All manuscripts must include a data availability statement. This statement should provide the following information, where applicable:
- Accession codes, unique identifiers, or web links for publicly available datasets
- A list of figures that have associated raw data
- A description of any restrictions on data availability

The data that support the findings of this study are available from the original study data contributors upon reasonable request (email contact: amertens@berkeley.edu).

# Field-specific reporting

Please select the one below that is the best fit for your research. If you are not sure, read the appropriate sections before making your selection.

☐ Life sciences　　☒ Behavioural & social sciences　　☐ Ecological, evolutionary & environmental sciences

# Behavioural & social sciences study design

All studies must disclose on these points even when the disclosure is negative.

| | |
|---|---|
| Study description | This study performed a quantitative analysis of de-identified secondary, longitudinal data on child growth. |
| Research sample | The data analyzed in this study were amassed as part of the Knowledge Integration (ki) initiative of the Bill & Melinda Gates Foundation, which aggregated observations on millions of participants from a global collection of studies on child birth, growth and development. We selected longitudinal cohorts from the database that met five inclusion criteria: 1) conducted in LMICs; 2) enrolled children between birth and age 24 months and measured their length and weight repeatedly over time; 3) did not restrict enrollment to acutely ill children; 4) enrolled at least 200 children; and 5) collected anthropometry measurements at least every 3 months. Thirty-one cohorts met inclusion criteria, including 52,640 children and 412,458 total measurements. |
| Sampling strategy | Not applicable. |
| Data collection | Not applicable. |
| Timing | Included datasets were collected between 1987 and 2017. |
| Data exclusions | We dropped 859 out of 413,317 measurements (0.2%) from the original datasets because length-for-age Z-scores were unrealistic (> 6 or < −6 Z). |
| Non-participation | Not applicable. |
| Randomization | Participants were not randomly assigned. |

# Reporting for specific materials, systems and methods

We require information from authors about some types of materials, experimental systems and methods used in many studies. Here, indicate whether each material, system or method listed is relevant to your study. If you are not sure if a list item applies to your research, read the appropriate section before selecting a response.

## Materials & experimental systems

| n/a | Involved in the study |
|---|---|
| ☒ | ☐ Antibodies |
| ☒ | ☐ Eukaryotic cell lines |
| ☒ | ☐ Palaeontology |
| ☒ | ☐ Animals and other organisms |
| ☒ | ☐ Human research participants |
| ☒ | ☐ Clinical data |

## Methods

| n/a | Involved in the study |
|---|---|
| ☒ | ☐ ChIP-seq |
| ☒ | ☐ Flow cytometry |
| ☒ | ☐ MRI-based neuroimaging |

# Reporting Summary

Nature Research wishes to improve the reproducibility of the work that we publish. This form provides structure for consistency and transparency in reporting. For further information on Nature Research policies, see Authors & Referees and the Editorial Policy Checklist.

## Statistics

For all statistical analyses, confirm that the following items are present in the figure legend, table legend, main text, or Methods section.

| n/a | Confirmed | |
|---|---|---|
| ☐ | ☒ | The exact sample size (*n*) for each experimental group/condition, given as a discrete number and unit of measurement |
| ☐ | ☒ | A statement on whether measurements were taken from distinct samples or whether the same sample was measured repeatedly |
| ☒ | ☐ | The statistical test(s) used AND whether they are one- or two-sided<br>*Only common tests should be described solely by name; describe more complex techniques in the Methods section.* |
| ☐ | ☒ | A description of all covariates tested |
| ☐ | ☒ | A description of any assumptions or corrections, such as tests of normality and adjustment for multiple comparisons |
| ☐ | ☒ | A full description of the statistical parameters including central tendency (e.g. means) or other basic estimates (e.g. regression coefficient) AND variation (e.g. standard deviation) or associated estimates of uncertainty (e.g. confidence intervals) |
| ☐ | ☒ | For null hypothesis testing, the test statistic (e.g. *F*, *t*, *r*) with confidence intervals, effect sizes, degrees of freedom and *P* value noted<br>*Give P values as exact values whenever suitable.* |
| ☒ | ☐ | For Bayesian analysis, information on the choice of priors and Markov chain Monte Carlo settings |
| ☐ | ☒ | For hierarchical and complex designs, identification of the appropriate level for tests and full reporting of outcomes |
| ☐ | ☒ | Estimates of effect sizes (e.g. Cohen's *d*, Pearson's *r*), indicating how they were calculated |

*Our web collection on statistics for biologists contains articles on many of the points above.*

## Software and code

Policy information about availability of computer code

| | |
|---|---|
| Data collection | This manuscript is a secondary data analysis of existing study data from 31 cohorts and trials, so we were not involved in original data collection. |
| Data analysis | All analyses were conducted using R statistical software, and scripts that reproduce all analyses are available on Github here: https://github.com/child-growth/ki-longitudinal-growth |

For manuscripts utilizing custom algorithms or software that are central to the research but not yet described in published literature, software must be made available to editors/reviewers. We strongly encourage code deposition in a community repository (e.g. GitHub). See the Nature Research guidelines for submitting code & software for further information.

## Data

Policy information about availability of data

All manuscripts must include a data availability statement. This statement should provide the following information, where applicable:
- Accession codes, unique identifiers, or web links for publicly available datasets
- A list of figures that have associated raw data
- A description of any restrictions on data availability

The data that support the findings of this study are available from the the original study data contributors upon reasonable request (email contact: amertens@berkeley.edu).

# Field-specific reporting

Please select the one below that is the best fit for your research. If you are not sure, read the appropriate sections before making your selection.

☐ Life sciences      ☒ Behavioural & social sciences      ☐ Ecological, evolutionary & environmental sciences

# Behavioural & social sciences study design

All studies must disclose on these points even when the disclosure is negative.

| | |
|---|---|
| Study description | This study performed a quantitative analysis of de-identified secondary, longitudinal data on child growth. |
| Research sample | The data analyzed in this study were amassed as part of the Knowledge Integration (ki) initiative of the Bill & Melinda Gates Foundation, which aggregated observations on millions of participants from a global collection of studies on child birth, growth and development. We selected longitudinal cohorts from the database that met five inclusion criteria: 1) conducted in LMICs; 2) enrolled children between birth and age 24 months and measured their length and weight repeatedly over time; 3) did not restrict enrollment to acutely ill children; 4) enrolled at least 200 children; and 5) collected anthropometry measurements at least every 3 months. Thirty-one cohorts met inclusion criteria, including 94,019 children and 645,869 total measurements, and an additional 4 studies including and 14,317 children and 70,659 additional measurements were used in an additional analysis of child mortality. |
| Sampling strategy | Not applicable. |
| Data collection | Not applicable. |
| Timing | Included datasets were collected between 1969 and 2014. |
| Data exclusions | We dropped 1,332 (0.2%) unrealistic measurements of length-for-age Z-scores (> 6 or < −6 Z), 1,493 (0.2%) measurements of weight-for-age Z-scores (> 6 or < −5 Z),  and 1,834 (0.3%) measurements of weight-for-length Z-scores (> 5 or < −5 Z), consistent with WHO recommendations. |
| Non-participation | Not applicable. |
| Randomization | Participants were not randomly assigned. |

# Reporting for specific materials, systems and methods

We require information from authors about some types of materials, experimental systems and methods used in many studies. Here, indicate whether each material, system or method listed is relevant to your study. If you are not sure if a list item applies to your research, read the appropriate section before selecting a response.

## Materials & experimental systems

| n/a | Involved in the study |
|---|---|
| ☒ | ☐ Antibodies |
| ☒ | ☐ Eukaryotic cell lines |
| ☒ | ☐ Palaeontology |
| ☒ | ☐ Animals and other organisms |
| ☒ | ☐ Human research participants |
| ☒ | ☐ Clinical data |

## Methods

| n/a | Involved in the study |
|---|---|
| ☒ | ☐ ChIP-seq |
| ☒ | ☐ Flow cytometry |
| ☒ | ☐ MRI-based neuroimaging |

