## [Peer Review File · Nature]

Manuscript Title: Early childhood linear growth faltering in low- and middle-income countries

Reviewer Comments & Author Rebuttals

Reviewer Reports on the Initial Version:

Referees' comments:

Referee #1 (Remarks to the Author):

This paper addresses a question that has been of interest to researchers in nutrition and health – specifically the age trajectory of growth and when it fails to meet the potential. As the paper points out, there has been large analyses of cross-sectional surveys; there are fewer longitudinal studies in low-income countries. A pooling of follow up studies can lever larger sample size to provide a more robust picture. I have put below comments related to how the cohorts are pooled – both in terms of how cohorts are grouped and methods used for pooling – that in my view fundamentally influence our interpretation and relevance of results.

Cross-sectional studies would be perfectly sufficient to reveal the age trajectory of growth if incidence and prevalence were in an equilibrium and not changing over time (or more broadly age-specific height/length patterns were not changing) because the synthetic cohort that can be created in a cross-section would look the same as a true cohort (repeated cross-sections can overcome this issue but that is outside the scope of this paper). But we know that growth patterns are changing – that is why we see substantial trends throughout the world in HAZ and stunting. This same issue also creates a dilemma for using longitudinal cohort – in methodology and interpretation.

The cohorts used in the paper span the period from late 1960s in Guatemala and 1980s in South/Southeast Asia and Africa to post-2010 in Brazil. The former would be some of the shortest children recorded and the latter close to WHO standard. It would be very likely that age patterns of growth and stunting incidence will vary in these groups (after all, for a group that is close to the reference population, there should be low incidence at any age; this may well be a reason why the paper did not use data from high-income countries where growth is well-monitored). From the perspective of interpreting the results, how would we use the results to inform contemporary policies if the patterns in each cohort are specific the place and time when the cohort was enrolled? Would Guatemala in 2020 look more like Guatemala in the 1960s or Brazil in the 2010s or ...? There is an analogous situation in child mortality: as under-five mortality goes down, a larger share is concentrated in infants, and subsequently in neonatal period calling for very different policies.

From a methodological perspective this real (vs. stochastic) heterogeneity in the quantity of interest creates a question for whether and how to pool cohorts. Cohort pooling is useful when the quantity of interest is believed to be etiological and largely transferable – strict epidemiologists would not even agree with random effects meta-analysis; less strict ones would consider it when the heterogeneity is believed to be a function of unobserved study design issues but not a real feature of quantity of interest. Here what is studied is fundamentally descriptive (unless there is a belief that

age is etiological which goes against the whole idea of intervention), which means regional pooling is not an appropriate way of dividing cohorts. If cohorts are pooled, their stratification should be based on specific criteria that leads to those pooled to be “similar” in the quantity of interest (would be it LAZ at birth or at another index age? etc).

As a specific issue, the methods section describes the statistical properties of random effects pooling but random effects on what parameters? Unlike traditional Cox models, where random effect is typically on baseline cohort-specific hazard, the outcome here is a series of age specific LAZs or stunting prevalences. Was it placed on LAZ/prevalence at age 0 or ...? How do these choices affect how each cohort influences the other quantities of interest?

The paper presents results for both LAZ and for stunting, which from Methods I understand was logit transformed (was the transformation on prevalence or was a logit link function used for individual participants?). The analysis of the continuous measure is conceptually straightforward. For stunting, modelling prevalence over age would be straightforward in a single cohort (I would have preferred a probit transformation which more closely represents the sort of distributions we see for physiological variables). When cohort are pooled however the challenge is to consider how different distributions influence the inferred parameters at each age because a shift in mean can lead to different changes in prevalence depending on starting mean and SD (if the mean is so low that the entire population is stunted, any shift will not change prevalence; similarly for a very high mean; and for a mean with the range of -2 and 0, the change depends on the value of mean, SD, skewness, etc). For the results to reflect the underlying distributional properties, in my view the analysis should start with a basic premise of how age acts to change stunting prevalence and consider how that should drive the model used for pooling: one possibility, not pursued in the paper, is that stunting acts by a combination of shift in population mean and perhaps higher moments; these can be modelled in a parametric distributional model (e.g. a skewed normal) or using a mixture model. The other possibility is to model (change in) prevalence directly, as done here. If this is done, the model should account for the fundamental distributional nature of LAZ that leads to the above-mentioned issue of dependence on baseline mean (for example, by modelling change in prevalence as a function of age, prevalence in the previous age and mean at previous age or some other way that takes into account the importance of the distribution). The former approach has the advantage of coherently generating mild, moderate, severe stunting prevalence which as citation 12 has shown are associated with different risks.

A result that in view of this reviewer was most interesting was the children who recovered from stunting – the lighter green sections of Figure 4b. I was eager to see what their characteristics were and why/how they improved but there was no information on that (nor in the 3rd paper in the series; recovery from wasting seemed to have been analyzed but not stunting)

Referee #2 (Remarks to the Author):

Summary of key results: This paper is an important contribution to the child growth literature given the paucity of available data and longitudinal growth data analyses. The paper provides insights into

the incidence of postnatal stunting, with much of growth failure occurring within the first 3 months of life. A majority of children whose attained LAZ improves either still experienced some growth deficits (LAZ<0) or become stunted again.

Given the age ranges of the children under study, it is hard to elucidate social and environmental factors that may have more influence on overall and sex-specific growth. As children age, they generally interact more with their environment and its socio-cultural norms which influence feeding and care practices and ultimately can influence nutrition well-being compared to the first months of life.

This paper's significance is its contribution to our understanding of population-level child growth in LMIC who have been followed over time. It allows for far more clarity in identifying age-specific patterns of faltering growth. More work such as this is warranted.

Data & methodology:

The overall methodology of this paper appears robust. There isn't a clear rationale for the characterization of "linear growth deficits" as LAZ< -1. Why not LAZ<0, or any other cut-off below the median between 0 and -1 for that matter?

Lines 375-376: For the sake of clarity, would include that you are comparing your findings to attained growth patterns, not growth velocity patterns.

Suggested improvements:

Lines 271-282: A striking finding not discussed by authors in this section nor mentioned in the discussion, revealed in Fig 3 is the trajectory of those who are classified as never stunted nor experiencing growth deficits $-1 < LAZ < 0$ continue to experience a downward trend in growth revealed by their average LAZ as they age. It makes the case that children in these study populations are not only on average born with a LAZ below the standard median but are experiencing decelerations and suboptimal growth regardless of their LAZ at birth. This includes those never identified as experiencing poor growth.

Lines 314-318: Velocity estimates showed that boys, compared to girls, displayed little variation in the patterns of growth, but there were differences in the rate of growth at every 3 month age interval except 3-6 and 6-9, 15-18 months of age. It would be helpful to know here based on Figure 5 whether the rate of growth was higher or lower in girls compared to boys at each age interval specific as being sig different because it appears, based on the figure, that girls' rate of growth was lower at some of these age timepoints. More clarity in the text on this would provide a more comprehensive understanding of sex-specific growth rate differentials.

The paragraph that begins on line 349 might benefit from the inclusion of the reference: Lassi ZS, Padhani ZA, Rabbani A, Rind F, Salam RA, Das JK, Bhutta ZA. Impact of Dietary Interventions during Pregnancy on Maternal, Neonatal, and Child Outcomes in Low- and Middle-Income Countries. *Nutrients*. 2020 Feb 19;12(2):531. DOI: 10.3390/nu12020531.

Lines 375-377: The consistency in patterns with national surveys alluded to in this section are attained growth patterns and not patterns of growth velocity. Thus, it is hard to assess the external validity of those patterns.

It is worth noting more explicitly that within child, changes in LAZ are minimal except early in life.

No legends noted for Extended Figure 4

In Figure 5, the footnote for 5a includes an additional comma that is not required.

One limitation of this study is that it captures growth dynamics only within the first 2 years of life. Thus it's difficult to make claims past the first 2 years where we note in other longitudinal growth literature. Examples of such literature include Toshiaki Aizawa, 2020. Trajectory of inequality of opportunity in child height growth: Evidence from the Young Lives study, Demographic Research, Max Planck Institute for Demographic Research, Rostock, Germany, vol. 42(7), pages 165-202. Georgiadis et al., 2017. Growth recovery and faltering through early adolescence in low- and middle-income countries: Determinants and implications for cognitive development. DOI: 10.1016/j.socscimed.2017.02.031. Epub 2017 Feb 22. Additionally, Gaussman et al. in 2019 also found growth deficits later in life which were differential based on stunting status during the first 5 years of life.

Referee #3 (Remarks to the Author):

1. This is an important topic – to investigate longitudinal growth trajectories of young children. I applaud the authors for tackling this issue, but like the other companion papers in this series, many of my concerns revolve around writing, language, and accurate representation of what is and is not known. A number of specific examples of this are below. The title should also be changed to be descriptive of what the paper is. I suggest perhaps “Linear growth failure dynamics: age patterns in incident and prevalent stunting in longitudinal cohort studies.”

2. Citations

a. Citation 6 (Martorell 2017) does NOT support the statement that “children in optimal environments have the same growth potential, regardless of their geographic location.” Recommend revising both the citation and the rather universal nature of the statement.

b. Also citation 7 – this whole sentence is a bit cart before the horse. The hypothesis of that optimal growth is different in different populations is one that is not supported by the available data, but this is not the same as saying that there is clear data that optimal child growth IS the same everywhere, which is what the authors imply here. There is such data on adult high BMI where minimum risk is consistently different for “Asian” vs. “non-Asian,” but no similar amount of information on child growth. The purpose of international standards is to have a common benchmark against which to judge, without any a priori value proposition on the meaning of that benchmark. Recommend revising the sentence.

c. Citation 47 – Wow ... this is a rather abrupt, anti-equity, and anti-woman evolutionary theory that is presented in this paper from 1973.

3. Introduction

a. Recommend either eliminating the first paragraph entirely or shortening to 2-3 sentences. It rambles, is somewhat repetitive, sometimes a little misleading (per above), and unnecessary to this paper. Maybe even start the paper with the sentence on line 179 and add a couple of the thoughts from the first paragraph to support.

b. Line 175 – specify that this is SDG 2.21. SDG 2 has eight targets and multiple indicators and SDG 2.2 has three indicators. This paper is not about all of SDG 2.2 or SDG 2.

c. Line 181-3: Authors state catch up growth is rare without radical improvements..., but one of the sources cited (Prentiss, et. al, #22) states the opposite, namely that there is substantial catch up growth in Gambia even without any specific interventions. Which is it?

d. Para 3 – the authors focus on incidence as the sole benefit of analyzing longitudinal vs. cross-sectional data (the discussion of this could be tightened up/ shortened), but neglect to mention other important considerations. This discussion should be the focus of the introduction.

1. First, what of the hypothesis that those that are the most marginalized/ have the worst growth failure are either a) underrepresented in surveys such as DHS/ MICS/ nutrition surveys or b) that there is potential survivor bias in cross-sectional analyses?

2. Second, what is the authors subsequent commentary – or opinion – on

♣ The fact that many surveys do not measure children < 6 months of age – would that omission, without adjustment, lead to likely bias of stunting estimates?

♣ the appropriateness of WNT 2025 / SDG targets being only for the aggregate of <5 year combined?

4. Line 222 – again, same question as for other papers. How did the number of children go from “millions” (line 214) to ~63k. Some of the criteria seem rather arbitrary – LMIC only, >200 children, quarterly growth measurements and likely to lead to very large exclusions. The authors explanations for these criteria appear ONLY in the caption for extended data figure 1. Also, I find that same figure misleading – the way in which the exclusion are presented implies, for example, that studies like intergrowth-21 and iterbio-21 enrolled <200 children, all after the age of 2, who were only acutely ill and measured less than quarterly. The grid presentation of study metadata is useful, but should be done accurately to represent the actual metadata of each of the considered studies, not just the criterion at which that study was dropped. Adding then a summary row of how many studies would be eliminated solely on the basis of each criterion would then be much more instructive.

5. I do not agree with the presented rationale for the criterion of >200 children enrolled. Stunting prevalence rates in many study locations are well in excess of 10%, in some cases exceeding 50%. These are not rare events. This appears to have dropped a substantial number of studies.

6. Line 647 – “Children whose first measurement occurred after birth were assumed to have experienced stunting onset at the age halfway between birth and the first measurement.” I had the same comment on this assumption from the other paper. This is a powerful assumption and, based on the fact that the authors found peak stunting in the first three months, has the potential to make their finding of peak incidence in 0-3 months merely an artifact of data processing. This needs to be carefully explored and explained.

7. Presentation of results (figure 2, 3) – similar to the other papers, I strongly contend that this analysis does not support the presentation of results as regional or global pooled results. What could/ should be the case is that each study has its data presented as individual observations and then the pooled effects could be presented (ie. a modified funnel plot for meta-regressions).

8. DHS comparisons –

a. Figure 2B. without seeing country, or subnational-specific comparisons, this comparison between KI and DHS is not meaningful. Pooling across locations is not valid either. Better would be to have this compare each of the 18 studies, matched to the specific subnational DHS data from the location in which the KI study is/ was performed. Need to also indicate the gap in # of years from the comparator DHS (the authors only mention that it was “the most recent”, but is that 1 year? 7 years? 13 years?) for each of the KI studies.

b. Extended data figure 3 – again, this method of pooling is not meaningful here. Show a) individual studies and b) actual descriptives of the studies, not splines of their pooled values.

9. Figure 3 – A big part of the story is missing here.

a. I recommend the authors add another panel that is “remission” – what is the age pattern of children getting better? Put another way, how specifically do you explain the gap between cumulative incidence and prevalence? Is it remission? Mortality? Drop outs (with or without assumed mortality)? Again, this needs to be done at the individual study level. This is sort of explored in figure 4, but would be better presented alongside the other metrics here.

b. Panel c (bottom) – would prefer this as a stacked to 100% column. It would also be better if the two images in figure 3c were both split into 3c/3d and also made tall/ side-by-side rather than wide/ short. They are very hard to read now.

10. Figure 4 –

a. Panel a: see comments above re: presentation of remission data. Fig 4a is complementary to that so can stay as it is another interesting way to look at the patterns.

b. Panel b and c: These data would be better presented differently – change the distributions to delta-LAZ between subsequent time periods rather than distribution of LAZ scores. To keep with the them from 4a, which is new vs. relapse vs. recovered stunting, it would be good to have those differentiated on the graphs. Maybe each “phenotype” can be a different color rather than having different colors for different cohort starting age groups.

c. figure 4c – going with the multiple comments previously, there is not value in presenting the global pooled mean here without also showing the data. Figure 4b shows data. If this sub-figure can be revised to also display data, that would be great (again suggest differentiating by phenotype). O/w if you change the plotted metric from 4b, then this can be dropped.

11. Growth velocity by age and sex

a. Figure 5 and extended data figure 4. Several comments

1. Need to have uncertainty displayed in panel 5A. Would this

2. EDFig 4 should include each study, not just pooled (per multiple previous comments). A pooled result between 15-25% doesn't say much. What is the distribution of individual studies/ locations?

b. Lines 322 – 324. The validity / correlates of worse growth in boys has not been explored in this paper (is partially explored in other papers), is not supported, and in this reviewer's opinion should either be revised / or deleted. Per above, this whole sentence is problematic, including the citations used to support it.

12. Discussion

a. Line 339 – need to also talk about the other factors in addition to these – e.g. pregnancy complications like HDoP, HTN, kidney issues, birth spacing/ family planning, education, women's autonomy.

b. This paragraph is a little mixed up - starts with listing risk factors for “postnatal” linear growth failure then all/ most of the examples are of items that would affect fetal / “intra”-natal growth predominantly. These need to be clearly differentiated here in how they are discussed. (Also need to

be differentiated from an analytical standpoint in one of the companion papers).

c. Paragraph starting line 348 – this is the whole point of the paper. This analysis reveals information on age trends in incidence, remission, and relapse and states as its purpose to then subsequently better inform intervention. If that is the case, that this work can help support improved evidence basis of interventions (which I believe it can), then this section of the discussion is the most important. What are the specific options that both ARE and ARE not included in WHO recommendations (vs. those of other nutrition and/or child health organizations)? How do they fit? How do they miss the mark? What do the results of this analysis tell us about *specific* blind spots in prevention and treatment with respect to age groups and growth dynamics? What are the specific knowledge/ data gaps that would be needed to be filled in order to make specific recommendations?

Author Rebuttals to Initial Comments:

Referees' comments:

Referee #1 (Remarks to the Author):

This paper addresses a question that has been of interest to researchers in nutrition and health – specifically the age trajectory of growth and when it fails to meet the potential. As the paper points out, there has been large analyses of cross-sectional surveys; there are fewer longitudinal studies in low-income countries. A pooling of follow up studies can lever larger sample size to provide a more robust picture. I have put below comments related to how the cohorts are pooled – both in terms of how cohorts are grouped and methods used for pooling – that in my view fundamentally influence our interpretation and relevance of results.

- 1. Cross-sectional studies would be perfectly sufficient to reveal the age trajectory of growth if incidence and prevalence were in an equilibrium and not changing over time(or more broadly age-specific height/length patterns were not changing) because the synthetic cohort that can be created in a cross-section would look the same as a true cohort (repeated cross-sections can overcome this issue but that is outside the scope of this paper). But we know that growth patterns are changing – that is why we see substantial trends throughout the world in HAZ and stunting. This same issue also creates a dilemma for using longitudinal cohort – in methodology and interpretation.**

Response: While we agree with the reviewer that both cross-sectional and longitudinal data are both potentially influenced by secular trends, we believe there are two particularly novel contributions to this paper that require longitudinal data: synthesis of 1) stunting reversal patterns and 2) growth velocity patterns from multiple cohorts around the world. We believe that our use of longitudinal data to investigate these outcomes is an important new contribution. With respect to secular trends, please see our response to Reviewer 1, Comment 2 below.

- 2. The cohorts used in the paper span the period from late 1960s in Guatemala and 1980s in South/Southeast Asia and Africa to post-2010 in Brazil. The former would be some of the shortest children recorded and the latter close to WHO standard. It would be very likely that age patterns of growth and stunting incidence will vary in these groups (after all, for a group that is close to the reference population, there should be low incidence at any age; this may well be a reason why the paper did not use data from high-income countries where growth is well-monitored). From the perspective of interpreting the results, how would we use the results to inform contemporary policies if the patterns in each cohort are specific the place and time when the cohort was enrolled? Would Guatemala in 2020 look more like Guatemala in the 1960s or Brazil in the 2010s or ...? There is an analogous situation in child mortality: as under-five mortality goes down, a larger share is concentrated in infants, and subsequently in neonatal period calling for very different policies.**

Response: We agree that observations from the 1960s reflect very different conditions from more recent ones. As such, we have revised the eligibility criterion to include studies that had a median year of birth in 1990 or later. This new criterion resulted in a slightly different set of studies, all of which span the period from approximately 1987 to 2017. This period is of policy relevance, because the Millennium Development Goals focused on the period from 1990 to 2015. The study from Guatemala in the 1960s is no longer included in our analyses, and a few smaller studies were added after we removed the criterion that excluded studies with $N < 200$ in response to Reviewer 3, Comment 5. We revised the text on lines 201-208 accordingly (relevant additions in bold):

*“We included cohorts from the database that met five inclusion criteria: 1) conducted in LMICs; 2) **had a median year of birth in 1990 or later** 3) enrolled children between birth and age 24 months and measured their length and weight repeatedly over time; 4) did not restrict enrollment to acutely ill children; and 5) collected anthropometry measurements at least every 3 months (Extended Data Fig 1). ... Thirty-two cohorts met inclusion criteria, including 53,210 children and 412,458 total **measurements from 1987 to 2017** (Fig 1, Extended Data Tables 1-2).”*

- 3. From a methodological perspective this real (vs. stochastic) heterogeneity in the quantity of interest creates a question for whether and how to pool cohorts. Cohort pooling is useful when the quantity of interest is believed to be etiological and largely transferable – strict epidemiologists would not even agree with random effects meta-analysis; less strict ones would consider it when the heterogeneity is believed to be a function of unobserved study design issues but not a real feature of quantity of interest. Here what is studied is fundamentally descriptive (unless there is a belief that age is etiological which goes against the whole idea of intervention), which means regional pooling is not an appropriate way of dividing cohorts. If cohorts are pooled, their stratification should be based on specific criteria that leads to those pooled to be “similar” in the quantity of interest (would be it LAZ at birth or at another index age? etc).**

Response: We agree with the Reviewer that pooling is appropriate when characteristics of cohorts are similar. We feel that restricting analyses to a narrower time period, as described in our response to Comment 2 above helps to increase the comparability of cohorts. In addition, we have revised nearly all of the figures in this study to include cohort-specific estimates when possible (see revised Figures 2, 3a, 4d, and 5 and Extended Data Figure 3 and added new Extended Data Figures 4-5). In addition, cohort-specific point estimates with confidence intervals are available in the online supplementary material: <https://child-growth.github.io/stunting/cohort.html>

Regarding the issue of regional pooling we accept the reviewer's comments but would add two rejoinders. First, the etiological drivers are assumed to be a complex mix of poor environmental conditions and poor diet (ultimately driven by poverty and low levels of health/infection awareness). These drivers do tend to cluster in regions and hence meet the 'similarity' constraint that you propose. Second, and most important, if our analyses are to assist in driving public health advances (primarily through use of the results in international, regional and national advocacy) then it is very important to provide contextual geographic relevance.

With regard to the reviewer's comment about random effects meta-analysis, we agree that strictly speaking, random effects analyses assume that studies are drawn from a hypothetical "population" of effects, whereas fixed effects make inferences conditional on each studies' estimates. In practice, random effects models produce more conservative pooled standard errors than fixed effects, and in our view, this is an advantage of using random effects as our primary analysis.

- 4. As a specific issue, the methods section describes the statistical properties of random effects pooling but random effects on what parameters? Unlike traditional Cox models, where random effect is typically on baseline cohort-specific hazard, the outcome here is a series of age specific LAZs or stunting prevalences. Was it placed on LAZ/prevalence at age 0 or ...? How do these choices affect how each cohort influences the other quantities of interest?**

Response: We revised the Materials and Methods section as follows in lines 161-166:

"We estimated each age-specific mean using a separate estimation and pooling step. We first estimated the mean in each cohort and then pooled age-specific means across cohorts, while allowing for a cohort-level random effect. This approach enabled us to include the most information possible for each age-specific mean, while accommodating slightly different measurement schedules across the cohorts. Each cohort's data only contributed to LAZ or stunting incidence estimates at the ages for which it contributed data."

- 5. The paper presents results for both LAZ and for stunting, which from Methods I understand was logit transformed (was the transformation on prevalence or was a logit link function used for individual participants?). The analysis of the continuous measure is conceptually straightforward. For stunting, modelling prevalence over age would be straightforward in a single cohort (I would have preferred a probit transformation which more closely represents the sort of distributions we see for physiological variables).**

Response: The logit link function was used for individual-level data (since prevalence aggregated across individuals would be a continuous measure). This is now described in the Methods in lines 204-207: *“For both types of outcomes, we pooled binary outcomes on the logit scale and then back-transformed estimates after pooling to constrain confidence intervals between 0 and 1.”*

In practice, there is little to no difference in the results using a probit vs. logit transformation. To illustrate this, in a single cohort, we fit models for stunting prevalence by age using both link functions. The deviance for each model was exactly the same (10782.63), so the fits from each model completely overlap in the plot below. We added the following sentences to the Materials and Methods section on lines 204-206:

“While the probit transformation more closely resembles common distributions for physiologic variables, in practice the logit transformation produces nearly identical estimates and is more convenient for estimation.”

6. When cohort are pooled however the challenge is to consider how different distributions influence the inferred parameters at each age because a shift in mean can lead to different changes in prevalence depending on starting mean and SD (if the mean is so low that the entire population is stunted, any shift will not change prevalence; similarly for a very high mean; and for a mean with the range of -2 and 0, the change depends on the value of mean, SD, skewness, etc). For the results to reflect the underlying distributional properties, in my view the analysis should start with a basic premise of how age acts to change stunting prevalence and consider how that should drive the model used for pooling: one possibility, not pursued in the paper, is that stunting acts by a combination of shift in population mean and perhaps higher moments; these can be modelled in a parametric distributional model (e.g. a skewed normal) or using a mixture model. The other possibility is to model (change in) prevalence directly, as done here. If this is done, the model should account for the fundamental distributional nature of LAZ that leads to the above-mentioned issue of dependence on baseline mean (for example, by modelling change in

prevalence as a function of age, prevalence in the previous age and mean at previous age or some other way that takes into account the importance of the distribution). The former approach has the advantage of coherently generating mild, moderate, severe stunting prevalence which as citation 12 has shown are associated with different risks.

Response: Thank you for this helpful comment. We agree with the reviewer that it is important to consider how the mean and the SD change among children of different ages in each region, since stunting is a continuous process. In response to this comment, we fit a parametric skewed normal model to the age-specific distributions of LAZ in each cohort. We revised the Methods (lines 154-157) as follows: *“To investigate how the mean, standard deviation, and skewness of LAZ distributions varied by age, we fit linear models with skew-elliptical error terms using maximum likelihood estimation. We fit models separately by cohort.”*

In the main text, we added the following on lines 231-233: *“In most cohorts, as children aged, LAZ distributions shifted downwards (Extended Data Fig 6); the standard deviation and skewness of LAZ distributions was similar across child ages (Extended Data Fig 7).”*

With regard to the reviewer’s comment about pooling, we estimated age-specific incidence and prevalence within each cohort prior to pooling. This approach helps to account for cohort-specific LAZ distribution at each age, which drives stunting prevalence (as the reviewer pointed out).

7. **A result that in view of this reviewer was most interesting was the children who recovered from stunting – the lighter green sections of Figure 4b. I was eager to see what their characteristics were and why/how they improved but there was no information on that (nor in the 3rd paper in the series; recovery from wasting seemed to have been analyzed but not stunting)**

Response: We agree with the reviewer that more clearly elucidating child characteristics associated with stunting reversal would be important and interesting. In the companion article (“Causes and consequences...”), we added a summary of this additional analysis in Extended Data Figure 2. We found that few measured characteristics were associated with stunting reversal — those that were strongly associated with an increased probability of reversal included: longer birth length; higher maternal height, weight, and BMI; exclusive breastfeeding in the first 6 months; and some measures of socioeconomic status (fewer people in the home, more rooms in the home, higher father’s education).

Referee #2 (Remarks to the Author):

Summary of key results: This paper is an important contribution to the child growth literature given the paucity of available data and longitudinal growth data analyses. The paper provides insights into the incidence of postnatal stunting, with much of growth failure occurring within the first 3 months of life. A majority of children whose attained LAZ improves either still experienced

some growth deficits (LAZ<0) or become stunted again.

1. **Given the age ranges of the children under study, it is hard to elucidate social and environmental factors that may have more influence on overall and sex-specific growth. As children age, they generally interact more with their environment and its socio-cultural norms which influence feeding and care practices and ultimately can influence nutrition well-being compared to the first months of life.**

Response: We agree that social and environmental risk factors for linear growth faltering vary by age. Our focus, as in the field of global nutrition, was on the first 2 years of life, which is described as the most vulnerable for growth faltering (Victora et al), recognizing that dramatic infant and young child feeding and caregiving transitions are occurring during the infancy and early childhood period. Risk factors associated with linear growth are the focus of the third paper in this series, which examined associations between such risk factors and growth faltering. There is a section in the third paper (Mertens et al. "Causes and consequences...") that focuses on age-specific effect modification of risk factor associations with child growth. That analysis found many household-, maternal-, and child characteristics have associations with postnatal child growth that vary by age. We added a reference to this manuscript on lines 200-202:

"Companion articles report results for child wasting (weight-for-length Z-score < 2 standard deviations below the reference median)²⁷ and household, maternal, and child-level risk factors associated with linear growth faltering.²⁸"

This paper's significance is its contribution to our understanding of population-level child growth in LMIC who have been followed over time. It allows for far more clarity in identifying age-specific patterns of faltering growth. More work such as this is warranted.

Data & methodology:

2. **The overall methodology of this paper appears robust. There isn't a clear rationale for the characterization of "linear growth deficits" as LAZ< -1. Why not LAZ<0, or any other cut-off below the median between 0 and -1 for that matter?**

Response: Since LAZ <-1 is not a standard cutoff for linear growth deficits, we revised this sentence in lines 248-251 as follows: *"21% of all children were born with mean LAZ < -1, and their mean LAZ stabilized around -2 from age 1 month onward, with differences at birth in this group narrowing over time, likely via regression to the mean. 14% of children were born with mean LAZ between -1 and 0 at birth, and in these children, mean LAZ approached -2 at subsequent ages."*

- 3. Lines 375-376: For the sake of clarity, would include that you are comparing your findings to attained growth patterns, not growth velocity patterns.**

Response: We revised this sentence in lines 346-348 of the Discussion as follows: *“The consistency between attained linear growth patterns in this and nationally representative DHS surveys (Fig 2) suggests that overall, our results have reasonably good external validity.”*

Lines 271-282: A striking finding not discussed by authors in this section nor mentioned in the discussion, revealed in Fig 3 is the trajectory of those who are classified as never stunted nor experiencing growth deficits $-1 < LAZ < 0$ continue to experience a downward trend in growth revealed by their average LAZ as they age. It makes the case that children in these study populations are not only on average born with a LAZ below the standard median but are experiencing decelerations and suboptimal growth regardless of their LAZ at birth. This includes those never identified as experiencing poor growth.

Response: We agree that this is an important finding and have revised the text on lines 251-254 as follows: *“The remaining 65% never met the criteria for stunting; yet, among these children, mean LAZ was between -0.5 and 0 at birth and declined steadily, reaching close to -1 by age 15 months.”*

In addition, we have revised the first paragraph of the Discussion section in lines 304-307 as follows: *“Even among children who never met the criteria for stunting before age 2 years, mean LAZ steadily declined with age (Fig 3b). Our findings reinforce that linear growth faltering among children in LMICs is a whole-population phenomenon, with both stunted and not-stunted children experiencing suboptimal growth trajectories in early life.²¹”*

- 4. Lines 314-318: Velocity estimates showed that boys, compared to girls, displayed little variation in the patterns of growth, but there were differences in the rate of growth at every 3 month age interval except 3-6 and 6-9, 15-18 months of age. It would be helpful to know here based on Figure 5 whether the rate of growth was higher or lower in girls compared to boys at each age interval specific as being sig different because it appears, based on the figure, that girls' rate of growth was lower at some of these age timepoints. More clarity in the text on this would provide a more comprehensive understanding of sex-specific growth rate differentials.**

Response: In the revision, we removed formal p-value comparisons between sexes. Although absolute growth velocity (in cm per year) for males was consistently higher than for females across many ages, when normalizing linear growth to sex-specific LAZ scores, the differences between boys and girls were not as large and were not statistically significantly different, as evidenced by 95% confidence intervals that clearly include the other sex's mean in Figure 5b.

5. **The paragraph that begins on line 349 might benefit from the inclusion of the reference: Lassi ZS, Padhani ZA, Rabbani A, Rind F, Salam RA, Das JK, Bhutta ZA. Impact of Dietary Interventions during Pregnancy on Maternal, Neonatal, and Child Outcomes in Low- and Middle-Income Countries. *Nutrients*. 2020 Feb 19;12(2):531. DOI: 10.3390/nu12020531.**

Response: We have added this citation to lines 309-313 in the Discussion section (citation number 34)

“Our findings that 13% of children were stunted at birth and that the birth prevalence was 20% in South Asia emphasize the importance of pre-pregnancy and prenatal interventions to reduce stunting, especially in South Asia. These interventions include maternal micronutrient and macronutrient supplementation,^{33,34} increasing women’s autonomy and education,³⁵ reducing adolescent pregnancies in LMICs by delaying the age of marriage and first pregnancy,³⁶ and promoting family planning.^{37”}

6. **Lines 375-377: The consistency in patterns with national surveys alluded to in this section are attained growth patterns and not patterns of growth velocity. Thus, it is hard to assess the external validity of those patterns.**

Response: We agree and have revised the sentence in lines 346-349 as follows:
“The consistency between attained linear growth patterns in this and nationally representative DHS surveys (Fig 2) suggests that overall, our results have good external validity. For growth velocity, the cohorts represented populations close to the 25th percentile of international standards (Figure 5a).”

7. **It is worth nothing more explicitly that within child, changes in LAZ are minimal except early in life.**

Response: We have added the following sentence in line 286: *“After 6 months, within-child changes in length and LAZ were minimal.”*

8. **No legends noted for Extended Figure 4**

Response: We have added a legend to this figure, which is now Extended Data Figure 9.

9. **In Figure 5, the footnote for 5a includes an additional comma that is not required.**

Response: We have fixed this.

10. **One limitation of this study is that it captures growth dynamics only within the first 2 years of life. Thus it's difficult to make claims past the first 2 years where**

we note in other longitudinal growth literature. Examples of such literature include Toshiaki Aizawa, 2020. Trajectory of inequality of opportunity in child height growth: Evidence from the Young Lives study, Demographic Research, Max Planck Institute for Demographic Research, Rostock, Germany, vol. 42(7), pages 165-202. Georgiadis et al., 2017. Growth recovery and faltering through early adolescence in low- and middle-income countries: Determinants and implications for cognitive development. DOI: 10.1016/j.socscimed.2017.02.031. Epub 2017 Feb 22. Additionally, Gaussman et al. in 2019 also found growth deficits later in life which were differential based on stunting status during the first 5 years of life.

Response: We added the following sentence to the limitations paragraph in the Discussion section: “*Finally, our inferences are limited to the first two years of life, since very few included studies measured children at older ages. Other studies, however, have found that stunting status in early life is associated with health outcomes later in life, and the timing and extent of early life linear growth faltering is associated with the magnitude of later catch-up growth.*”^{6-8,17,18,20}

6. de Onis, M. & Branca, F. Childhood stunting: a global perspective. *Matern. Child. Nutr.* **12**, 12–26 (2016).

7. Prendergast, A. J. & Humphrey, J. H. The stunting syndrome in developing countries. *Paediatr. Int. Child Health* **34**, 250–265 (2014).

8. Adair, L. S. *et al.* Associations of linear growth and relative weight gain during early life with adult health and human capital in countries of low and middle income: findings from five birth cohort studies. *Lancet* **382**, 525–534 (2013).

17. Aizawa, T. Trajectory of inequality of opportunity in child height growth: Evidence from the Young Lives study. *Demogr. Res.* **42**, 165–202 (2020).

18. Georgiadis, A. *et al.* Growth recovery and faltering through early adolescence in low- and middle-income countries: Determinants and implications for cognitive development. *Soc. Sci. Med.* **179**, 81–90 (2017).

20. Gausman, J., Kim, R. & Subramanian, S. V. Stunting trajectories from post-infancy to adolescence in Ethiopia, India, Peru, and Vietnam. *Matern. Child. Nutr.* **15**, (2019).

Referee #3 (Remarks to the Author):

This is an important topic – to investigate longitudinal growth trajectories of young children. I applaud the authors for tackling this issue, but like the other companion papers in this series, many of my concerns revolve around writing, language, and accurate representation of what is and is not known. A number of specific examples of this are below.

1. **The title should also be changed to be descriptive of what the paper is. I suggest perhaps “Linear growth failure dynamics: age patterns in incident and prevalent stunting in longitudinal cohort studies.”**

Response: While the suggested title would be more descriptive, the original title it conforms with Nature’s requirement that the title be ≤ 75 characters.

2. Citations

- a. **Citation 6 (Martorell 2017) does NOT support the statement that “children in optimal environments have the same growth potential, regardless of their geographic location.” Recommend revising both the citation and the rather universal nature of the statement.**

Response: Based on Comment 3a below, we have removed this sentence.

- b. **Also citation 7 – this whole sentence is a bit cart before the horse. The hypothesis of that optimal growth is different in different populations is one that is not supported by the available data, but this is not the same as saying that there is clear data that optimal child growth IS the same everywhere, which is what the authors imply here. There is such data on adult high BMI where minimum risk is consistently different for “Asian” vs. “non-Asian,” but no similar amount of information on child growth. The purpose of international standards is to have a common benchmark against which to judge, without any a priori value proposition on the meaning of that benchmark. Recommend revising the sentence.**

Response: Based on Comment 3a below, we have removed this sentence.

- c. **Citation 47 – Wow ... this is a rather abrupt, anti-equity, and anti-woman evolutionary theory that is presented in this paper from 1973.**

Response: We have removed this clause and citation.

3. Introduction

- a. **Recommend either eliminating the first paragraph entirely or shortening to 2-3 sentences. It rambles, is somewhat repetitive, sometimes a little misleading (per above), and unnecessary to this paper. Maybe even start the paper with the sentence on line 179 and add a couple fo the thoughts from the first paragraph to support.**

Response: We have shortened the first paragraph substantially. We feel that since *Nature* targets a general scientific audience, readers would benefit from

a few initial sentences describing the frequency of stunting, its associations with health and other outcomes, and global goals for reducing stunting. It now reads as follows:

“In 2018, 149 million children under 5 years (22% globally) were stunted (length-for-age Z-score >2 standard deviations below the median of the growth standard for age and sex), with the largest burden in South Asia and Africa.^{1,2} Early-life stunting is associated with increased risk of mortality,³ diarrhea, pneumonia, and measles in childhood^{4,5} and impaired cognition and productivity in adulthood.^{6–8} Global income would increase by an estimated \$176.8 billion per year if linear growth faltering could be eliminated.⁹ The WHO 2025 Global Nutrition Targets¹⁰ and Sustainable Development Goal 2.2.1 propose to reduce stunting prevalence among children under 5 years from 2012 levels by 40% by 2025,¹¹ reflecting renewed attention to child growth as a key indicator of a population’s progression toward improved health and human capital.^{12”}

- b. Line 175 – specify that this is SDG 2.21. SDG 2 has eight targets and multiple indicators and SDG 2.2 has three indicators. This paper is not about all of SDG 2.2 or SDG 2.**

Response: Changed as suggested.

- c. Line 181-3: Authors state catch up growth is rare without radical improvements..., but one of the sources cited (Prentiss, et. all, #22) states the opposite, namely that there is substantial catch up growth in Gambia even without any specific interventions. Which is it?**

Response: Thank you for pointing this out. We have revised the language and citations in lines 174-177 of the Introduction. The text now states:

“Children who experience linear growth faltering prior to prior to age 2 years can experience catch-up growth at older ages, particularly with improvements to their nutrition, health, and environment.^{15–19} However, the extent of catch-up growth is associated with the timing and extent of early life linear growth faltering.^{20”}

15. Dewey, K. G. & Begum, K. Long-term consequences of stunting in early life. *Matern. Child. Nutr.* 7, 5–18 (2011).

16. Leroy, J. L., Ruel, M., Habicht, J.-P. & Frongillo, E. A. Using height-for-age differences (HAD) instead of height-for-age z-scores (HAZ) for the meaningful measurement of population-level catch-up in linear growth in children less than 5 years of age. *BMC Pediatr.* 15, 145 (2015).
17. Aizawa, T. Trajectory of inequality of opportunity in child height growth: Evidence from the Young Lives study. *Demogr. Res.* 42, 165–202 (2020).
18. Georgiadis, A. et al. Growth recovery and faltering through early adolescence in low- and middle-income countries: Determinants and implications for cognitive development. *Soc. Sci. Med.* 179, 81–90 (2017).
19. Prentice, A. M. et al. Critical windows for nutritional interventions against stunting. *Am. J. Clin. Nutr.* 97, 911–918 (2013).
20. Gausman, J., Kim, R. & Subramanian, S. V. Stunting trajectories from post-infancy to adolescence in Ethiopia, India, Peru, and Vietnam. *Matern. Child. Nutr.* 15, (2019).

- d. **Para 3 – the authors focus on incidence as the sole benefit of analyzing longitudinal vs. cross-sectional data (the discussion of this could be tightened up/ shortened), but neglect to mention other important considerations. This discussion should be the focus of the introduction.**
1. **First, what of the hypothesis that those that are the most marginalized/ have the worst growth failure are either a) underrepresented in surveys such as DHS/ MICS/ nutrition surveys or b) that there is potential survivor bias in cross-sectional analyses?**

Response: Thank you for this helpful suggestion. We have revised this section of Paragraph 3 of the Introduction (lines 180-182) as follows:

“Analyses of cross-sectional studies cannot identify longitudinal patterns of linear growth faltering or reversal. Further, they may be subject to survivor bias and fail to include those children most vulnerable to undernutrition.”

- e. **2. Second, what is the authors subsequent commentary – or opinion – on**
- The fact that many surveys do not measure children < 6 months of age – would that omission, without adjustment, lead to likely bias of stunting estimates?**
 - the appropriateness of WNT 2025 / SDG targets being only for the aggregate of <5 year combined?**

Response: Thank you for this helpful suggestion. We have added the following sentences to the Discussion section in lines 358-361:

“Current WHO 2025 Global Nutrition Targets and Sustainable Development Goal 2.2.1 aim to reduce stunting prevalence among children under 5 years. Our findings suggest that defining stunting targets at earlier ages (e.g., stunting by 3 or 6 months) would help focus attention on the period when interventions may be most impactful.”

4. **Line 222 – again, same question as for other papers. How did the number of children go from “millions” (line 214) to ~63k. Some of the criteria seem rather arbitrary – LMIC only, >200 children, quarterly growth measurements and likely to lead to very large exclusions. The authors explanations for these criteria appear ONLY in the caption for extended data figure 1. Also, I find that same figure misleading – the way in which the exclusion are presented implies, for example, that studies like intergrowth-21 and iterbio-21 enrolled <200 children, all after the age of 2, who were only acutely ill and measured less than quarterly. The grid presentation of study metadata is useful, but should be done accurately to represent the actual metadata of each of the considered studies, not just the criterion at which that study was dropped. Adding then a summary row of how many studies would be eliminated solely on the basis of each criterion would then be much more instructive.**

Response: The original sentence referring to millions of children included cross-sectional datasets and data from high-income and historic cohorts. We have revised this sentence in lines 199-201 of the main text as follows: *“These data were aggregated by the Bill & Melinda Gates Foundation Knowledge Integration (ki) initiative and comprise approximately 100 longitudinal studies on child birth, growth and development.”*

Previously, Extended Data Figure 1 showed the exclusion criteria applied to cohorts in a progressive way, after each individual criterion was applied in succession. We have revised Extended Data Figure 1 to indicate all exclusion criteria that applied to each cohort. Each colored cell in the figure indicates a criterion that was met. For studies that met all inclusion criteria, all cells in their row are colored. The bars at the top of the plot now show the number of observations in each study that met each inclusion criterion by region. This now shows that the ki database includes approximately 880,000 observations (N=188,445 people) in all longitudinal cohorts, comprising approximately 200 total studies, of which 98 are longitudinal cohorts.

As mentioned in our response below to Reviewer 3, Comment 5, we have removed the exclusion criterion for <200 children enrolled. We have shortened the text of the Extended Data Figure 1 caption and added the following details on eligibility criteria to the Materials and Methods section on lines 7-22:

“(1) Studies that were conducted in low- or middle-income countries. Children in these countries have the largest burden linear growth faltering and are the key target population for preventive interventions.

(2) Studies that had a median year of birth in 1990 or later. This restriction resulted in a set of studies spanning the period from 1987 to 2017 and excluded older studies that are less applicable to current policy dialogues.

(3) Studies that enrolled children between birth and age 24 months and measured their length and weight repeatedly over time. We were principally interested in growth faltering during the first 1,000 days (including gestation), thought to be the key window for linear growth faltering.

(4) Studies that did not restrict enrollment to acutely ill children. Our focus on descriptive analyses led us to target, to the extent possible, the general population. We thus excluded some studies that exclusively enrolled acutely ill children, such as children who presented to hospital with acute diarrhea or who were severely malnourished.

(5) Studies that collected anthropometry measurements at least every 3 months to ensure that we adequately captured incident episodes and recovery.”

5. **I do not agree with the presented rationale for the criterion of >200 children enrolled. Stunting prevalence rates in many study locations are well in excess of 10%, in some cases exceeding 50%. These are not rare events. This appears to have dropped a substantial number of studies.**

Response: We have removed this criterion and updated Extended Data Figure 1 (see response to Reviewer 3, comment 4 above). This change resulted in 5 cohorts being added to the study (the “Transmission Dynamics of Cryptosporidial Infections” study in India and 4 new Child Malnutrition and Infection Network cohorts). The bars at the top of this figure now show the number of observations in each study that met each inclusion criterion. The criterion that led to the largest number of studies being excluded related to the frequency of measurements, not the study size. We have revised the text in lines 203-210 accordingly:

“We included cohorts from the database that met five inclusion criteria: 1) conducted in LMICs; 2) had a median year of birth in 1990 or later; 3) enrolled children between birth and age 24 months and measured their length and weight repeatedly over time; 4) did not restrict enrollment to acutely ill children; and 5) collected anthropometry

measurements at least every 3 months (Extended Data Fig 1). These criteria ensured we could rigorously evaluate the timing and onset of stunting among children who were broadly representative of general populations in LMICs. Thirty-two cohorts met inclusion criteria, including 53,210 children and 412,458 total measurements from 1987 to 2017 (Fig 1, Extended Data Table 1)."

- 6. Line 647 – “Children whose first measurement occurred after birth were assumed to have experienced stunting onset at the age halfway between birth and the first measurement.” I had the same comment on this assumption from the other paper. This is a powerful assumption and, based on the fact that the authors found peak stunting in the first three months, has the potential to make their finding of peak incidence in 0-3 months merely an artifact of data processing. This needs to be carefully explored and explained.**

Response: We agree that this assumption is a possible limitation, particularly for Figure 3a, which estimated age-specific incidence rates including 4 of 32 cohorts that did not include measurements at birth. To further assess the validity of the assumption, we enhanced Figure 4 of the manuscript, which included analyses among 15 monthly-measured cohorts from birth through 15 months. In that analysis, summarized in the new Figure 4b, we show that incidence was substantially higher in the first 3 months of life and remained relatively constant thereafter. This is evident not only in the pooled estimates (black line) but also in the cohort-specific estimates (pink lines). Elevated incidence at 0-3 months among children who were not born stunted (Figure 4b) increases our confidence that the high incidence in the 0-3 month window when estimated from all cohorts in the analysis (Figure 3a) was not a result of this data processing assumption in the 4 cohorts that did not include at-birth measurements.

- 7. Presentation of results (figure 2, 3) – similar to the other papers, I strongly contend that this analysis does not support the presentation of results as regional or global pooled results. What could/ should be the case is that each study has its data presented as individual observations and then the pooled effects could be presented (ie. a modified funnel plot for meta-regressions).**

Response: We agree that cohort-specific estimates are valuable. In response to this comment, as well as Reviewer 1's Comment #3, we significantly enhanced the figures throughout the manuscript to show cohort-specific estimates whenever possible. Specifically, we have added cohort-specific estimates to Figures 2, 3a, 4b, 4d, 5a, 5b, and Extended Data Figures 3-7 and 9. Including cohort-specific means provides readers with a comprehensive view of between-study heterogeneity alongside the pooled estimates. For full transparency and completeness, cohort-specific means with confidence intervals are now also available in the online supplementary material: <https://child-growth.github.io/stunting/cohort.html>.

8. DHS comparisons –

- a. **Figure 2B. without seeing country, or subnational-specific comparisons, this comparison between KI and DHS is not meaningful. Pooling across locations is not valid either. Better would be to have this compare each of the 18 studies, matched to the specific subnational DHS data from the location in which the KI study is/ was performed. Need to also indicate the gap in # of years from the comparator DHS (the authors only mention that it was “the most recent”, but is that 1 year? 7 years? 13 years?) for each of the KI studies.**

Response: We have revised Figure 2A and 2B to show ki cohort-specific estimates matched to DHS country-specific LAZ distributions and mean LAZ by age. The ki cohort from Guinea-Bissau did not have a corresponding DHS survey and is shown without that external reference. We feel this response to the reviewer’s comment has significantly enhanced the display of the ki cohort distributions vis-a-vis DHS surveys, and we have made similar improvements to figures in the companion paper focused on wasting. Overall, the revised figure still shows that LAZ distributions and age-specific means are broadly comparable between ki cohorts and DHS, with ki cohorts from South Asia generally falling below DHS estimates.

- b. **Extended data figure 3 – again, this method of pooling is not meaningful here. Show a) individual studies and b) actual descriptives of the studies, not splines of their pooled values.**

Response: We have revised this figure, now ED Figure 5, to show results for individual cohorts within each country. The shaded area indicates the area between the 5th and 95th percentile of LAZ at each age in each cohort. The line indicates the median LAZ by age in each cohort.

9. Figure 3 – A big part of the story is missing here.

- a. **I recommend the authors add another panel that is “remission” – what is the age pattern of children getting better? Put another way, how specifically do you explain the gap between cumulative incidence and prevalence? Is it remission? Mortality? Drop outs (with or without assumed mortality)? Again, this needs to be done at the individual study level. This is sort of explored in figure 4, but would be better presented alongside the other metrics here.**

Response: We agree that it is important to consider stunting reversals when interpreting incidence results. We felt that including additional panels with reversal data would complicate Figure 3a, which includes both monthly and quarterly cohorts. Stunting reversal is difficult to show clearly with quarterly cohorts, because fluctuations above the -2 cutoff within a 3-month quarter

would be missed. Instead, we have added a new panel to Figure 4, Figure 4b, which presents both cohort-specific and pooled incidence estimates of stunting onset, reversal, and relapse by age in monthly cohorts. This new view of the data shows that stunting reversal was rare at all ages but was slightly more likely before age 6-7 months than at older ages. The incidence of new stunting and stunting relapse exceeds reversal and hence explains the steady increase in stunting prevalence (shown in Figure 4a).

In the revision, we have added information to better display participant follow-up and drop-out over time (Extended Data Figures 3-4 show the percentage of enrolled children measured at each age in each cohort). Overall, the percentage of children measured at each age exceeded 80% at each age in most cohorts. Mortality data were not consistently tracked by age in the included cohorts, so it was not possible to create a similar figure showing the number of deaths by cohort and by age. Overall, however, we note that in cohorts that carefully measured and reported mortality, only a small percentage of children died during follow-up.

Study ID	Country	Under 2 years mortality rate	Number of deaths under 2 years
Burkina Faso Zn	BURKINA FASO	0.54	39
EE	PAKISTAN	1.05	4
GMS-Nepal	NEPAL	1.15	8
iLiNS-DOSE	MALAWI	2.74	53
iLiNS-DYAD-M	MALAWI	4.37	54
JiVitA-3	BANGLADESH	3.41	934
JiVitA-4	BANGLADESH	0.90	49
Keneba	GAMBIA	2.22	65
MAL-ED	BANGLADESH	1.13	3
MAL-ED	INDIA	0.80	2

MAL-ED	PERU	0.66	2
MAL-ED	SOUTH AFRICA	0.32	1
MAL-ED	TANZANIA	1.15	3
PROVIDE	BANGLADESH	0.57	4
SAS-CompFeed	INDIA	3.39	52
SAS-FoodSuppl	INDIA	1.44	6
VITAMIN-A	INDIA	2.70	108
ZVITAMBO	ZIMBABWE	7.89	1113

- b. Panel c (bottom) – would prefer this as a stacked to 100% column. It would also be better if the two images in figure 3c were both split into 3c/3d and also made tall/ side-by-side rather than wide/ short. They are very hard to read now.**

Response: We experimented with the suggested layout but decided to remove the bottom panel, which showed a stacked bar plot of the number of observations by age. This allowed us to leave sufficient space for panel B to ensure it is easily readable. We feel that the information previously conveyed by the counts of observations is conveyed in the width of the confidence intervals around each line (i.e., wider confidence intervals reflect smaller sample sizes).

10. Figure 4 –

- a. Panel a: see comments above re: presentation of remission data. Fig 4a is complementary to that so can stay as it is another interesting way to look at the patterns.**

Response: Please see our response to Comment 9a. In addition, we have now added Figure 4b, which we believe clarifies some of the time/age trends in stunting relapse.

- b. Panel b and c: These data would be better presented differently – change the distributions to delta-LAZ between subsequent time periods rather than distribution of LAZ scores. To keep with the them from 4a, which is new vs. relapse vs. recovered stunting, it would be good to have those differentiated on the graphs. Maybe each “phenotype” can be a different color rather than having different colors for different cohort starting age groups.

Response: Following the reviewer’s suggestion, we have created a version of Figure 4b (now numbered 4c in the revision) that shows the “phenotypes” in the same color palette as Figures 4a. This new figure is now included in the revision as Extended Data Figure 8. We considered replacing the main text Figure panel with the new color palette, but ultimately we felt that the original color scheme allowed for a more direct and important link between panels 4c and 4d. The phenotype color palette could not be used with Figure 4d without substantially redesigning the figure and increasing its space on the page.

We also considered changing the distributions from LAZ to delta-LAZ, as the reviewer suggested (see below), but we felt that the resulting figure was more difficult to interpret. Moreover, it did not clearly show the important trend evident in the original Figure: namely, that among children who reversed their stunting status at a given age, the LAZ distribution shifted downwards at subsequent ages, with many children crossing back below the “stunting” cutoff at -2 z. This trend is important, because it illustrates how classifying a child as having reversed their stunting status more likely reflects inherent variation in the continuous process around the -2 z cutoff, and that among these children, the mean of the LAZ distribution shifts downwards over time and re-centers around the -2 z cutoff.

- c. **figure 4c – going with the multiple comments previously, there is not value in presenting the global pooled mean here without also showing the data. Figure 4b shows data. If this sub-figure can be revised to also display data, that would be great (again suggest differentiating by phenotype). O/w if you change the plotted metric from 4b, then this can be dropped.**

Response: We have revised Figure 4d (previously 4c) to show individual cohort estimates, and we agree with the reviewer that this addition strengthens our inference both in this analysis and throughout the manuscript.

11. Growth velocity by age and sex

- a. **Figure 5 and extended data figure 4. Several comments**
 - 1. **Need to have uncertainty displayed in panel 5A. Would this**
 - 2. **EDFig 4 should include each study, not just pooled (per multiple previous comments). A pooled result between 15-25% doesn't say much. What is the distribution of individual studies/ locations?**

Response: Figure 5a previously included confidence intervals, but they were hard to see, so we have revised the figure design to make the CIs more visible with wider whiskers. In addition, we have added the cohort-specific estimates to each panel in Figure 5, as well as ED Figure 7 (previously ED Fig 4).

- b. **Lines 322 – 324. The validity / correlates of worse growth in boys has not been explored in this paper (is partially explored in other papers), is not supported, and in this reviewer's opinion should either be revised / or deleted. Per above, this whole sentence is problematic, including the citations used to support it.**

Response: We have removed this sentence from the manuscript.

12. Discussion

- a. **Line 339 – need to also talk about the other factors in addition to these – e.g. pregnancy complications like HDoP, HTN, kidney issues, birth spacing/ family planning, education, women's autonomy.**

Response: Thank you for this helpful suggestion. However, in responding to other reviewers' comments and synthesizing the text to stay within *Nature's* 2,500 word limit, we chose to remove these sentences from the Discussion section.

- b. **This paragraph is a little mixed up - starts with listing risk factors for “postnatal” linear growth failure then all/ most of the examples are of items that would affect fetal / “intra”-natal growth predominantly. These need to be clearly differentiated here in how they are discussed. (Also need to be differentiated from an analytical standpoint in one of the companion papers).**

Response: In responding to other reviewers' comments and synthesizing the text to stay within *Nature's* 2,500 word limit, we chose to remove these sentences from the Discussion section.

With regard to analyses in the companion papers, we have presented evidence related to pre- versus post-natal exposures from several perspectives across the three companion papers. First, substantial growth faltering at birth and highest incidence from birth to three months implicate pre-natal causes of growth faltering for both linear growth (Figure 3) and wasting (companion paper “Child wasting and concurrent stunting...”, Figures 3, 4). Second, in “Causes and consequences...” we have a section focused on age-varying effects of different exposures, summarized in that article's Figure 3. We studied how maternal stature, which likely integrates many of the key biologic exposures listed above in response to comment 12a, affected child growth faltering at birth and thereafter. We also quantified differences in the magnitude of associations between key exposures on growth outcomes measured at very young ages (0-6 months) vs. at older ages (6-24 months). Third, we assessed to what degree the association between parental stature (weight, height) and child growth was mediated by a child's anthropometry at birth (“Causes and consequences...” ED Figure 5). In the revised manuscripts, we have drawn a sharper distinction vis-à-vis the implications of the findings for pre- and post-natal exposures.

- c. **Paragraph starting line 348 – this is the whole point of the paper. This analysis reveals information on age trends in incidence, remission, and relapse and states as its purpose to then subsequently better inform intervention. If that is the case, that this work can help support improved evidence basis of interventions (which I believe it can), then this section of the discussion is the most important. What are the specific options that both ARE and ARE not included in WHO recommendations (vs. those of other nutrition and/or child health organizations)? How do they fit? How do they miss the mark? What do the results of this analysis tell us about *specific* blind spots in prevention and treatment with respect to age groups and growth dynamics? What are the specific knowledge/ data gaps that would be needed to be filled in order to make specific recommendations?**

Response: Thank you for this helpful suggestion. We have revised the Discussion on lines 317-325 as follows (bold font indicates the additions relevant to this comment):

“In this study, 25% of children became stunted between birth and age 6 months, yet few child nutrition interventions are recommended by the World Health Organization in this age range. In the neonatal period, those interventions include delayed cord clamping, neonatal vitamin K administration, and kangaroo mother care.³⁹ Beyond the neonatal period, the sole recommended intervention is exclusive breastfeeding,³⁹ which significantly reduces the risk of mortality and morbidity but has not been found to reduce infant stunting.^{4,40-43} Additional research is needed to identify interventions that prevent linear growth faltering between birth and 6 months including nutritional support of the lactating parent and the vulnerable infant. Interventions may need to focus on upstream risk factors, such as maternal pre-conception and prenatal health and nutrition, and microbiota.

*We found that 31% of children became stunted during the complementary feeding phase (age 6-24 months). Meta-analyses evaluating the effectiveness of interventions during complementary feeding on stunting prevalence and mean LAZ have reported modest impacts of lipid-based nutrient supplements,⁴⁴ modest or no impact of micronutrient supplementation,⁴⁵ and no impact of water and sanitation improvements, deworming, or maternal education.⁴⁵ **The dearth of postnatal interventions that effectively improve child linear growth motivates renewed efforts to identify alternative, possibly multisectoral interventions, and to improve intervention targeting and implementation.**^{46,47”}*

Reviewer Reports on the First Revision:

Referees' comments:

Referee #1 (Remarks to the Author):

The authors have systematically revised the analyses and presentation in response to the last comments. As in my earlier comments, I believe that pooling data is valuable, the cohorts pooled are of good quality and the authors are aware of, and state, their limitation many of which would affect any pooling study that uses existing data. In my view, the revised paper should (and can) address two key issues in its design, analysis and interpretation if it is to make a significant contribution to state of knowledge:

The revised paper has removed cohorts that are very old, specifically pre-1990 (or 1987 recruitment), and continues to pool by region. In my view, this doesn't resolve the issue how subgroups (or in practice a single subgroup, namely region) are selected and whether they are the appropriate way of stratifying the cohorts, nor was it the point of my earlier comment, which more broadly raised heterogeneity/similarity and what it means for how data are analyzed and results are interpreted. 1987 to 2017 is a long period through which child nutrition has changed substantially. And the regions selected are internally highly variable in their economies and environment – Brazil, in the time periods covered by this study, is not nearly similar to Peru; same goes for South Africa versus Zimbabwe. So there is little justification for regional pooling, at least as the sole way of stratifying data. Many readers, like this reviewer, may legitimately want to see the overall pooled result as well as other subgroups. To allow this, I suggest that the results are shown for all cohorts pooled, and then for a few, and not just one, potentially relevant subgroups (e.g., as done in Figure 3a for region): region can be one of them; two others that I think should be definitely shown are time period (e.g., decade or pre/post-MDGs) and especially subgroups based on mean length at birth which can tell us about whether starting off badly can dominate things. Others may include a measure of national income or child mortality in the country at the time of recruitment (or child mortality in the cohort if it exists). The study can then formally test the importance of these subgroups, through comparing subgroup pooled estimates or using a meta-regression, rather than having selected one (region) only with no assessment of its relevance compared to other legitimate subgroups. With the data already collected and organized, the work involved for this is relatively small, I would estimate a few days of analysis and some time to revise the paper.

Substantively, and partly related to design and subgroup comparisons, the study needs a more novel and actionable conclusion than "Early timing and low reversal rates emphasize the importance of preventive intervention delivery within the prenatal and early postnatal phases coupled with continued delivery of postnatal interventions through the first 1000 days of life." Without underestimating the importance of pooling studies, which I think is valuable, surely the work that some of the authors and others have done on birth size and on life course growth using both cohorts and repeated cross-sectional data has told us this, been summarized extensively in the Lancet Nutrition Series by some of the authors, and led to years of investment in the first 1000 days. Rather, we pool cohorts so that we can get precision and granularity on important epidemiological parameters. To achieve this, the subgroup analyses, and the "parameters" shown, should be selected to reveal actionable details that may not have come out of the decades of existing work (e.g., relevant age windows for vulnerability/resilience/recovery). Alternatively, does the additional precision of pooled cohorts convince us that all ages are equally important in all situations (regions, time periods, baseline birth lengths, etc) or does the role of age vary? The figures should then be presented to reveal and support these conclusions (e.g., can Figure 3a become more specific than a lot of age-region-specific points and test hypotheses? Is acting prior to birth really more important in south Asia than Africa/Latin America given the higher incidence proportion at birth in the region in Figure 3a? why is this the case based on what we know about epidemiology of stunting in this well-studied region? This should then be used to revised the text

on page 6 to go beyond being “more common”) In summary, if this is the definitive study of growth over age, then it should take a more definitive stand than state of knowledge on how/where/in what conditions age matters and under what conditions it doesn't matter. This sort of generalizability is essential if the study is to go beyond a description of how growth/stunting changed over age in cohorts who are now in their 20s or 30s and be generalizable to contemporary situations.

As a specific issues

“Eye-balling” the curves in Figure 2a for different cohorts and DHSs, the curves for south Asia don't seem that much lower than those in Africa. Acknowledging that eye-balling is not the way to draw quantitative conclusions, can the authors show the pooled regional length-for-age curve for regions to see if it is consistent with the much higher incidence proportion at birth for south Asia in Figure 3a?

The results in Figure 3b seem rather obvious by construction. If we stratify participants on age at which stunting happens, surely those that become stunted in older ages maintain higher mean HAZ than those that become stunted in younger ages, until the age of stunting! Given this tautological relationship, it seems unnecessary to present this as a “result”. Or is there some other message that is not obvious by how the children are grouped?

Referee #2 (Remarks to the Author):

The authors have made important edits to the manuscript and it reads well - concise and informative. Only a few edits/ comments remain:

Abstract, lines 42-44: There is no magnitude of growth faltering mentioned here which makes the <5% monthly reversal of stunting status without context.

Lines 210-215: Were there major differences between the children who met the eligibility criteria for the study vs. those originally enrolled in the study? Thinking through any issues here of selection bias.

Lines 259: In this section, something that seemed an important finding is that after 6 months, the proportion experiencing stunting reversal every month was the month, regardless of month of age. So there isn't more stunting reversal at age 8 months vs 9 month vs 14 months. Is this correct?

Lines 273-282 - This section would benefit from just a brief line from the authors as to why this approach was taken. What are you hoping to show readers by doing this? I understand why upon reading it over very, very carefully but it would be worthwhile being explicit.

Lines 386-297 - You do not discuss within child length velocity heterogeneity other than to say it was minimal after 6 months. This does not reflect findings from other studies so would be worth reflecting on (Ilana R Cliffer, William A Masters, Nandita Perumal, Elena N Naumova, Augustin N Zeba, Franck Garanet, Beatrice L Rogers, Monthly measurement of child lengths between 6 and 27 months of age in Burkina Faso reveals both chronic and episodic growth faltering, *The American Journal of Clinical Nutrition*, Volume 115, Issue 1, January 2022, Pages 94–104, <https://doi.org/10.1093/ajcn/nqab309>)

Referee #3 (Remarks to the Author):

I appreciate the thoughtfulness and comprehensiveness with which the authors have responded to my comments. I am satisfied and feel the paper is now much stronger. No further comments at this time.

Referee #4 (No remarks to the Author)

Referee #5 (Remarks to the Author):

This review focuses on the description of the analytical approach of pooling the data and related results.

- I believe there is an error on p19, in lines 148-149 that describe the different parameters in equation 1. Further, τ^2 should be defined in case readers are unfamiliar with these equations. This would also make the difference between equations 1 and 2 clearer.
- Please define all parameters in equation 2 in lines 159-160.
- The following statement is included: If a model failed to converge, models were fit using a maximum likelihood estimator instead. In what analyses did this occur?
- In multiple places it is stated that the results using random effects models were comparable to those using fixed effects models. This would not be surprising if there is little heterogeneity in the results across studies. It would be helpful to provide a measure of the heterogeneity in the study-specific results either in the text or in the tables/figures when pooled results are provided. For example, there appears to be quite a bit of heterogeneity in the study-specific results in figure 3 for the South Asia region and in extended data figure 10.
- When conducting pooled analyses, you can examine potential sources of heterogeneity if results were heterogeneous across studies. Did you investigate potential sources of heterogeneity if present? This would provide additional information beyond just generating a pooled estimate.
- In Fig 4b, you may want to clarify what the colored lines are in each panel.
- I may have missed it, but I couldn't find a description of the results for the following statement as was done for the other sensitivity analyses. This is particularly important if there are systematic differences in the results based on the number of measures within a study.
Second to explore the influence of differing numbers of cohorts contributing data at different ages, we conducted a sensitivity analysis in which we subset data to cohorts that measured anthropometry monthly from birth to 24 months (n=21 cohorts in 10 countries, 11,424 children.

Author Rebuttals to First Revision:

Referee #1 (Remarks to the Author):

1. The authors have systematically revised the analyses and presentation in response to the last comments. As in my earlier comments, I believe that pooling data is valuable, the cohorts pooled are of good quality and the authors are aware of, and state, their limitation many of which would affect any pooling study that uses existing data. In my view, the revised paper should (and can) address two key issues in its design, analysis and interpretation if it is to make a significant contribution to state of knowledge:

The revised paper has removed cohorts that are very old, specifically pre-1990 (or 1987 recruitment), and continues to pool by region. In my view, this doesn't resolve the issue how sub-groups (or in practice a single subgroup, namely region) are selected and whether they are the appropriate way of stratifying the cohorts, nor was it the point of my earlier comment, which more broadly raised heterogeneity/similarity and what it means for how data are analyzed and results are interpreted. 1987 to 2017 is a long period through which child nutrition has changed substantially. And the regions selected are internally highly variable in their economies and environment – Brazil, in the time periods covered by this study, is not nearly similar to Peru; same goes for South Africa versus Zimbabwe. So there is little justification for regional pooling, at least as the sole way of stratifying data. Many readers, like this reviewer, may legitimately want to see the overall pooled result as well as other subgroups. To allow this, I suggest that the results are shown for all cohorts pooled, and then for a few, and not just one, potentially relevant subgroups (e.g., as done in Figure 3a for region): region can be one of them; two others that I think should be definitely shown are time period (e.g., decade or pre/post-MDGs) and especially subgroups based on mean length at birth which can tell us about whether starting off badly can dominate things. Others may include a measure of national income or child mortality in the country at the time of recruitment (or child mortality in the cohort if it exists). The study can then formally test the importance of these subgroups, through comparing subgroup pooled estimates or using a meta-regression, rather than having selected one (region) only with no assessment of its relevance compared to other legitimate subgroups. With the data already collected and organized, the work involved for this is relatively small, I would estimate a few days of analysis and some time to revise the paper.

Response: We agree with the Reviewer that investigating patterns within additional subgroups could yield worthwhile insights. Our principal stratification by region reflected a consensus interest across the consortium because of programmatic differences in intervention strategies considered by ministries of health in different

regions, and because of potential differences in development of growth faltering that had been observed in different populations (esp. South Asia vs. Africa).

In response to this suggestion, we explored a variety of country-level indicators from the World Bank and the UN Development Programme as potential new stratification variables. This included subgroups suggested by the Reviewer (decade, child mortality, gross domestic product) as well as the gender inequality index, percentage of gross domestic product spent on health expenditures, percentage of the population living on less than \$1.90 per day, Gini coefficient, and coefficient of human inequality. We selected these variables because data were available for them in most years of the study period for most countries and because we hypothesized that they could plausibly modify the relationship between age and stunting onset. In addition, we investigated individual-level birth LAZ as another subgroup. Please see our response to Reviewer 1 comment 2 for additional details on stratification by birth LAZ.

We classified each country-level variable into subgroups using tertiles or alternative cut points that resulted in as balanced a distribution of study cohorts as possible. However, for some variables, it was not possible to tease apart differences in geographic region from differences in other study-level variables because many potential modifiers were highly correlated with geographic region (and in many cases, perfectly predicted by geographic region). It is therefore impossible to separate the effect of geography from other possible modifiers such as decade or GDP in these data. Please see the table at the end of this document for a summary of these distributions. To give one example, when stratifying by decade, very few studies from Africa were included in the 2010s, and very few studies from South Asia were included in the 1990s. In addition, in many cases, the range of country-level values was so narrow that it was not possible to create categories with meaningful differences (e.g., gender inequality index, coefficient of human inequality, GINI coefficient).

A further limitation is that the country-level indicators might not reflect the values for the study cohorts. The cohorts included in this study were not intended to be nationally representative. Though cohorts are broadly similar to Demographic and Health Survey Samples (Fig 2), they might exclude higher income populations. National level indicators average over potentially important heterogeneity within countries, but unfortunately we are not aware of readily available, more granular datasets that could be matched to our study populations.

Given these limitations, we selected three indicators for which subgroup categories contained a relatively balanced distribution of countries: % of GDP devoted to health expenditures, % of population living below \$1.90 US per day, and under-5 mortality.

Given the limitations we noted above, we chose to include these new results in the Extended Data (Extended Data Fig 8) instead of in the main text.

We added a brief summary of our methods to the Materials and Methods section lines 903-908:

“We obtained country-level data on the percentage of gross domestic product devoted to healthcare goods and spending from the United Nations Development Programme³ and the percentage of the country living on less than \$1.90 US per day and under-5 mortality rates from the World Bank.⁴ In years without available data, we linearly interpolated values from the nearest years with available data and extrapolated values within 5 years of available data using linear regression models based on all available years of data.”

We added the following results in the main text on lines 251-259:

“Early onset of stunting was consistent across geographic regions and countries with different levels of health spending, poverty, and under-5 mortality. Very early life stunting onset was most common in South Asia, where 20% of children were stunted at birth, and another 18% became stunted by age 3 months (Fig 3a). In Africa and Latin America, the percentage stunted at birth was lower than the percentage that became stunted between birth and age 3 months. In all regions, the rate of onset declined at subsequent ages. Overall, the proportion stunted at birth or by age 3 months was higher, and onset was lower at subsequent ages in countries with a lower proportion of gross domestic product devoted to health spending, higher child mortality, and a higher percentage of the population living on less than \$1.90 US per day (Extended Data Fig 8).”

We added the following results in the discussion on lines 334-338:

“we found that incident stunting onset was highest between birth and age 3 months, a pattern consistent across geographic regions, and was most pronounced in countries with a lower proportion of gross domestic product devoted to health spending, higher under-5 mortality rates, and higher poverty levels (Fig 3a, Extended Data Fig 8).”

And on lines 365-6:

“Our finding that stunting incidence at birth was lower in countries with a greater level of national health expenditures suggests that overall investments in health care systems may also improve linear growth.”

2. Substantively, and partly related to design and subgroup comparisons, the study needs a more novel and actionable conclusion than “Early timing and low reversal rates emphasize the importance of preventive intervention delivery within the prenatal and early postnatal phases coupled with continued delivery of postnatal interventions through the first 1000 days of life.” Without underestimating the importance of pooling studies, which I think is valuable, surely the work that some of the authors and others have done on birth size and on life course growth using both cohorts and repeated cross-sectional data has told us this, been summarized extensively in the Lancet Nutrition Series by some of the authors, and led to years of investment in the first 1000 days. Rather, we pool cohorts so that we can get precision and granularity on important epidemiological parameters. To achieve this, the subgroup analyses, and the “parameters” shown, should be selected to reveal actionable details that may not have come out of the decades of existing work (e.g., relevant age windows for vulnerability/resilience/recovery). Alternatively, does the additional precision of pooled cohorts convince us that all ages are equally important in all situations (regions, time periods, baseline birth lengths, etc) or does the role of age vary? The figures should then be presented to reveal and support these conclusions (e.g., can Figure 3a become more specific than a lot of age-region-specific points and test hypotheses? Is acting prior to birth really more important in south Asia than Africa/Latin America given the higher incidence proportion at birth in the region in Figure 3a? why is this the case based on what we know about epidemiology of stunting in this well-studied region? This should then be used to revise the text on page 6 to go beyond being “more common”) In summary, if this is the definitive study of growth over age, then it should take a more definitive stand than state of knowledge on how/where/in what conditions age matters and under what conditions it doesn’t matter. This sort of generalizability is essential if the study is to go beyond a description of how growth/stunting changed over age in cohorts who are now in their 20s or 30s and be generalizable to contemporary situations.

Response: Thank you for this helpful comment. First, we will address the Reviewer’s question: “does the additional precision of pooled cohorts convince us that all ages are equally important in all situations (regions, time periods, baseline birth lengths, etc) or does the role of age vary?” Our additional analyses in response to comment #1 showed that the age of early stunting onset was consistent across a range of different subgroups. We also revised Figure 4, which is now split into Figures 4 and 5. The new Figure 4 includes incidence of stunting, stunting relapse, and stunting reversal overall (as shown in previous Figure 4b) as well as stratified by birth LAZ category (new Fig 4b) and by region (new Extended Data Fig 10). We agree with the

reviewer that it would be interesting to examine stratification by both birth LAZ and region, but unfortunately, because this analysis includes a subset of cohorts with monthly measurements from 0-15 months, the sample size was not sufficient. Taken together, we believe that these new analyses strengthen our conclusion that a greater focus on the prenatal period is needed to prevent stunting onset in early life, especially in South Asia.

We revised the results section of the manuscript on lines 280-300 as follows:

“New incidence of stunting was highest at birth and declined steadily to 3.3% per month by age 4 months (Fig 4a), a pattern that was most marked in South Asia (Extended Data Fig 10). Incidence rates of new and relapse stunting exceeded rates of reversal at all ages, new results that illustrate the underlying dynamics of a gradually accumulating stunting burden as children age: by 15 months 34.0% of children were stunted, 50.5% had ever been stunted, and 16.5% had experienced stunting reversal and were no longer stunted (Fig 4a). Incident stunting relapse following reversal ranged from 2.0-3.5% per month from ages 6 to 15 months, and patterns were similar across regions (Extended Data Fig 10). In South Asia, stunting reversal declined as children aged but rates were stable across ages in Africa and Latin America; overall reversal was slightly less common in Latin America (Extended Data Fig 10).”

To assess whether a child’s birth length influenced their propensity to recover from stunting, we summarized incident stunting, relapse, and reversal rates stratified by birth LAZ subgroup in monthly measured cohorts (Fig 4b). Eighty-six percent of children who ever became stunted had LAZ <0 at birth. Rates of stunting relapse increased with age and were generally higher among children who were born stunted. Stunting reversal was more common at young ages for children born with LAZ < -2, which likely reflects regression to the mean. After age 6 months, stunting reversal rates were similarly low among children with birth LAZ < -2 (<7% per month) and birth LAZ -2 to 0 (<5% per month). These results suggest that linear growth faltering at birth is a key determinant of children’s linear growth trajectories in early life, recovery is rare among all children who become stunted through age 15 months, and children who are stunted at birth are more prone to transient stunting reversal followed by stunting relapse.”

We also modified the discussion section as follows on lines 332-356:

“This large-scale analysis of 32 longitudinal cohorts from LMICs revealed new insights into the timing, persistence, and recurrence of linear growth faltering from

birth to age 2 years. Prior cross-sectional studies found that stunting prevalence increased gradually with age.^{15,20–22} In contrast, we found that incident stunting onset was highest between birth and age 3 months, a pattern consistent across geographic regions, and was most pronounced in countries with a lower proportion of gross domestic product devoted to health spending, higher under-5 mortality rates, and higher poverty levels (Fig 3a, Extended Data Fig 8). Stunting at birth was a key predictor of children’s linear growth trajectories through age 15 months: stunting relapse in the first year of life was substantially higher among children who were stunted at birth compared to those who were not born stunted (Fig 4b). The burden and persistence of very early life linear growth faltering was most stark in South Asia, where 20% of children were stunted at birth (Fig 3a) and children who were stunted at birth had a mean LAZ of approximately -2.5 at all subsequent ages, substantially lower than children in other regions (Fig 3b). Most children who experienced stunting reversal continued to experience linear growth deficits, and over 20% who achieved reversal were stunted again at later measurements (Fig 5a). Even among children who never met criteria for stunting, mean LAZ steadily declined by over 0.5 z by age 15 months (Fig 3b) — a result that shows linear growth faltering among children in LMICs is a whole-population phenomenon, with both stunted and not stunted children experiencing suboptimal growth trajectories in early life.²¹

Two key conclusions from the 2021 Lancet series on child maternal and child undernutrition³⁴ were that improving children’s linear growth will require a life course approach with an emphasis on women’s health and that targeting interventions by age and geography may yield greater benefits than one-size-fits-all approaches. Our results provide new quantitative evidence that strengthen these conclusions and enable more precise statements about the extent of the whole-population burden, age windows for preventive interventions, and the uniquely high incidence and low reversal rates among children in South Asia compared with other geographic regions.”

With regard to the Reviewer’s comment about whether acting prior to birth really is more important in south Asia than Africa/Latin America, we added the following paragraph to lines 368-382 of the Discussion:

“In South Asia in particular, where stunting at birth was highest, intervening to improve the health of women of childbearing age may be critical to improving children’s linear growth. Prior work has identified South Asian women’s nutrition prior to and during pregnancy and poor sanitation conditions as key contributors to stunting at birth.⁴¹ However, in 2020 the prevalence of open defecation was 18% in Sub-Saharan Africa, 12% in South Asia, and 2% in Latin America, and access to basic sanitation was lower in Sub-Saharan Africa than in South Asia.⁴² Recent trials found that improving household-level sanitation did not improve children’s linear growth, but studies did not measure impacts on mothers.⁴³ A more likely explanation for

higher stunting at birth in South Asia is women's nutritional status. Prevalence of low body mass index in women is highest in South Asia (24%), with much higher prevalence in some geographic hot spots.³⁴ In addition, 40-70% of women in South Asia are less than 150 cm tall,⁴⁴ and the prevalence of infants born small for gestational age is 34% in South Asia compared to 17% in Sub-Saharan Africa and 9% in Latin America.⁴⁵ Our analysis of risk factors for stunting in a companion paper in this series reports that maternal height, weight, and body mass index were the strongest predictors of stunting at birth and child linear growth trajectories.²⁹ These findings point to the need to tailor interventions to the unique factors influencing women's nutrition and prenatal health in South Asia."

In response to the Reviewer's comment that our conclusions should take a more definitive stand on how/where/in what conditions age matters and under what conditions it doesn't matter, we have revised the abstract as follows and made a similar revision to the concluding sentences of the manuscript (lines 428-432):

"Our findings suggest that defining stunting targets at earlier ages (e.g., stunting by 3 or 6 months) would help focus attention on the period when interventions may be most impactful. In addition, our results motivate a life course approach that targets interventions to women of childbearing age, includes the youngest children during their first months of life, and has a special focus on children in South Asia where the burden is highest and stunting reversal is lowest."

3. "Eye-balling" the curves in Figure 2a for different cohorts and DHSs, the curves for south Asia don't seem that much lower than those in Africa. Acknowledging that eye-balling is not the way to draw quantitative conclusions, can the authors show the pooled regional length-for-age curve for regions to see if it is consistent with the much higher incidence proportion at birth for south Asia in Figure 3a?

Response: Thank you for this suggestion. We have added the pooled ki curves to

each panel of Figure 2 to facilitate comparison. On lines 227-8 we added the

sentence: *"The distribution of LAZ was shifted to the left for ki cohorts in South Asia compared to those in Latin America and Africa."*

4. The results in Figure 3b seem rather obvious by construction. If we stratify participants on age at which stunting happens, surely those that become stunted in older ages maintain higher mean HAZ than those that become stunted in younger ages, until the age of stunting! Given this tautological relationship, it seems unnecessary to present this as a “result”. Or is there some other message that is not obvious by how the children are grouped?

Response: We agree that the original figure could be improved. We chose to revise figure 3b to use fewer ages of stunting onset and to show mean LAZ by age as well as region. We feel that this revised figure is valuable because it shows that children stunted at birth in South Asia fare worse at all subsequent ages compared to children in other regions, which supports our conclusion that stunting at birth is a key driver of linear growth patterns in South Asia. In addition, the new age categories are more clearly linked to ages at which different interventions are targeted (birth, exclusive breastfeeding phase from 0-6 months, complementary feeding phase thereafter). An important new finding, still displayed in the revised figure and now shown for each geographic region, is the extent of linear growth faltering among children who are never classified as “stunted” — children in this group still lost, on average, over 0.5 z by age 15 months. This is important because it shows quantitatively how linear growth faltering is a whole population phenomenon.

We revised the results text as follows on lines 260-271:

“We summarized age trends in LAZ stratified by geographic region and timing of stunting onset (Fig 3b; Extended Data Fig 9). Among children stunted at birth, LAZ differed markedly between geographic regions: mean LAZ rose in the first month of life in all regions and then remained close to -0.5 in Latin America, close to -2 in Africa, and close to -2.5 in South Asia. Regional differences were less pronounced among children stunted at later ages, though children in South Asian cohorts had consistently lower mean LAZ than children from African and Latin American cohorts. Children who became stunted between birth and age 6 months started at low birth LAZ (mean = -2.7) and had moderate rates of decline, whereas children who became stunted between ages 6 and 15 months started at higher birth LAZ (mean = -1.4) but had much faster rates of decline in LAZ, from above -1 z at birth to below -2 z by age 15 months. Children who were never stunted still experienced a drop of approximately 0.5 z in mean LAZ from birth to 15 months in all regions, showing that even children not classified as “stunted” on average experienced significant, postnatal linear growth faltering.

We revised the discussion text as follows on lines 339-347:

“The burden and persistence of very early life linear growth faltering was most stark in South Asia, where 20% of children were stunted at birth (Fig 3a) and children who were stunted at birth had a mean LAZ of approximately -2.5 at all subsequent ages, substantially lower than children in other regions (Fig 3b). Most children who experienced stunting reversal continued to experience linear growth deficits, and over 20% who achieved reversal were stunted again at later measurements (Fig 5a). Even among children who never met criteria for stunting, mean LAZ steadily declined by over 0.5 z by age 15 months (Fig 3b) — a result that shows linear growth faltering among children in LMICs is a whole-population phenomenon, with both stunted and not stunted children experiencing suboptimal growth trajectories in early life.²¹

Referee #2 (Remarks to the Author):

The authors have made important edits to the manuscript and it reads well - concise and informative. Only a few edits/ comments remain:

1. Abstract, lines 42-44: There is no magnitude of growth faltering mentioned here which makes the <5% monthly reversal of stunting status without context.

Response: We have revised this to say:

“From 0 to 15 months, stunting reversal was rare; children who reversed their stunting status frequently relapsed, and relapse rates were substantially higher among children born stunted.”

2. Lines 210-215: Were there major differences between the children who met the eligibility criteria for the study vs. those originally enrolled in the study? Thinking through any issues here of selection bias.

Response: To clarify this, we added the following to line 840 of the Materials and Methods section.

“All children from each eligible cohort were included in the study.”

3. Lines 259: In this section, something that seemed an important finding is that after 6 months, the proportion experiencing stunting reversal every month was the month, regardless of month of age. So there isn't more stunting reversal at age 8 months vs 9 month vs 14 months. Is this correct?

Response: We agree that the point estimate for the incidence of stunting reversal is slightly higher at 8 months than at subsequent ages. However, given the overlap of the 95% confidence intervals for estimates from 8-15 months, we consider the incidence of stunting reversal to be similar during this age range.

4. Lines 273-282 - This section would benefit from just a brief line from the authors as to why this approach was taken. What are you hoping to show readers by doing this? I understand why upon reading it over very, very carefully but it would be worthwhile being explicit.

Response: We added the following text to motivate this analysis on lines 289-291:

“To assess whether a child’s birth length influenced their propensity to recover from stunting, we summarized incident stunting, relapse, and reversal rates stratified by birth LAZ subgroup in monthly measured cohorts (Fig 4b).”

5. Lines 286-297 - You do not discuss within child length velocity heterogeneity other than to say it was minimal after 6 months. This does not reflect findings from other studies so would be worth reflecting on (Ilana R Cliffer, William A Masters, Nandita Perumal, Elena N Naumova, Augustin N Zeba, Franck Garanet, Beatrice L Rogers, Monthly measurement of child lengths between 6 and 27 months of age in Burkina Faso reveals both chronic and episodic growth faltering, The American Journal of Clinical Nutrition, Volume 115, Issue 1, January 2022, Pages 94–104, <https://doi.org/10.1093/ajcn/nqab309>)

Response: we appreciate this interesting comment and reflected on whether it would be feasible to explore child-specific velocity curves. However, given space constraints and the size of our dataset, we decided to focus on summary measures of length and LAZ velocity, which average across individual children's velocity curves. We revised the results paragraph on lines 316-327 focused on growth velocity to make it clearer that the results reflect averages of within-child velocity measurements (new/revised text in bold).

*"We defined linear growth velocity as **a child's** change in length between two time points divided by the number of months between the time points (cm/month). From 0-3 months, cohort-specific length velocity ranged from below the 1st percentile of the WHO standard to above the 50th for boys and above the 75th percentile for girls (Fig 6a). At subsequent ages, length velocity in each cohort was mostly between the 15th and 50th percentiles of the WHO standard, except in one cohort in Belarus, which had higher length velocity. Larger deficits at the youngest ages were consistent with highest incidence of stunting from birth to age 3 months (Fig 3a). **From ages 3 to 24 months, on average, children's change in length was between 0.75 and 1.25 cm per month. We also estimated within-child rates of LAZ change per month, which compares changes in a child's length relative to the WHO standard over time. The difference in LAZ within child per month was largest from 0-3 months; after age 3 months, the mean change in LAZ within child was <0.3 between different age intervals (Fig 6b). Generally, velocity within age was higher in Latin America than in South Asia and Africa (Extended Data Fig 12)."***

Referee #3 (Remarks to the Author):

6. I appreciate the thoughtfulness and comprehensiveness with which the authors have responded to my comments. I am satisfied and feel the paper is now much stronger. No further comments at this time.

Referee #4 (No remarks to the Author)

Referee #5 (Remarks to the Author):

This review focuses on the description of the analytical approach of pooling the data and related results.

7. I believe there is an error on p19, in lines 148-149 that describe the different parameters in equation 1. Further, Tau2 should be defined in case readers are unfamiliar with these equations. This would also make the difference between equations 1 and 2 clearer.

Response: Thank you for pointing this out. We have revised this section as follows on lines 148-155:

“Random effects models assume that the true population outcomes ϑ are normally distributed ($\vartheta \sim N(\mu, \tau^2)$), where ϑ has mean μ and variance τ^2 . To estimate outcomes in this study, the random effects model is defined as follows for each study in the set of $i = 1, \dots, k$ studies:

$$y_i = \mu + u_i + e_i \quad (1)$$

where y_i is the observed outcome in study i , u_i is the random effect for study i , μ is the estimated outcome for study i , and e_i is the sampling error within study i .”

8. Please define all parameters in equation 2 in lines 159-160.

Response: We have revised this section as follows on lines 163-169:

$$\bar{\theta}_w = \frac{\sum_{i=1}^k w_i \theta_i}{\sum_{i=1}^k w_i}$$

where $\bar{\theta}_{ms}$ is the weighted mean outcome in the set of k included studies, and w_i is a study-specific weight, defined as the inverse of the study-specific sampling variance v_i . $\hat{\theta}_i$ is the estimate from study i . $\bar{\theta}_{ms}$ is the estimated mean outcome in the specific studies included in this analysis.

9. The following statement is included: If a model failed to converge, models were fit using a maximum likelihood estimator instead. In what analyses did this occur?

Response: In all analyses except for those presented in Figure 4a, the restricted maximum likelihood estimator converged. Analyses in Figure 4a had a higher degree of data sparsity because analyses were restricted to cohorts with monthly measurements from birth to 15 months and stratified by stunting status at each age in months as well as birth LAZ category. In addition, in some analyses with zero stunting cases, random effects models failed to converge, so we used fixed effects models. In the Materials and Methods, the revised text on lines 973-975 is:

“If a model failed to converge, we attempted to fit models with a maximum likelihood estimator. If random effects models failed to converge due to zero stunting cases, we used a fixed effects estimator.”

We also included the following description in the Figure 4 caption:

“The black line presents estimates pooled using random effects with restricted maximum likelihood estimation (N=168 models); in 11 models, alternative pooling methods were used because restricted maximum likelihood estimator did not converge (fixed effects N=8 models; maximum likelihood N=3 models).”

10. In multiple places it is stated that the results using random effects models were comparable to those using fixed effects models. This would not be surprising if there is little heterogeneity in the results across studies. It would be helpful to provide a measure of the heterogeneity in the study-specific results either in the text or in the tables/figures when pooled results are provided. For example, there appears to be quite a bit of heterogeneity in the study-specific results in figure 3 for the South Asia region and in extended data figure 10.

Response: We agree that we would expect the random effects and fixed effects models to produce similar results if there is little heterogeneity. In the online

supplemental materials (<https://child-growth.github.io/stunting/fixed-effects.html>), we did not mean to imply the estimates themselves are similar between each type of model, but rather that the two approaches did not lead us to make substantially different scientific inferences. We have added the following to this online supplemental materials page:

“The pooled estimates using random effects vs. fixed effects differed in some cases, indicating the presence of heterogeneity in underlying cohort-specific estimates. For example, stunting incidence at ages peaked at ages 0-3 months in Latin America using random effects models, but in fixed effects models, incidence was similar at ages 0-12 months. However, overall, our scientific inferences from results produced by each method were similar.”

We agree that the degree of heterogeneity between studies is of interest and considered reporting an estimate of heterogeneity (e.g., I-squared statistic). However, separate estimates would be required for each age in each subgroup (e.g., region). We feel that including this information on our figures would make them harder to read and that our inclusion of cohort-specific estimates alongside the pooled estimates at each age sufficiently represents cohort-specific heterogeneity.

11. When conducting pooled analyses, you can examine potential sources of heterogeneity if results were heterogeneous across studies. Did you investigate potential sources of heterogeneity if present? This would provide additional information beyond just generating a pooled estimate.

Response: While traditional meta-analyses typically investigate heterogeneity between studies at each analysis, in this study, the large number of meta-analyses prevented us from doing so on a study-by-study basis for every single category. To give an example, Figure 3a essentially contains 36 distinct meta-analyses (9 age categories x 4 panels). We included cohort-specific estimates in as many figures as possible, which allows readers to visually examine heterogeneity in each analysis. In addition, we examined heterogeneity by investigating whether there was modification of age-specific linear growth patterns across geographic regions and other country-level variables (see response to Reviewer 1, comment 1).

12. In Fig 4b, you may want to clarify what the colored lines are in each panel.

Response: The figure caption for 4b describes these: *“The black line presents estimates pooled using random effects with restricted maximum likelihood estimation. Colored lines indicate cohort-specific estimates.”*

13. I may have missed it, but I couldn't find a description of the results for the following statement as was done for the other sensitivity analyses. This is particularly important if there are systematic differences in the results based on the number of measures within a study.

Second to explore the influence of differing numbers of cohorts contributing data at different ages, we conducted a sensitivity analysis in which we subset data to cohorts that measured anthropometry monthly from birth to 24 months (n=21 cohorts in 10 countries, 11,424 children).

Response: We described this in the limitations paragraph in the Discussion section on lines 415-419:

“Fourth, the included cohorts measured child length every 1-3 months, and ages of measurement varied, so different numbers of children and cohorts contributed to each estimate. However, when we repeated analyses in cohorts with monthly measurements from birth to 24 months (n=18 cohorts in 10 countries, 10,830 children), results were similar (<https://child-growth.github.io/stunting/monthly.html>).”

Table. Number of children included in each stratification variable category and region

	Africa (N=21759)	South Asia (N=40279)	Latin America (N=1619)	Europe (N=16898)	Overall (N=80555)
Decade					
1990s	16364 (75.2%)	2233 (5.5%)	868 (53.6%)	16898 (100%)	36363 (45.1%)
2000s	5063 (23.3%)	23108 (57.4%)	423 (26.1%)	0 (0%)	28594 (35.5%)
2010s	332 (1.5%)	14938 (37.1%)	328 (20.3%)	0 (0%)	15598 (19.4%)
Gross domestic product per capita (in millions of USD)					
< \$1,026	21445 (98.6%)	38119 (94.6%)	0 (0%)	0 (0%)	59564 (73.9%)
≥ \$1,026	314 (1.4%)	2160 (5.4%)	1619 (100%)	16898 (100%)	20991 (26.1%)
Gender development index					
< 10	2223 (10.2%)	25265 (62.7%)	0 (0%)	0 (0%)	27488 (34.1%)
10 – 16	14898 (68.5%)	15014 (37.3%)	315 (19.5%)	0 (0%)	30227 (37.5%)
≥ 16	821 (3.8%)	0 (0%)	1009 (62.3%)	16898 (100%)	18728 (23.2%)
Missing	3817 (17.5%)	0 (0%)	295 (18.2%)	0 (0%)	4112 (5.1%)
Gender inequality index					
< 0.596	10378 (47.7%)	2606 (6.5%)	0 (0%)	0 (0%)	12984 (16.1%)
0.596 – 0.614	7250 (33.3%)	14604 (36.3%)	0 (0%)	0 (0%)	21854 (27.1%)
≥ 0.614	314 (1.4%)	23069 (57.3%)	1522 (94.0%)	0 (0%)	24905 (30.9%)
Missing	3817 (17.5%)	0 (0%)	97 (6.0%)	16898 (100%)	20812 (25.8%)
Coefficient of human inequality					
< 28.3	2117 (9.7%)	7255 (18.0%)	358 (22.1%)	0 (0%)	9730 (12.1%)
28.3 – 29.3	0 (0%)	9959 (24.7%)	0 (0%)	0 (0%)	9959 (12.4%)
≥ 29.3	0 (0%)	20459 (50.8%)	393 (24.3%)	0 (0%)	20852 (25.9%)
Missing	19642 (90.3%)	2606 (6.5%)	868 (53.6%)	16898 (100%)	40014 (49.7%)
GINI coefficient					
< 32.32	0 (0%)	30713 (76.3%)	0 (0%)	0 (0%)	30713 (38.1%)
32.32 – 32.54	0 (0%)	5899 (14.6%)	0 (0%)	16898 (100%)	22797 (28.3%)
≥ 32.54	7107 (32.7%)	3667 (9.1%)	1409 (87.0%)	0 (0%)	12183 (15.1%)
Missing	14652 (67.3%)	0 (0%)	210 (13.0%)	0 (0%)	14862 (18.4%)
Total expenditure on health (as % of gross domestic product)					
1 – 3%	0 (0%)	28901 (71.8%)	0 (0%)	0 (0%)	28901 (35.9%)
3 – 5%	1337 (6.1%)	11098 (27.6%)	742 (45.8%)	0 (0%)	13177 (16.4%)
≥ 5%	5399 (24.8%)	0 (0%)	582 (35.9%)	16898 (100%)	22879 (28.4%)
Missing	15023 (69.0%)	280 (0.7%)	295 (18.2%)	0 (0%)	15598 (19.4%)
% of population living on below \$1.90 per day					
< 18%	314 (1.4%)	7891 (19.6%)	1207 (74.6%)	16898 (100%)	26310 (32.7%)
18 – 28%	129 (0.6%)	29339 (72.8%)	202 (12.5%)	0 (0%)	29670 (36.8%)
≥ 28%	6664 (30.6%)	3049 (7.6%)	0 (0%)	0 (0%)	9713 (12.1%)
Missing	14652 (67.3%)	0 (0%)	210 (13.0%)	0 (0%)	14862 (18.4%)
Under-5 mortality rate					
<50 per 100,000	196 (0.9%)	22027 (54.7%)	852 (52.6%)	16898 (100%)	39973 (49.6%)
50-95 per 100,000	4250 (19.5%)	16591 (41.2%)	767 (47.4%)	0 (0%)	21608 (26.8%)
>95 per 100,000	17313 (79.6%)	1661 (4.1%)	0 (0%)	0 (0%)	18974 (23.6%)

Reviewer Reports on the Second Revision:

Referees' comments:

Referee #1 (Remarks to the Author):

The additional analyses are a valuable addition and help better understand what features are associated with higher/lower stunting. The only observation that I make on the revised manuscript is that there is as much or more variability in stunting dynamics among cohorts in each of the analysis regions as there is between them. Given this, and the results of the new analyses, I would suggest that the role of region is de-emphasized in the paper's conclusions and in its abstract; rather, what seems to matter is the economic and epidemiological features of each population, including those that affect pregnancy and birth conditions. These are on average worse in south Asia but there are cohorts in other regions that do worse south Asia as a whole, and certainly worse than the better off cohorts in this region. I also suggest that the table at the end of the rebuttal/response document included in the paper as a supplement; these are useful information and should be available to the readers beyond the reviewers.

Referee #5 (Remarks to the Author):

Overall, the responses to the comments about the analytical approach of pooling the data are clear. However, one remaining concern regards the following response to comment #10 from referee 5: The pooled estimates using random effects vs. fixed effects differed in some cases, indicating the presence of heterogeneity in underlying cohort-specific estimates. For example, stunting incidence at ages peaked at ages 0-3 months in Latin America using random effects models, but in fixed effects models, incidence was similar at ages 0-12 months. However, overall, our scientific inferences from results produced by each method were similar.”

We agree that the degree of heterogeneity between studies is of interest and considered reporting an estimate of heterogeneity (e.g., I-squared statistic). However, separate estimates would be required for each age in each subgroup (e.g., region). We feel that including this information on our figures would make them harder to read and that our inclusion of cohort-specific estimates alongside the pooled estimates at each age sufficiently represents cohort-specific heterogeneity.

The statement that “stunting incidence at ages peaked at ages 0-3 months in Latin America using random effects models, but in fixed effects models, incidence was similar at ages 0-12 months” seems to disagree with the overall conclusion of the manuscript (lines 428-429):

Our findings suggest that defining stunting targets at earlier ages (e.g., stunting by 3 or 6 months) would help focus attention on the period when interventions may be most impactful.

It could be that the pattern in Latin America is a bit different from the other regions, but these conclusions (incidence was similar at ages 0-12 months vs stunting targets at earlier ages (e.g., stunting by 3 or 6 months) do not appear consistent.

I am assuming a measure of the heterogeneity between the study results was generated for each analysis. If that is the case, the authors could potentially provide an indication of whether results were heterogeneous as a general statement for different figures such as providing the range of I² or the median and interquartile or interdecile range for the I² in each figure/analysis. This could further provide information on heterogeneity beyond the reader's personal interpretation of the results presented in the figures.

Author Rebuttals to Second Revision:

Referee #1 (Remarks to the Author):

The additional analyses are a valuable addition and help better understand what features are associated with higher/lower stunting. The only observation that I make on the revised manuscript is that there is as much or more variability in stunting dynamics among cohorts in each of the analysis regions as there is between them. Given this, and the results of the new analyses, I would suggest that the role of region is de-emphasized in the paper's conclusions and in its abstract; rather, what seems to matter is the economic and epidemiological features of each population, including those that affect pregnancy and birth conditions. These are on average worse in south Asia but there are cohorts in other regions that do worse south Asia as a whole, and certainly worse than the better off cohorts in this region. I also suggest that the table at the end of the rebuttal/response document is included in the paper as a supplement; these are useful information and should be available to the readers beyond the reviewers.

Response: We have de-emphasized the discussion of region in the abstract and conclusion of the manuscript. The final sentence of the abstract now reads as follows:

“Early onset and low reversal rates suggest that improving children’s linear growth will require life course interventions for women of childbearing age and a greater emphasis on interventions for children under 6 months.”

The conclusion of the manuscript now reads as follows:

“Current WHO 2025 Global Nutrition Targets and Sustainable Development Goal 2.2.1 aim to reduce stunting prevalence among children under 5 years by 2025. Our findings suggest that defining stunting targets at earlier ages (e.g., stunting by 3 or 6 months) would help focus attention on the period when interventions may be most impactful. In addition, our results motivate a life course approach that targets interventions to women of childbearing age and includes interventions for children during their first months of life.”

We also included the table of the distribution of study children by subgroup and region (Extended Data Table 3). We also included this text on lines 92-97 of the Materials and Methods section:

“We also considered additional subgroups, including decade in which data was collected, gross domestic product,⁴ gender development index,³ gender inequality index,³ coefficient of human inequality,⁴ and the GINI coefficient.⁴ However, for these variables, subgroup levels were strongly correlated with geographic region, making it impossible to separate the effects of each (Extended Data Table 3). Thus, we did not conduct subgroup analyses for these variables.”

Referee #5 (Remarks to the Author):

Overall, the responses to the comments about the analytical approach of pooling the data are clear. However, one remaining concern regards the following response to comment #10 from referee 5:

[previous comment 10] The pooled estimates using random effects vs. fixed effects differed in some cases, indicating the presence of heterogeneity in underlying cohort-specific estimates. For example, stunting incidence at ages peaked at ages 0-3 months in Latin America using random effects models, but in fixed effects models, incidence was similar at ages 0-12 months. However, overall, our scientific inferences from results produced by each method were similar.”

[previous author response] We agree that the degree of heterogeneity between studies is of interest and considered reporting an estimate of heterogeneity (e.g., I-squared statistic). However, separate estimates would be required for each age in each subgroup (e.g., region). We feel that including this information on our figures would make them harder to read and that our inclusion of cohort-specific estimates alongside the pooled estimates at each age sufficiently represents cohort-specific heterogeneity.

The statement that “stunting incidence at ages peaked at ages 0-3 months in Latin America using random effects models, but in fixed effects models, incidence was similar at ages 0-12 months” seems to disagree with the overall conclusion of the manuscript (lines 428-429):
Our findings suggest that defining stunting targets at earlier ages (e.g., stunting by 3 or 6 months) would help focus attention on the period when interventions may be most impactful.

It could be that the pattern in Latin America is a bit different from the other regions, but these conclusions (incidence was similar at ages 0-12 months vs stunting targets at earlier ages (e.g., stunting by 3 or 6 months) do not appear consistent.

I am assuming a measure of the heterogeneity between the study results was generated for each analysis. If that is the case, the authors could potentially provide an indication of whether results were heterogeneous as a general statement for different figures such as providing the range of I² or the median and interquartile or interdecile range for the I² in each figure/analysis. This could further provide information on heterogeneity beyond the reader’s personal interpretation of the results presented in the figures.

Response: We have added the median and interquartile range for the I-squared statistic to all main text figures that present meta-analyses.

Regarding the sentence about age-specific incidence in Latin America in the online supplement, we believe that the previous statement requires some revision. We have revised it as follows based on the figures below, and we believe it is not consistent with the conclusions of the manuscript.

“For example, stunting incidence at ages peaked at ages 0-3 months in Latin America using random effects models, but in fixed effects models, incidence was similar at ages 0-3 and 3-6 months.”

9.2.1 Random effects

9.2.2 Fixed effects